# Visual attention modulates the integration of goal-relevant evidence and not value

Pradyumna Sepulveda[1]*, Marius Usher[2], Ned Davies[1], Amy A Benson[1], Pietro Ortoleva[3], Benedetto De Martino[1,4]*

[1]Institute of Cognitive Neuroscience, University College London, London, United Kingdom; [2]School of Psychological Sciences and Sagol School of Neuroscience, Tel Aviv University, Tel Aviv, Israel; [3]Department of Economics and Woodrow Wilson School, Princeton University, Princeton, United States; [4]Wellcome Centre for Human Neuroimaging, University College London, London, United Kingdom

**Abstract** When choosing between options, such as food items presented in plain view, people tend to choose the option they spend longer looking at. The prevailing interpretation is that visual attention increases value. However, in previous studies, 'value' was coupled to a behavioural goal, since subjects had to choose the item they preferred. This makes it impossible to discern if visual attention has an effect on value, or, instead, if attention modulates the information most relevant for the goal of the decision-maker. Here, we present the results of two independent studies—a perceptual and a value-based task—that allow us to decouple value from goal-relevant information using specific task-framing. Combining psychophysics with computational modelling, we show that, contrary to the current interpretation, attention does not boost value, but instead it modulates goal-relevant information. This work provides a novel and more general mechanism by which attention interacts with choice.

*For correspondence:
p.sepulveda@ucl.ac.uk (PS);
benedettodemartino@gmail.com
(BDM)

## Introduction

How is value constructed and what is the role played by visual attention in choice? Despite their centrality to the understanding of human decision-making, these remain unanswered questions. Attention is thought to play a central role, prioritising and enhancing which information is accessed during the decision-making process. How attention interacts with value-based choice has been investigated in psychology and neuroscience (*Krajbich et al., 2010*; *Krajbich and Rangel, 2011*; *Cavanagh et al., 2014*; *Polanía et al., 2014*; *Gluth et al., 2015*; *Gluth et al., 2020*; *Folke et al., 2017*; *Tavares et al., 2017*; *Glickman et al., 2018*; *Gluth et al., 2018*; *Thomas et al., 2019*) and this question is at the core of the theory of rational inattention in economics (*Sims, 2003*; *Sims, 2010*; *Caplin and Dean, 2015*; *Hébert and Woodford, 2017*).

In this context, robust empirical evidence has shown that people tend to look for longer at the options with higher values (*Anderson et al., 2011*; *Gluth et al., 2018*; *Gluth et al., 2020*) and that they tend to choose the option they pay more visual attention to (*Krajbich et al., 2010*; *Krajbich and Rangel, 2011*; *Folke et al., 2017*; *Cavanagh et al., 2014*; *Thomas et al., 2019*). The most common interpretation is that attention is allocated to items based on their value and that looking or attending to an option boosts its value, either by amplifying it (*Krajbich et al., 2010*; *Krajbich and Rangel, 2011*; *Smith and Krajbich, 2019*) or by shifting it upwards by a constant amount (*Cavanagh et al., 2014*). This intuition has been elegantly formalised using models of sequential sampling, in particular the attentional drift diffusion model (aDDM), which considers that visual attention boosts the drift rate of the stochastic accumulation processes (*Krajbich et al., 2010*).

More recently, this same model has been also used to study the role of attention in the accumulation of perceptual information (*Tavares et al., 2017*). These lines of investigation have been extremely fruitful, as they have provided an elegant algorithmic description of the interplay between attention and choice.

As consequence of this development, the predominant assumption in the field of neuroeconomics has become that attention operates over the value of the alternatives (*Smith and Krajbich, 2019*). However, this view overlooks the fact that in the majority of these studies, value is coupled to the agents' behavioural goal, that is, participants had to choose the item they found more rewarding. However, some recent studies have called into question this assumption and have hinted towards a flexible role of attention on sampling goal-relevant options (*Kovach et al., 2014*; *Glickman et al., 2018*). Even further, recent developments have shown that the 'value networks' in the brain could be tracking not purely reward value, but actually goal-congruent information (*Frömer et al., 2019*; *Suri et al., 2020*). Considering all this, our study aims to understand in more detail the role of goals on visual attention during both value-based and perceptual decisions: we aim to test the hypothesis that attention acts in a flexible way upon the accumulation of *goal-relevant information* and to examine the effects on the mechanism of preference formation and confidence.

Our experimental design decouples reward value from choice by means of a simple task-framing manipulation. In the main eye-tracking part of our value-based experiment, participants were asked to choose between different pairs of snacks. We used two frame manipulations: *like* and *dislike*. In the *like* frame, they had to indicate which snack they would like to consume at the end of the experiment; this is consistent with the standard tasks used in value-based decision studies. But in the *dislike* frame, subjects had to indicate the snack that they would prefer *not* to eat, equivalent to choosing the other option. Crucially, in the latter frame value is distinct from the behavioural goal of which item to select. In fact, in the *dislike* frame participants need to consider the 'anti-value' of the item to choose the one to reject.

To anticipate our results, in the *like* frame condition we replicated the typical gaze-boosting effect: participants looked for longer at the item they were about to choose – the item they deemed most valuable. In the *dislike* frame, however, participants looked for longer at the item that they then chose to eliminate, that is, the *least* valuable item. This means that agents paid more attention to the option they selected in the task, *not* to the option to which they deemed more valuable or wanted to consume. This suggests that attention does *not* boost value but rather is used to gather task-relevant information.

In order to understand the mechanism via which attention interacts with value in both framings, we use a dynamic accumulation model, which allows us to account for the preference formation process and its dependency on task variables (values of the options). We also show how goal-relevance shapes confidence and how confidence interacts with attention.

To test the generality of our findings, we also conducted a new perceptual decision-making experiment and tested a new set of participants. In this perceptual task, participants were asked to choose between two circles filled with dots. In some blocks, they had to indicate the circle with more dots – *most frame*; in others, the circle with fewer dots – *fewest frame*. In this second study, we replicated all the effects of the first, value-based one, corroborating the hypothesis of a domain-general role for attention in modulating goal-relevant information that drives choice.

This work questions the dominant view in neuroeconomics about the relationship between attention and value, showing that attention does not boost value per se but instead modulates goal-relevant information. We conclude our work by presenting an economic model of optimal evidence accumulation. Using this model, we suggest that the behavioural strategy we observe in our experiment may be the result of deploying, in the context of binary choice, a behavioural strategy that is optimal when agents face more natural larger sets of options.

## Results

In our first experiment, hungry participants (n = 31) made binary choices between snacks in one of two task-frames, *like* and *dislike*. In the *like* frame, participants had to report the item they would prefer to eat; in the *dislike* frame, they chose the item they wanted to avoid eating (*Figure 1A*). After each choice, participants reported their confidence in having made a good choice (*De Martino et al., 2013*; *Folke et al., 2017*). At the beginning of the experiment, participants

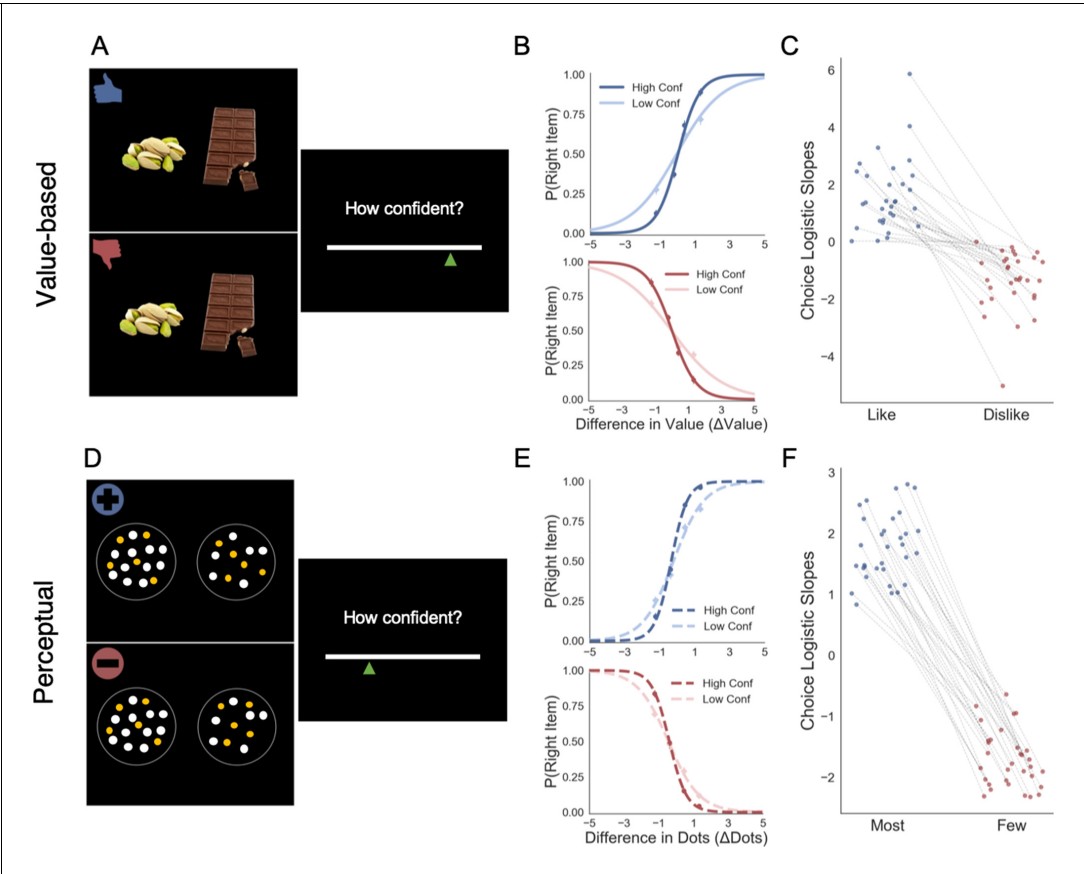

**Figure 1.** Task and behavioural results. Value-based decision task (**A**). Participants choose between two food items presented in an eye-contingent way. Before the choice stage, participants reported the amount of money they were willing to bid to eat that snack. In the *like* frame (top) participants select the item they want to consume at the end of the experiment. In the *dislike frame* (bottom) participants choose the opposite, the item they would prefer to avoid. After each choice participants reported their level of confidence. (**B**) After a median split for choice confidence, a logistic regression was calculated for the probability of choosing the right-hand item depending on the difference in value ($Value_{Right}$– $Value_{Left}$) for *like* (top) and *dislike* (bottom) framing conditions. The logistic curve calculated from the high confidence trials is steeper, indicating an increase in accuracy. (**C**) Slope of logistic regressions predicting choice for each participant, depending on the frame. The shift in sign of the slope indicates that participants are correctly modifying their choices depending on the frame. Perceptual decision task (**D**) Participants have to choose between two circles containing dots, also presented eye-contingently. In the *most* frame (top), participants select the circle with more white dots. In the *fewest* frame (bottom), they choose the circle with the lower number of white dots. Distractor dots (orange) are included in both frames to increase the difficulty of the task. Confidence is reported at the end of each choice. We obtained a similar pattern of results to the one observed in the Value Experiments in terms of probability of choice (**E**) and the flip in the slope of the choice logistic model between *most* and *fewest* frames (**F**).

reported the subjective value of individual items using a standard incentive-compatible Becker-DeGroot-Marschak mechanism (BDM; see Materials and methods).

Our second experiment was done to test whether the results observed in value-based decisions could be generalised to perceptual decisions. A different group of participants (n = 32) made binary choices between two circles containing a variable number of dots (*Figure 1D*). In the *most* frame, participants reported the circle containing the higher number of dots; in the *fewest* frame, the one with the lower. As in the Value Experiment, at the end of each trial participants reported their confidence in their choice.

## The effect of attention on choice

### Value experiment

Our results confirmed that participants understood the task and chose higher value items in the *like* frame and lower value items in the *dislike* frame (*Figure 1B,C*). This effect was modulated by confidence (*Figure 1B*) similarly to previous studies (*De Martino et al., 2013*; *Folke et al., 2017*;

*Boldt et al., 2019*). For a direct comparison of the differences between the goal manipulations in the two tasks (Value and Perceptual) see Appendix 1 (*Appendix 1—figure 1*).

We then tested how attention interacts with choice by examining the eye-tracking variables. Our frame manipulation, which orthogonalised choice and valuation, allowed us to distinguish between two competing hypotheses. The first hypothesis, currently dominant in the field, is that visual attention is always attracted to high values items and that it facilitates their choice. The alternative hypothesis is that the attention is attracted to items whose value matches the goal of the task. These two hypotheses make starkly different experimental predictions in our task. According to the first, gaze will mostly be allocated to the more valuable item independently of the frame. The second hypothesis instead predicts that in the *like* frame participants will look more at the more valuable item, while this pattern would reverse in the *dislike* frame, with attention mostly allocated to the least valuable item. In other words, according to this second hypothesis, visual attention should predict choice (and the match between value and goal) and not value, independently of the frame manipulation.

Our data strongly supported the second hypothesis because we found participants preferentially gaze (*Figure 2A*) the higher value option during *like* (t(30) = 7.56, p<0.001) and the lower value option during *dislike* frame (t(30) = -4.99, p<0.001). From a hierarchical logistic regression analysis predicting choice (*Figure 2B*), the difference between the time participants spent observing the right over left item (ΔDT) was a positive predictor of choice both in *like* ($z$ = 6.448, p<0.001) and *dislike* ($z$ = 6.750, p<0.001) frames. This means that participants looked for longer at the item that better fits the frame and not at the item with the highest value. Notably, the magnitude of this effect was slightly lower in the *dislike* case (t(30) = 2.31, p<0.05). In *Figure 2B* are also plotted the predictors of the other variables on choice from the best fitting model.

## Perceptual experiment

We then analysed the effect of attention on choice in the perceptual case to test the generality of our findings. As in the Value Experiment, our data confirmed that participants did not have issues in choosing the circle with more dots in the *most* frame and the one with least amount dots in the *fewest* frame (*Figure 1D,F*). Furthermore, as in the Value Experiment and many other previous findings (*De Martino et al., 2013*; *Folke et al., 2017*), confidence modulated the accuracy of their decisions (*Figure 1E*). Critically for our main hypothesis, we found that participants' gaze was preferentially allocated to the relevant option in each frame (*Figure 2C*): they spent more time observing the circle with more dots during *most* frame (t(31)=13.85, p<0.001) and the one with less dots during *fewest* frame (t(31)=-10.88, p<0.001). ΔDT was a positive predictor of choice (*Figure 2D*) in *most* ($z$ = 10.249, p<0.001), and *fewest* ($z$ = 10.449, p<0.001) frames. Contrary to the results in the Value Experiment in which the effect of ΔDT on choice was slightly more marked in the *like* condition (*Figure 2B*), in the Perceptual Study the effect of ΔDT was the opposite: ΔDT had a higher effect in the *fewest* frame ($\Delta DT_{Most-Few}$: t(31)=-2.17, p<0.05)(*Figure 2D*). However, and most importantly, in both studies ΔDT was a robust positive predictor of choice in both frame manipulations. To summarise, these results show that in the context of a simple perceptual task, visual attention also has a specific effect in modulating information processing in a goal-directed manner: subjects spend more time fixating the option they will select, not necessarily the option with the highest number of dots.

In both Value and Perceptual Experiments, the most parsimonious models were reported in the manuscript and in *Figure 2B and D*. For a full model comparison see *Appendix 2—figure 1* and *Appendix 2—table 1*. More details on the choice models are reported in the Appendix 2.

## Fixations effects in choice

An important prediction of attentional accumulation models is that the chosen item is generally fixated last (unless that item is much worse than the other alternative), with the magnitude of this effect related to the difference in value between the alternatives. This feature of the decision has been consistently replicated in various previous studies (*Krajbich et al., 2010*; *Krajbich and Rangel, 2011*; *Krajbich et al., 2012*). We therefore tested how the last fixation was modulated by the frame manipulation.

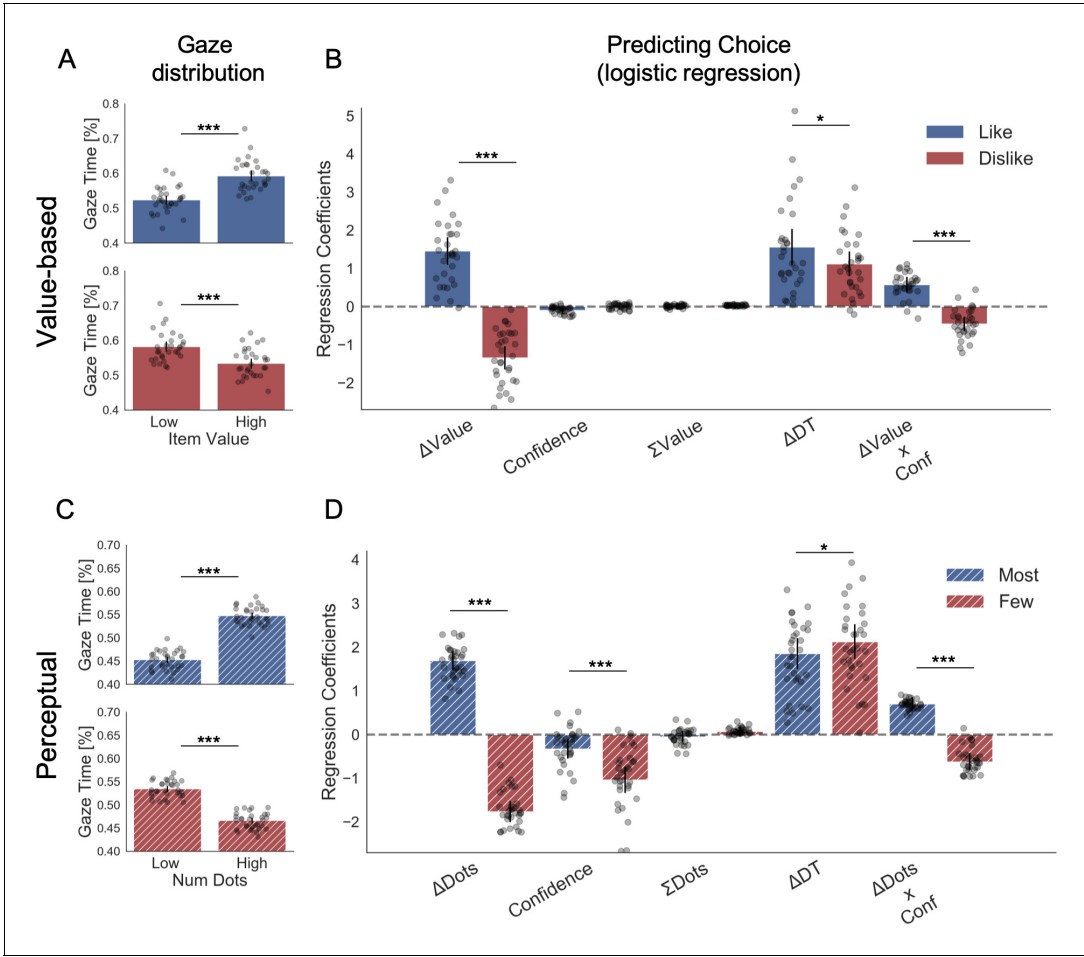

**Figure 2.** Attention and choice in Value and Perceptual Experiments. (**A**) Gaze allocation time depends on the frame: while visual fixations in the *like* frame go preferentially to the item with higher value (top), during the *dislike* frame participants look for longer at the item with lower value (bottom). Dots in the bar plot indicate participants' average gaze time across trials for high and low value items. Time is expressed as the percentage of trial time spent looking at the item. Similar results were found for gaze distribution in the Perceptual Experiment (**C**) participants gaze the circle with higher number of dots in *most* frame and the circle with lower number of dots in *fewest* frame. Hierarchical logistic modelling of choice (probability of choosing right item) in Value (**B**) and Perceptual (**D**) Experiments, shows that participants looked for longer ($\Delta$DT) at the item they chose in both frames. All predictors are z-scored at the participant level. In both regression plots, bars depict the fixed-effects and dots the mixed-effects of the regression. Error bars show the 95% confidence interval for the fixed effect. In Value Experiment: $\Delta$Value: difference in value between the two items ($Value_{Right} - Value_{Left}$); RT: reaction time; $\Sigma$Value: summed value of both items; $\Delta$DT: difference in dwell time ($DT_{Right} - DT_{Left}$); Conf: confidence. In Perceptual Experiment: $\Delta$Dots: difference in dots between the two circles ($Dots_{Right} - Dots_{Left}$); $\Sigma$Dots: summed number of dots between both circles. \*\*\*p<0.001, \*\*p<0.01, \*p<0.05.

### Value experiment

In the Value Experiment in both frames, we replicated the last fixation effect and its modulation by value difference between the last fixated option and the other one (***Figure 3A***). In the *like* frame, the probability of choosing the last item fixated upon increases when the value of the last item is higher, as is shown by the positive sign of the slope of the logistic curve (mean $\beta_{Like}$ = 0.922). Crucially, during the *dislike* frame the opposite effect was found: the probability of choosing the last seen option increases when the value of the non-chosen item is higher, seen from the negative slope of the curve (mean $\beta_{Dislike}$ = −0.951; $\Delta\beta_{Like-Dislike}$: t(30)=7.963, p<0.001).

### Perceptual experiment

We observed the same pattern of results that in the Value Experiment (***Figure 3B***). In the *most* frame, it was more probable that the last fixation was on the chosen item when the fixated circle had a higher number of dots (mean $\beta_{Most}$ = 1.581). In the *fewest* frame, the effect flipped: it was

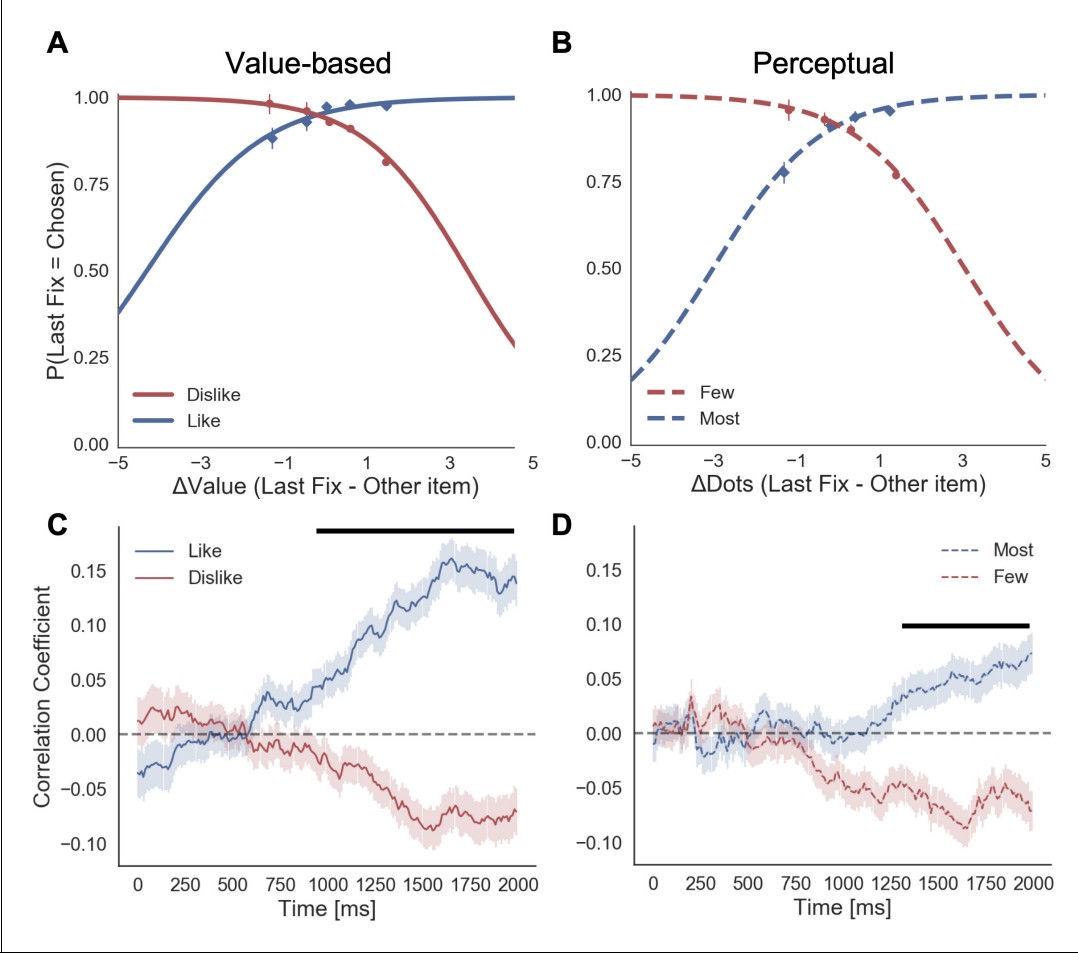

**Figure 3.** Fixation effects on the chosen item. Last fixation effects: (**A**) in the Value Experiment, a logistic regression was calculated for the probability the last fixation is on the chosen items (P(LastFix = Chosen)) depending on the difference in value of the item last fixated upon and the alternative item. As reported in previous studies, in *like* frame, we find it is more probable that the item last fixated upon will be chosen when the value of that item is relatively higher. In line with the hypothesis that goal-relevant evidence, and not value, is being integrated to make the decision, during the *dislike* frame the effect shows the opposite pattern: P(LastFix = Chosen) is higher when the value of the item last fixated on is lower, that is, the item fixated on is more relevant given the frame. (**B**) A similar analysis in the Perceptual Experiment mirrors the results in the Value Experiment with a flip in the effect between *most* and *fewest* frames. Lines represent the model predictions and dots are the data binned across all participants. ΔValue and ΔDots measures are z-scored at the participant level. Gaze preference in time: (**C**) Pearson correlation between gaze position and difference in value (ΔValue) was calculated for each time point during the first 2 s of the trials. In the Value Experiment, after an initial phase of random exploration, fixations are positively correlated with the high value item in *like* frame, while this effect is the opposite for *dislike* frame, that is, fixations are directed to the low value item. (**D**) In the Perceptual Experiment, a similar pattern of goal-relevant fixations emerges. Lines in both figures correspond to the time point correlation considering all trials and participants. Shaded area corresponds to the standard error. Black line indicates time points with statistically significant difference between frames, resulting from a permutation test (p-value<0.01 for at least 6 time bins, 60 ms). Correction for multiple comparison was performed using FDR, α ≤ 0.01.

more likely that the last circle seen was chosen when it had fewer dots (mean $\beta_{Few}$ = −0.944; $\Delta\beta_{Most-Few}$: t(31)=3.727, p<0.001).

The previous set of analysis shows that the last fixation is modulated by the difference in evidence according to the goal that the participant is set to achieve. However, since the last fixation is in general followed by the participant response, one could suspect that the goal-dependent modulation of attention (i.e. ΔDT) we identified in our choice regression analysis (*Figure 2*) is entirely driven by the final fixation. This would be problematic since one would have similar results to the one presented in *Figure 2* even if participants' pattern of attention is not modulated by the goal (i.e. attention is

directed in both frames to the most valuable item) or even if the pattern of fixation, before the last fixation, is random. To control for this possibility, we performed a series of further analyses:

First of all, we repeated the analysis presented in the previous section (hierarchical choice regression – *Figure 2*), removing the last two fixations when calculating the ΔDT. Note that we removed the last two fixations and not just the last one to avoid statistical artefacts (i.e. since the final fixation is mostly directed towards the chosen item there would be an increased probability that second to last fixation is on the unchosen item). In *Appendix 2—figure 3*, we show that once removed the last two fixations the pattern of results is unchanged.

Second, we specifically investigated the middle fixations. Previous studies (*Krajbich et al., 2010*; *Krajbich and Rangel, 2011*; *Tavares et al., 2017*) have reported that middle fixations duration increases when the difference in value ratings (or perceptual evidence) of the fixated minus unfixated item increases. We replicated this result for our *like* and *most* frames but critically the effect was reversed in *dislike* and *fewest* frames (i.e. middle fixations durations decreased when the relative value of the fixated item was higher). The results suggesting that the goal-relevant modulation of attention affects also the middle fixations are presented in the *Appendix 3—figure 4*.

Finally, we investigated in more detail how the relation between attentional allocation and difference in value or perceptual evidence changed over time in the context of the goal manipulation. We calculated the Pearson correlation between fixation position (0: left, 1:right) and the difference in evidence (i.e. ΔValue or ΔDots, in both cases right – left item) at different time points (*Figure 3C*). We observed that after an initial phase in which there was no clear gaze preference for any of the items (note that given the gaze-contingent design participants must explore both alternatives), fixations were correlated with the frame-relevant item: during *like* frame, fixations positions were positively correlated with ΔValue, that is the fixations were directed towards the item with higher value; during *dislike* frame the behaviour was the opposite: fixations were negatively correlated with ΔValue, indicating a preference for the option with lower value. Note that these results are in line with the ones reported by *Kovach et al., 2014*. We see a very similar pattern of results in the Perceptual Experiment too (*Figure 3D*).

## Which factors determine confidence?

### Value experiment

To explore the effect that behavioural factors had over confidence, we fitted a hierarchical linear model (*Figure 4A*). As it was the case for the results presented above for the choice regression, the results for the confidence regression in the *like* frame replicated all the effects reported in a previous study from our lab (*Folke et al., 2017*). Again, we presented here the most parsimonious model (*Appendix 4—figure 1* and *Appendix 4—table 1* for model comparison). We found that the magnitude of ΔValue (|ΔValue|) had a positive influence on confidence in *like* ($z = 5.465$, $p<0.001$) and *dislike* ($z = 6.300$, $p<0.001$) frames, indicating that participants reported higher confidence when the items have a larger difference in value; this effect was larger in the *dislike* frame ($t(30) = -4.72$, $p<0.01$). Reaction time (RT) had a negative effect on confidence in *like* ($z = -6.373$, $p<0.001$) and *dislike* ($z = -7.739$, $p<0.001$) frames, that is, confidence was lower when the RTs were longer. Additionally, we found that, in both conditions, higher number of gaze switches (i.e. gaze shift frequency, GSF) predicted lower values of confidence in *like* ($z = -2.365$, $p<0.05$) and *dislike* ($z = -2.589$, $p<0.05$) frames, as reported in *Folke et al., 2017*.

We then looked at the effect of the summed value of both options, ΣValue, on confidence. As in *Folke et al., 2017*, we found a positive effect of ΣValue on confidence in the *like* frame ($z = 3.206$, $p<0.01$); that is, participants reported a higher confidence level when both options were high in value. Interestingly, this effect was inverted in the *dislike* frame ($z = -4.492$, $p<0.001$), with a significant difference between the two frames ($t(30)=9.91$, $p<0.001$) This means that, contrary to what happened in the *like* frame in which confidence was boosted when both items had high value, in the *dislike* frame confidence increased when both items had *low* value. This novel finding reveals that the change in context also generates a reassessment of the evidence used to generate the confidence reports; that is, confidence also tracks goal-relevant information.

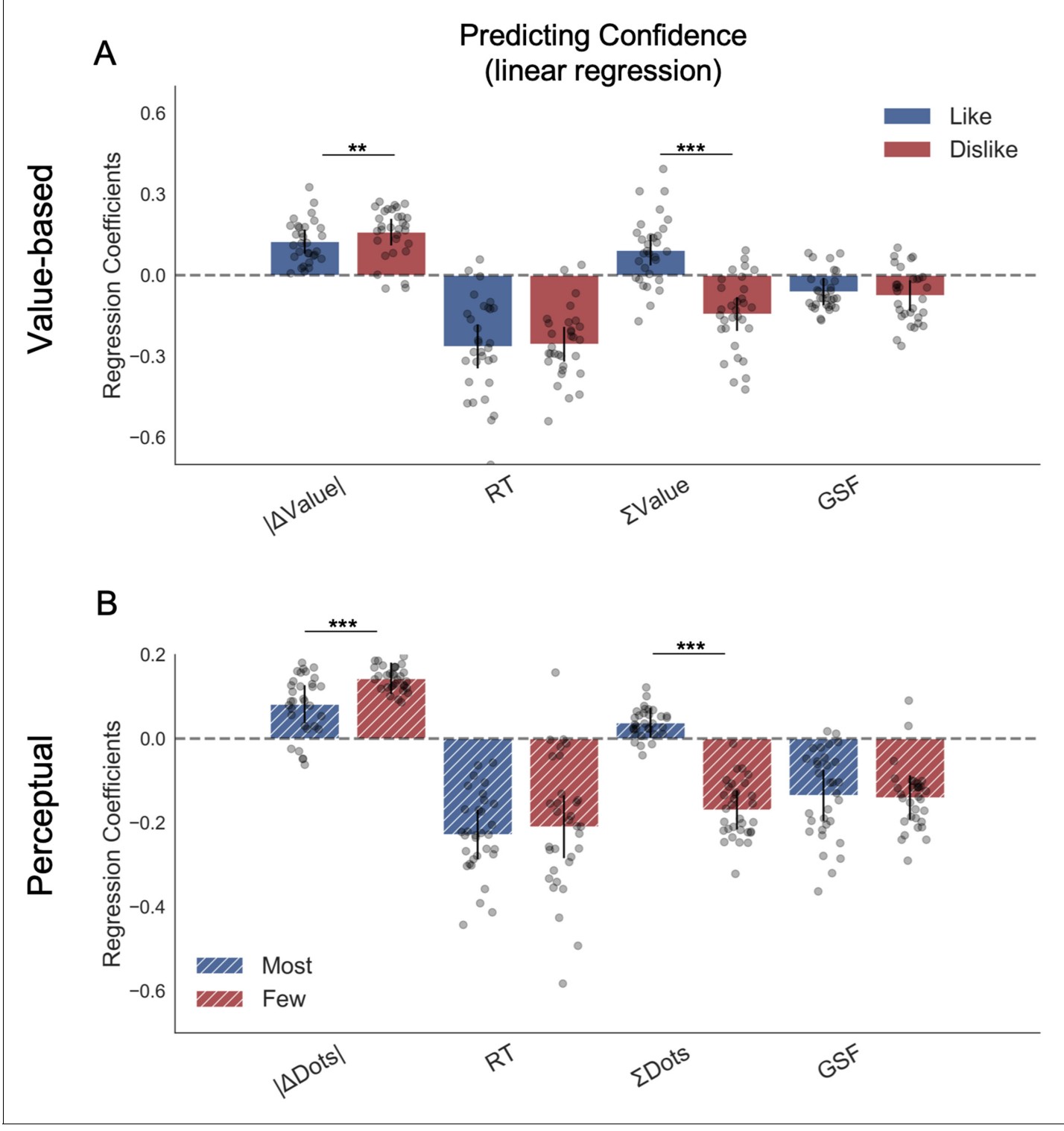

**Figure 4.** Hierarchical linear regression model to predict confidence. (**A**) In Value Experiment, a flip in the effect of ΣValue over confidence in the *dislike* frame was found. (**B**) In Perceptual Experiment, a similar pattern was found in the effect of ΣDots over confidence in the *fewest* frame. The effect of the other predictors on confidence in both experiments and frames coincides with previous reports (*Folke et al., 2017*). All predictors are z-scored at the participant level. In both regression plots, bars depict the fixed-effects and dots the mixed-effects of the regression. Error bars show the 95% confidence interval for the fixed effect. In Value Experiment: ΔValue: difference in value between the two items ($Value_{right}$ – $Value_{left}$); RT: reaction time; ΣValue: summed value of both items; ΔDT: difference in dwell time ($DT_{right}$ – $DT_{left}$); GSF: gaze shift frequency; ΔDT: difference in dwell time. In
*Figure 4 continued on next page*

Figure 4 continued

Perceptual Experiment: ΔDots: difference in dots between the two circles ($Dots_{right} - Dots_{left}$); ΣDots: summed number of dots between both circles. ***p<0.001, **p<0.01, *p<0.05.

## Perceptual experiment

We repeated the same regression analysis in the perceptual decision experiment, replacing value evidence input with perceptual evidence (i.e. absolute difference in the number of dots, |ΔDots|). We directly replicated all the results of the Value Experiment, generalising the effects we isolated to the perceptual realm (*Figure 4B*). Specifically, we found that |ΔDots| had a positive influence on confidence in *most* (z = 3.546, p<0.001) and *fewest* frames (z = 7.571, p<0.001), indicating that participants reported higher confidence when the evidence was stronger. The effect of absolute evidence |ΔDots| on confidence was bigger in the *fewest* frame (t(31)=-4.716, p<0.001). RT had a negative effect over confidence in *most* (z = −7.599, p<0.001) and *fewest* frames (z = −5.51, p<0.001), that is, faster trials were associated with higher confidence. We also found that GSF predicted lower values of confidence in *most* (z = −4.354, p<0.001) and *fewest* (z = −5.204, p<0.001) frames. Critically (like in the Value Experiment), the effect of the sum of evidence (ΣDots) on confidence also changes sign depending on the frame. While ΣDots had a positive effect over confidence in the *most* frame (z = 2.061, p<0.05), this effect is the opposite in the *fewest* frame (z = −7.135, p<0.001), with a significant difference between the parameters in both frames (t(31)=14.621, p<0.001). The magnitude of ΣDots effect was stronger in the *fewest* frame (t(31)=-10.438, p<0.001). For further details on the confidence models see the Appendix 4 (Appendix 4—table 2 and Appendix 4—table 3).

## Attentional model: GLAM

To gain further insights into the dynamic of the information accumulation process, we modelled the data from both experiments adapting a Gaze-weighted Linear Accumulator Model (GLAM) recently developed by *Thomas et al., 2019*. The GLAM belongs to the family of race models and approximates the aDDM model (*Krajbich et al., 2010*; *Krajbich and Rangel, 2011*) in which the dynamic aspect is discarded, favouring a more efficient estimation of the parameters. This model was chosen since, unlike the aDDM, it allowed us to test the prediction of the confidence measures as balance of evidence (*Vickers, 1979*; *Kepecs et al., 2008*; *De Martino et al., 2013*). Crucially, in both experiments, we used goal-relevant evidence (not the value or the number of dots) to fit the models in the *dislike* and *fewest* frames (for further details see the Materials and methods *Attentional Model: Glam* section).

### Parameter fit and simulation

#### Value experiment

The simulations estimated with the parameters fitted for *like* and *dislike* frames data (even-trials) reproduced the behaviour observed in the data not used to fit the model (odd-trials). In both *like* and *dislike* frames, the model replicated the observed decrease of RT when |ΔValue| is high, that is, the increase in speed of response in easier trials (bigger value difference). The RT simulated by the models significantly correlated with the RT values observed in participants odd-numbered trials (*Like*: r(29)=0.90, p<0.001; *Dislike*: r(29)=0.89, p<0.001) (*Figure 5A*). In the *like* frame, the model also correctly predicted a higher probability of choosing the right item when ΔValue is higher. In the *dislike* frame, the model captured the change in the task goal and predicted that the selection of the right item will occur when -ΔValue is higher, that is when the value of the left item is higher. Overall, in both frames the observed and predicted probabilities of choosing the most valuable item were significantly correlated (*Like*: r(29)=0.80, p<0.001; *Dislike*: r(29)=0.79, p<0.001) (*Figure 5B*). See *Appendix 5—figure 4A* and *Appendix 5—figure 5A* for further details.

In both frames, the models also predicted choice depending on the difference in gaze (ΔGaze = $g_{right} - g_{left}$), that is, that the probability of choosing the right item increases when the time spent observing that item is higher. However, in this case, we cannot say if gaze allocation itself is predicting choice if we do not account for the effect of |ΔValue|. To account for the relationship between choice and gaze, we used a measure devised by *Thomas et al., 2019*, 'gaze influence'.

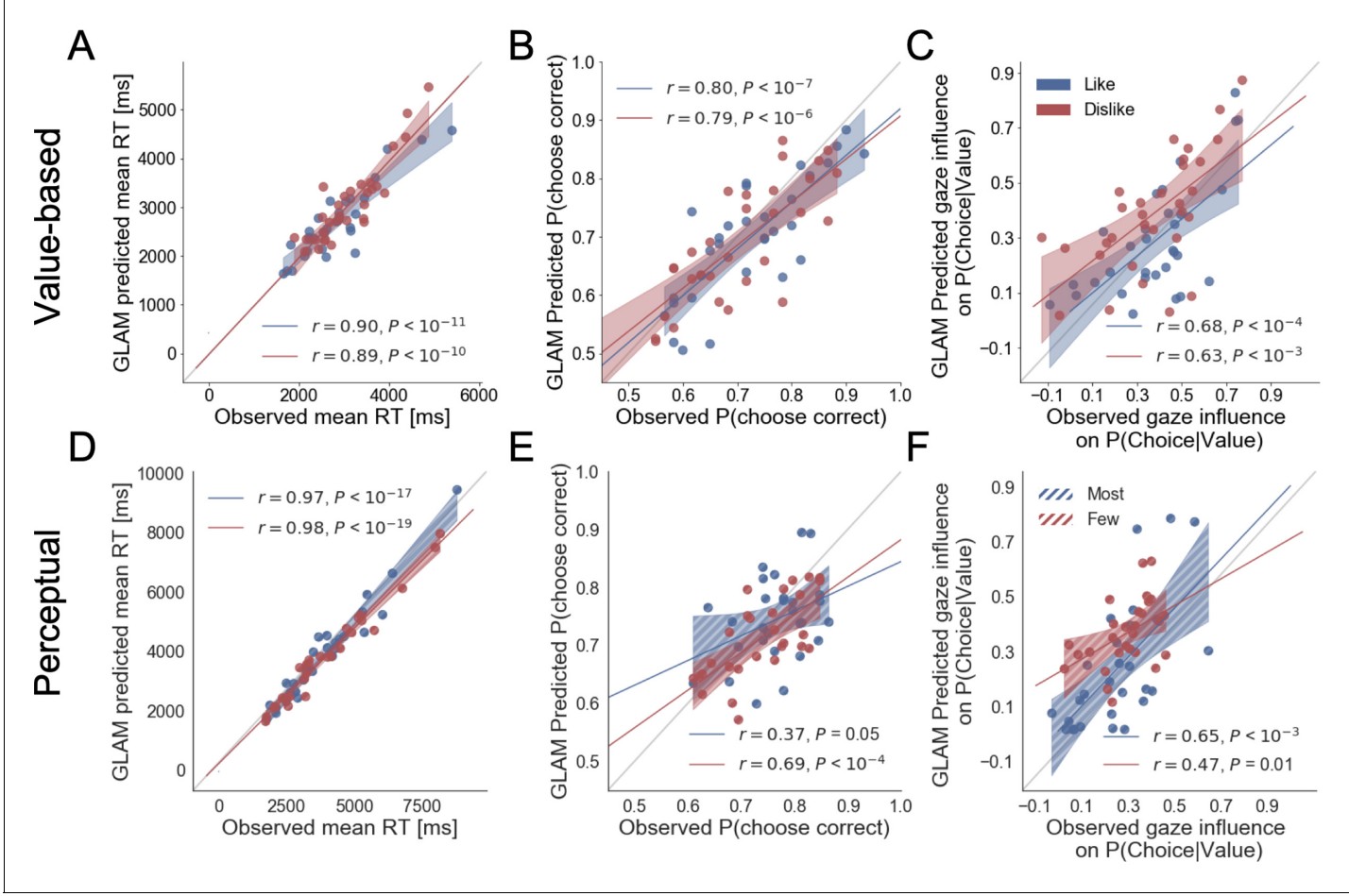

**Figure 5.** Individual out-of-sample GLAM predictions for behavioural measures in Value (A-C) and Perceptual Experiments (D–F). In value-based decision, (A) the model predicts individuals mean RT; (B) the probability of choosing the item with higher value in *like* frame, and the item with lower value in *dislike* frame; and (C) the influence of gaze in choice probability. In the Perceptual Experiment, (D) the model also predicts RT and (F) gaze influence. (E) The model significantly predicts the probability of choosing the best alternative in the *fewest* frame only (in the *most* frame a trend was found). The results corresponding to the models fitted with *like/most* frame data are presented in blue, and with *dislike/fewest* frame data in red. Dots depict the average of observed and predicted measures for each participant. Lines depict the slope of the correlation between observations and the predictions. Mean 95% confidence intervals are represented by the shadowed region in blue or red, with full colour representing Value Experiment and striped colour Perceptual Experiment. All model predictions are simulated using parameters estimated from individual fits for even-numbered trials.

Gaze influence is calculated taking the actual choice (1 or 0 for right or left choice, respectively) and subtracting the probability of choosing the right item given by a logistic regression for ΔValue calculated from actual behaviour. The averaged 'residual' choice probability indicates the existence of a positive or negative gaze advantage. Then, we compared the gaze influence predicted by GLAM with the empirical one observed for each participant. As in *Thomas et al., 2019*, most of the participants had a positive gaze influence and it was properly predicted by the model in both frames (Like: r(29)=0.68, p<0.001; Dislike: r(29)=0.63, p<0.001) (*Figure 5C*).

## Perceptual experiment

As in the Value Experiment, we fitted the GLAM to the data and we conducted model simulations. Again, these simulations showed that we could recover most of the behavioural patterns observed in participants. We replicated the relationship between RT and |ΔDots| (*Most*: r(26)=0.97, p<0.001; *Fewest*: r(26)=0.98, p<0.001) (*Figure 5D*). As in the value-based experiment, the model also predicted a higher probability of choosing the right-hand item when ΔDots is higher in the *most* frame and when -ΔDots is higher in the *fewest* frame. However, in the Perceptual Experiment, the simulated choices only in the *fewest* frame were significantly correlated with the observed data, although

we observed a non-significant trend in the *most* frame (*Most*: r(26)=0.69, p<0.001; *Fewest*: r(26) =0.37, p=0.051) (*Figure 5E*). In both frames, we observed that the model predicted that choice was linked to ∆Gaze and, as in the Value Experiment, we show that the gaze influence predicted by the model is indeed observed in the data (*Most*: r(26)=0.65, p<0.001; *Fewest*: r(26)=0.47, p<0.05) (*Figure 5F*). See *Appendix 5—figure 4B* and *Appendix 5—figure 5B* for further details.

Results of the models fitted without accounting for the change in goal-relevant evidence provided a poor fit of the data, these results are presented in *Appendix 5—figures 1–3* and *6*. For a direct comparison of the different GLAM parameters see Appendix 6. Additionally, we were able to mirror the results obtained with GLAM using aDDM (*Krajbich et al., 2010*; *Tavares et al., 2017*). For *dislike* and *fewest* frames, the best model was the one fitted using goal-relevant evidence (see Appendix 7 for details).

## Balance of evidence and confidence

The GLAM belongs to the family of race models in which evidence is independently accumulated for each option. Therefore, using the GLAM we were able to adapt the model to estimate a measure of confidence in the decision that is defined by the balance of evidence (*Vickers, 1979*; *Vickers, 1970*; *Kepecs et al., 2008*; *De Martino et al., 2013*) allowing us to characterise the pattern of the

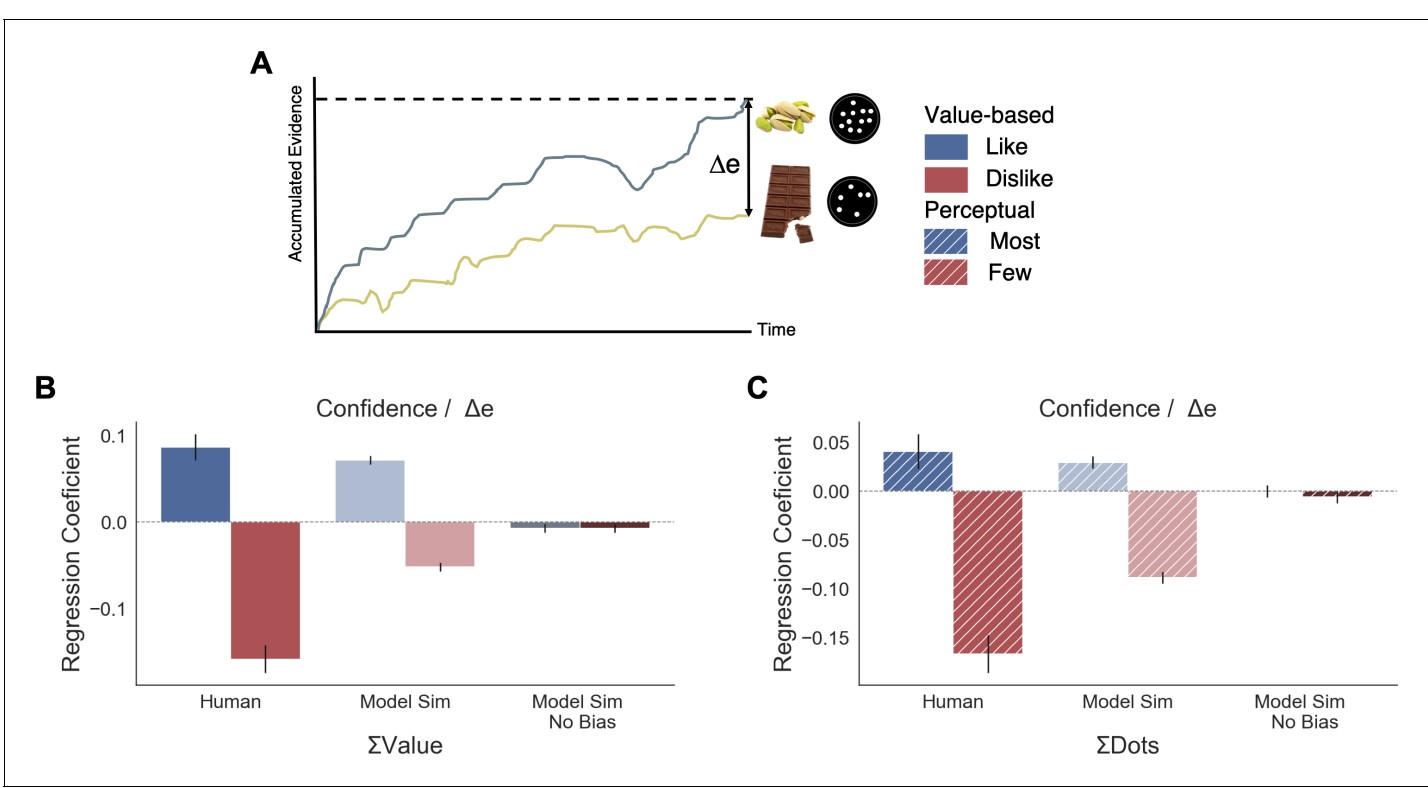

**Figure 6.** Balance of evidence (∆e) simulated with GLAM reproduces ΣValue and ΣDots effects over confidence. (**A**) GLAM is a linear stochastic race model in which two alternatives accumulate evidence until a threshold is reached by one of them. ∆e has been proposed as a proxy for confidence and it captures the difference in evidence available in both accumulators once the choice for that trial has been made. (**B**) Using ∆e simulations, we captured the flip of the effect of ΣValue over confidence between *like* and *dislike* frames. ∆e simulations were calculated using the model with parameters fitted for each individual participant. A pooled linear regression model was estimated to predict ∆e. The effects of ΣValue predicting ∆e are presented labelled as 'Model Sim'. A second set of simulations was generated using a model in which no asymmetries in gaze allocation were considered (i.e. no attentional biases). This second model was not capable of recovering ΣValue effect on ∆e and is labelled as 'Model Sim No Bias'. ΣValue coefficients for a similar model using participants' data predicting confidence are also presented labelled as 'Human' for comparison. (**C**) A similar pattern of results is found in the Perceptual Experiment, with the model including gaze bias being capable of recovering ΣDots effect on ∆e. This novel effect may suggests that goal-relevant information is also influencing the generation of second-order processes, as confidence. This effect may be originated by the attentional modulation of the accumulation dynamics. Coloured bars show the parameter values for ΣValue and ΣDots and the error bars depict the standard error. Solid colour indicates the Value Experiment and striped colours indicate the Perceptual Experiment. All predictors are z-scored at participants level.

confidence measures. Balance of evidence is defined as the absolute difference between the accumulators for each option at the moment of choice, which is when one of them reaches the decision threshold (i.e., $\Delta e = |E_{right}(t_{final}) - E_{left}(t_{final})|$) (**Figure 6A**). To estimate $\Delta e$, we performed a large number of computer simulations using the fitted parameters for each participant in both experiments.

## Value experiment

To confirm that the relationship between confidence and other experimental variables was captured by the balance of evidence simulations, we constructed a linear regression model predicting $\Delta e$ as function of the values and the RTs obtained in the simulations ($\Delta e \sim |\Delta Value| +$ simulated RT + $\Sigma Value$). We found that this model replicated the pattern of results we obtained experimentally (**Figure 4**). We then explored whether the model was able to recover the effect of $\Sigma Value$ on confidence (**Figure 6B**). As we have shown when analysing confidence, $\Sigma Value$ boosted $\Delta e$ in the *like* frame ($\beta_{\Sigma Value}$ = 0.071, t(37196) = 14.21, p<0.001) and reduced $\Delta e$ in the *dislike* frame ($\beta_{\Sigma Value}$ = −0.061, t(37196) = −12.07, p<0.001). The effect of $\Sigma Value$ over confidence was replicated in the simulations with an increase of $\Delta e$ when high value options are available to choose (**Appendix 8—figure 1** and **Appendix 8—figure 3A,D** for more details). In the *dislike* frame, the fitted model also replicated this pattern of behaviour, including the adaptation to context which predicts higher $\Delta e$ when both alternatives have low value. Interestingly, the replication of the effect for $\Sigma Value$ over $\Delta e$ with GLAM did not hold when the gaze bias was taken out of the model in *like* ($\beta_{\Sigma Value}$ = −0.007, t(37196) = −1.495, p=0.13, ns) and *dislike* ($\beta_{\Sigma Value}$ = −0.002, t(37196) = −0.413, p=0.679, ns) frames (**Figure 6B**). We also found that the effect of $|\Delta Value|$ on confidence was replicated by the simulated balance of evidence, increasing $\Delta e$ when the difference between item values is higher (i.e. participants and the model simulations are more 'confident' when $|\Delta Value|$ is higher) (**Appendix 8—figure 1**).

## Perceptual experiment

We conducted a set of similar analyses and model simulations in the Value Experiment (**Figure 6C**). We found that $\Sigma Dots$ boosted $\Delta e$ in the *most* frame (*Most* : $\beta_{\Sigma Dots}$ = 0.029, t(33596) = 4.71, p<0.001) and reduced $\Delta e$ in the *fewest* frame (*Fewest* : $\beta_{\Sigma Dots}$ = −0.088, t(33596) = −14.41, p<0.001) . As in the Value Experiment, this effect disappeared when the gaze bias was taken out of the model (*Most*: $\beta_{\Sigma Dots}$ = −0.0002, t(33596) = −0.04, p=0.96, ns; *Fewest*: $\beta_{\Sigma Dots}$ = −0.006, t(33596) = −1.03, p=0.29, ns) (see **Appendix 8—figure 2** and **Appendix 8—figure 3B,E** for more details).

Overall, these results show how the model is capable of capturing the novel empirical effect on confidence we identified experimentally, giving computational support to the hypothesis that goal-relevant evidence is fed to second order processes like confidence. It also hints at a potential origin to the effects of the sum of evidence (i.e. $\Sigma Value$, $\Sigma Dots$) on confidence: asymmetries in the accumulation process, in particular the multiplicative effect of attention over accumulation of evidence, may enhance the differences between items that are more relevant for the frame. This consequentially boosts the level of confidence that participants have in their decisions.

## A model of optimal information acquisition

We then sought to understand why participants systematically accumulated evidence depending on the task at hand, instead of first integrating evidence using a task-independent strategy and then emitting a response appropriate with the task. We reasoned that this may reflect a response in line with models of rational information acquisition popular in economics. These include models of so-called rational inattention, according to which agents are rationally choosing which information to acquire considering the task, the incentives, and the cost of acquiring and processing information (**Sims, 2003**; **Sims, 2010**; **Caplin and Dean, 2015**; **Hébert and Woodford, 2017**). As opposed to DDM or GLAM, these models attempt to investigate not only what the consequences of information acquisition are, but also *which* information is acquired.

In this model, we consider an agent facing $n$ available options. Each item $i$ has value $v_i$ to the agent, which is unknown, and agents have a prior such that values follow an independent, identical distribution; for simplicity, we assume it to be a Normal $v_i \sim N(\mu, \sigma_\mu^2)$. Agents can acquire information in the form of signals $x_i = v_i + \epsilon_i$, with $\epsilon_i$ independently and identically distributed with $\epsilon_i \sim N(0, \sigma_\epsilon^2)$.

They follow Bayes' rule in updating their beliefs after information. Once they finish acquiring information, they then choose the item with the highest expected value.

Consider first the case in which an agent needs to pick the best item out of $n$ possible ones. Suppose that she already received one signal for each item. Denote $i_1$ the item for which the agent received the highest signal, which is also the item with the highest expected value; $i_2$ the second highest, etc. (Because each of these is almost surely unique, let us for simplicity assume they are indeed unique). The agent can acquire one additional signal about any of the available items or select any probability distribution over signals. The following proposition shows that it is (weakly) optimal for the agent to acquire a second signal about the item that is currently best, that is, $i_1$.

Denote $\Delta$ the set of all probability distributions over signals and $V(i)$ the utility after acquiring a new signal $x_{i,2}$ about item $i$, that is,

$$V(i) := \max_{j \in 1,\dots,n} \mathbb{E}[v_j | x_1, \dots, x_N, x_{i,2}] \tag{1}$$

**Proposition 1.** *The optimal strategy when choosing the best option is to acquire one more signal about item $i_1$ or $i_2$, that is, either the item with the currently highest expected value or the one with second highest value. That is:*

$$\mathbb{E}[V(i_1)] = \mathbb{E}[V(i_2)] \geq \max_{p \in \Delta} \sum_{i=1}^{n} p(i) V(i)$$
$$\text{and } \mathbb{E}[V(i_1)] > \mathbb{E}[V(i_j)] \quad \forall j \neq 1, 2 \tag{2}$$

This proposition shows that agents have *asymmetric* optimal sampling strategies: they are not indifferent between which item to sample, but rather want to acquire extra signals about items that current look best or second-best. (They are indifferent between the latter two). When $n > 2$, these strategies are strictly better than acquiring signals about any other item.

How would this change if agents need instead to pick which item to eliminate, assuming that she gets the average utility of the items she keeps? In this case, the expected utility after acquiring a new signal $x_{i,2}$ about item $i$ is:

$$\widehat{V}(i) := \max_{j \in 1,\dots,n} \mathbb{E}\left[ \frac{\sum_{i \neq j} v_j}{n-1} | x_1, \dots, x_N, x_{i,2} \right]. \tag{3}$$

Then, it is optimal to receive an additional signal about the *least* valuable item $i_n$ or the next one, $i_{n-1}$.

**Proposition 2.** *The optimal strategy when choosing which item to discard is to acquire one more signal about item $i_n$ or $i_{n-1}$, that is, either the one with the lowest or the one with the second lowest value. That is:*

$$\mathbb{E}\left[\widehat{V}(i_n)\right] = \mathbb{E}\left[\widehat{V}(i_{n-1})\right] \geq \max_{p \in \Delta} \sum_{i=1}^{n} p(i) \widehat{V}(i)$$
$$\text{and } \mathbb{E}\left[\widehat{V}(i_n)\right] > \mathbb{E}\left[\widehat{V}(i_j)\right] \quad \forall j \neq n, n-1 \tag{4}$$

For a proof of both propositions, see Appendix 9.

Again, agents have *asymmetric* optimal sampling strategies: but now, they want to sample the items that currently look *worse* again. The intuition behind both results is that when one has to choose the best item, it is more useful to acquire information that is likely to change the ranking at the top (i.e. between best or second best item) than information that changes the ranking at the bottom, since these items won't be selected (e.g. 4th and 5th item). Crucially, the reverse is true when one is tasked to select which item to eliminate.

This shows how in these simple tasks it is strictly more advantageous to acquire information in line with the current goal rather than adopting a goal-independent information-acquisition strategy.

Our model suggests that in many ecological settings, in which there are more than two options, the optimal strategy involves acquiring *asymmetric* information depending on the goal. It is only when there are only two options that individuals are indifferent about which information to acquire. We propose that the asymmetric strategies we observe even in this latter case might be a consequence of the fact that individuals have developed a strategy that is optimal for the more frequent,

real-life cases in which $n>2$, and continue to implement this same asymmetric strategy to binary choices, where it remains optimal.

## Discussion

In this study, we investigated how framing affects the way in which information is acquired and integrated during value-based and perceptual choices. Here, using psychophysics together with computational and economic models we have been able to adjudicate between two contrasting hypotheses. The first one, currently the dominant one in the field of neuroeconomics, proposes that attention modulates (either by biasing or boosting) a value integration that starts at the beginning of the deliberation process. Subsequently, at the time of the decision, the participant would give the appropriate response (in our task accepting the option with the highest value or rejecting the one with lowest one) using the value estimate constructed during this deliberation phase. The second hypothesis suggests that, from the very start of the deliberation process, the task-frame (goal) influences the type of information that is integrated. In this second scenario, attention is not automatically attracted to high value items to facilitate their accumulation but has a more general role in prioritising the type of information that is useful for achieving the current behavioural goal. Importantly, these two hypotheses make very distinct predictions about the pattern of attention and suggest very different cognitive architecture underpinning the decision process.

Our results favour the second hypothesis: specifically, we show that, in both perceptual and value-based tasks, attention is allocated depending on the behavioural goal of the task. Although our study does not directly contradict previous findings (*Krajbich et al., 2010*; *Krajbich and Rangel, 2011*; *Cavanagh et al., 2014*; *Smith and Krajbich, 2019*), it adds nuance to the view that this is a process specifically tied to value integration (defined as a hedonic or reward attribute). Our findings speak in favour of a more general role played by attention in prioritising the information needed to fulfil a behavioural goal in both value and perceptual choices (*Gottlieb, 2012*; *Kovach et al., 2014*; *Glickman et al., 2018*). Importantly, the seeking of goal-relevant information is observed along the trial, opposing the assumption that attentional sampling is random except for the last fixation (*Krajbich et al., 2010*; *Krajbich and Rangel, 2011*; see *Gluth et al., 2018*; *Gluth et al., 2020* for additional support for this idea). Pavlovian influences have been proposed to play a key role in the context of accept/reject framing manipulation (*De Martino et al., 2006*; *Guitart-Masip et al., 2012*; *Guitart-Masip et al., 2014*; *Dayan, 2012*). However, the fact that we found almost identical results in a follow-up perceptual study in which the choice was not framed in terms of 'accept' or 'reject' but using a different kind of instruction (i.e. 'choose the option with fewer or more dots') suggests that attention acts on a more fundamental mechanism of information processing that goes beyond simple Pavlovian influences.

We also measured the trial-by-trial fluctuations in confidence to gain a deeper insight in the dynamics of this process. We found that the role of confidence goes beyond that of simply tracking the probability of an action being correct, as proposed in standard signal detection theory. Instead, it is also influenced by the perceived sense of uncertainty in the evaluation process (*Navajas et al., 2017*; *Vaghi et al., 2017*), and contextual cues (*Lebreton et al., 2019*). In turn, confidence influences future responses and information seeking (*Folke et al., 2017*; *Guggenmos et al., 2016*; *Fleming et al., 2018*; *Rollwage et al., 2018*). In previous work (*Folke et al., 2017*), we reported how, in value-based choice, confidence was related not only to the difference in value between the two items, but also to the summed value (ΔValue and ΣValue using the current notation), and we found that confidence was higher if both items have a high value (*Folke et al., 2017*). Here, we replicate this effect in both experiments in the *like* and *most* conditions. However, this effect flips in the *dislike* or *fewest* frame: in these cases, confidence increases when the summed value or number of dots is *smaller*. This result is particularly striking since the frame manipulation should be irrelevant for the purpose of the decision and has little effect on the objective performance. This suggests that similarly to attention, the sense of confidence is also shaped by the behavioural goal that participants are set to achieve.

In both experiments, the incorporation of goal-relevant evidence to fit the GLAM resulted in a better model fit compared with the model in which the value or perceptual evidence was integrated independently of the frame. We then modified the GLAM to include a measure of confidence defined as balance of evidence (Δe) (*Vickers, 1979*; *Kepecs et al., 2008*; *De Martino et al., 2013*).

In doing so we confirm that our model can replicate all the main relations between confidence, choice and RT. We then tested if the model simulation was also recovering the flip in the relationship between confidence and summed evidence ($\Sigma$Value or $\Sigma$Dots) triggered by the frame manipulation. We found the model captures this effect only if the attentional bias is included in the simulations. The boost in $\Delta e$ when goal-relevant evidence in both alternatives is high can attributed to the architecture of the model: gaze has a multiplicative effect over evidence accumulation. For example, consider a case with two items of value $A_1 = 2$ and $A_2 = 1$, and a discount factor for the unattended item $u = 0.3$. Assuming the item with higher value is gazed more we could express, in a very simplified way, the $\Delta e$ for this choice as $\Delta e_A = A_1 - A_2 * u = 2 - 1 * 0.3 = 1.7$. Consider now two new items with identical $\Delta$Value but higher magnitude of the $\Sigma$Value, $B_1 = 10$ and $B_2 = 9$. Notice that since $\Delta$Value is the same, this choice in absence of attentional effect should be considered of identical difficulty than in case A ($A_1 - A_2 = B_1 - B_2 = 1$), and therefore the agent should be neither more, nor less confident. But, keeping the same attentional factors than for the first set, we have that the $\Delta e$ between the items increases, $\Delta e_B = B_1 - B_2 * u = 10 - 9 * 0.3 = 7.3$ ($\Delta e_A < \Delta e_B$). This effect would not be observed if attention affected evidence accumulation in an additive way ($A_1 - (A_2 - u) = B_1 - (B_2 - u)$). Our empirical confidence data therefore provide further support to a multiplicative (*Smith and Krajbich, 2019*) instead of additive effect of attention into goal-relevant information. Overall, these data speak in favour of a coding scheme in which the goal sets, from the beginning of the task, the allocation of attention and, by doing so, influences first-order processes such as choice, but also second order process such as confidence. Further empirical data will be required to test this idea more stringently.

The idea that the goal of the task plays a central role in shaping value-based decisions should not be surprising. Indeed, value-based decision is often called goal-directed choice. Nevertheless, there has been a surprisingly little amount of experimental work in which the behavioural goal has been directly manipulated as the key experimental variable for studying the relation between attention and value. Notable exceptions are two recent studies from *Frömer et al., 2019* and *Kovach et al., 2014*. In the first study (*Frömer et al., 2019*), participants were shown a set of four items and asked, in half of the trials, to determine the best item and, in the second half, the worst item. In line with our findings, they found that behaviour and neural activity in the 'value network', including vmPFC and striatum, was determined by goal-congruency and did not simply reflected the expected reward. In the second study, *Kovach et al., 2014* implemented a design similar to our value-based experiment in which participants were required to indicate the item to keep and the one to discard. They found, similarly to our findings in the value-based experiment, that the overall pattern of attention was mostly allocated according to the task goal. However, in the first few hundred milliseconds, these authors found that attention was directed more prominently to the most valuable item in both conditions. We did not replicate this last finding in our experiment (see *Figure 3C and D* and *Appendix 2—figure 2*, showing that fixations were randomly allocated during the early moments of the trial). One possible reason for this discrepancy is that the experiment by Kovach and colleagues presented both items on the screen at the beginning of the task – unlike in our task, in which the item was presented in a gaze-contingent way (to avoid processing in the visual periphery). This setting might have triggered an initial and transitory bottom-up attention grab from the most valuable (and often most salient) item before the accumulation process started.

To gain a deeper insight into our findings, we developed a normative model of optimal information acquisition rooted in economic decision theory. Our model shows that in many real-life scenarios in which the decision set is larger than two, the optimal strategy to gather and integrate information depends on the behavioural goal. Intuitively, this happens because new information is all the more useful the more likely it is to change the behavioural output, that is, the choice. When the agent needs to select the best item in a set, it is best to search for evidence that it is more likely to affect the top of the ranking (e.g. is the best item still the best one?); information that changes the middle or the bottom of the ranking is instead less valuable (e.g. is the item ranked as seven is now ranked as six?) because it would not affect the behavioural output. When choosing which item to discard, instead, the optimal strategy involves acquiring information most informative of the *bottom* of the ranking and not the top. We propose that even in the context of binary choice studied here, humans might still deploy this normative strategy (for multi-alternative choice), and that while it does not provide a normative advantage, it is not suboptimal. Further work in which the size of the set is increased would be required to test this idea more stringently. Notably, two recent pre-prints have also introduced models to explain how the attentional patterns in choice are generated assuming

optimal information sampling (*Jang et al., 2020*; *Callaway et al., 2020*). Both models are based on Bayesian updates of value beliefs, with visual attention playing a role in selecting the information to sample. However, both studies were developed considering only a standard appetitive like frame (*Krajbich et al., 2010* study was used as benchmark in both cases).

The most far reaching conclusion of our work is that context and behavioural demand have a powerful effect on how information is accumulated and processed. Notably, our data show that this is a general effect that spans both more complex value-based choice and simpler perceptual choice. Our conclusion is that, given the limited computational resources of the brain, humans have developed a mechanism that prioritises the processing or recollection of the information that is most relevant for the behavioural response that is required. This has profound implications when we think about the widespread effect of contextual information on decision making that has been at the core of the research in psychology, behavioural economics and more recently neuroeconomics (*Kahneman and Tversky, 1984*; *Kahneman and Tversky, 2000*; *Camerer et al., 2004*; *De Martino et al., 2006*; *Glimcher and Fehr, 2014*). Most of these contextual or framing effects have been labelled as 'biases' because, once one strips away the context, the actual available options should remain identical. However, this perspective may not be putting enough emphasis on the fact that the decision maker has to construct low dimensional (and therefore imperfect) representations of the decision problem. As we have shown here, from the very beginning of the deliberation process, the context — even when it is simple (*like/dislike*, *most/fewest*) or irrelevant from the experimenter perspective — affects which information is processed, recalled, or attended to, with effects that spread into post-decision processing such as confidence estimation. This, as a consequence, will produce profoundly dissimilar representations according to the behavioural goal set by the context. With this shift of perspective, it may well be the case that many of the so-called 'biases' will be shown in a new light, given that participants are dealing with very different choices once the behavioural goal changes. This viewpoint might provide a more encouraging picture of the human mind, by suggesting that evolution has equipped us well to deal with ever-changing environments in the face of limited computational resources.

## Materials and methods

### Procedure

#### Value experiment

At the beginning of this experiment, participants were asked to report on a scale from £0 to £3 the maximum they would be willing to pay for each of 60 snack food items. They were informed that this bid will give them the opportunity to purchase a snack at the end of the experiment, using the BDM (*Becker et al., 1964*), which gives them incentives to report their true valuation. Participants were asked to fast for 4 hr previous to the experiment, expecting they would be hungry and willing to spend money to buy a snack.

After the bid process, participants completed the choice task: in each trial, they were asked to choose between two snack items, displayed on-screen in equidistant boxes to the left and right of the centre of the screen (*Figure 1A*). After each binary choice, participants also rated their subjective level of confidence in their choice. Pairs were selected using the value ratings given in the bidding task: using a median split, each item was categorised as high- or low-value for the agent; these were then combined to produce 15 high-value, 15 low-value, and 30 mixed pairs, for a total of 60 pairs tailored to the participant's preferences. Each pair was presented twice, inverting the position to have a counterbalanced item presentation.

The key aspect of our experimental setting is that all participants executed the choice process under two framing conditions: (1) a *like* frame, in which participants were asked to select the item that they liked the most, that is, the snack that they would prefer to eat at the end of the experiment and (2) a *dislike* frame in which participants were asked to select the item that they liked the least, knowing that this is tantamount to choosing the other item for consumption at the end of the experiment. See *Figure 1A* for a diagram of the task.

After four practice trials, participants performed a total of 6 blocks of 40 trials (240 trials in total). *Like* and *dislike* frames were presented in alternate blocks and the order was counterbalanced across participants (120 trials per frame). An icon in the top-left corner of the screen ('thumbs up' for *like*

and 'stop sign' for *dislike*) reminded participants of the choice they were required to make; this was also announced by the investigator at the beginning of every block. The last pair in a block would not be first in the subsequent block.

Participants' eye movements were recorded throughout the choice task and the presentation of food items was gaze-contingent: participants could only see one item at a time depending on which box they looked at; following *Folke et al., 2017*, this was done to reduce the risk that participant, while gazing one item, would still look at the other item in their visual periphery.

Once all tasks were completed, one trial was randomly selected from the choice task. The BDM bid value of the preferred item (the chosen one in the *like* frame and the unchosen one in the *dislike* frame) was compared with a randomly generated number between £0 and £3. If the bid was higher than the BDM generated value, an amount equivalent to the BDM value was subtracted from their £20 payment and the participant received the food item. If the bid was lower than the generated value, participants were paid £20 for their time and did not receive any snack. In either case, participants were required to stay in the testing room for an extra hour and were unable to eat any food during this time other than food bought in the auction. Participants were made aware of the whole procedure before the experiment began.

## Perceptual experiment

Perceptual Experiment had a design similar to the one implemented in Value Experiment, except that alternatives were visual stimuli instead of food items. In this task, participants had to choose between two circles filled with dots (for a schematic diagram see *Figure 1*), again in two frames. In the *most* frame, they had to pick the one with more dots; and the one with fewer dots in the *fewest* frame. The total number of dots presented in the circles could have three numerosity levels (=50, 80 and 110 dots). For each pair in those three levels, the dot difference between the circles varied in 10 percentage levels (ranging from 2% to 20% with 2% steps). To increase the difficulty of the task, in addition to the target dots (blue-green coloured), distractor dots (orange coloured) were also shown. The number of distractor dots was 80% of that of target dots (40, 64, 88 for the three numerosity levels, respectively). Pairs were presented twice and counterbalanced for item presentation. After 40 practice trials (20 initial trials with feedback, last 20 without), participants completed 3 blocks of 40 trials in the *most* frame and the same number in the *fewest* frame; they faced blocks with alternating frames, with a presentation order counterbalanced across participants. On the top left side of the screen a message indicating *Most* or *Fewest* reminded participants of the current frame. Participants reported their confidence level in making the correct choice at the end of each trial. As in the previous experiment, the presentation of each circle was gaze contingent. Eye tracking information was recorded for each trial. Participants received £7.5 for 1 hr in this study.

Both tasks were programmed using Experiment Builder version 2.1.140 (SR Research).

## Exclusion criteria

### Value experiment

We excluded individuals that met any of the following criteria:

1. Participants used less than 25% of the BDM value scale.
2. Participants gave exactly the same BDM value for more than 50% of the items.
3. Participants used less than 25% of the choice confidence scales.
4. Participants gave exactly the same confidence rating for more than 50% of their choices.
5. Participants did not comply with the requirements of the experiment (i.e., participants that consistently choose the *preferred* item in *dislike* frame or their average blink time is over 15% of the duration of the trials).

### Perceptual experiment

Since for Perceptual Experiment the assessment of the value scale is irrelevant, we excluded participants according to criteria 3, 4, and 5.

## Participants

### Value experiment

Forty volunteers gave their informed consent to take part in this research. Of these, 31 passed the exclusion criteria and were included in the analysis (16 females, 17 males, aged 20–54, mean age of 28.8). One participant was excluded for using less than 25% of the bidding scale (criteria 1). A second participant was excluded according to criteria 2 as they frequently gave the same bid value. A further four participants were excluded under criteria 4. Three participants were excluded due to criteria 5. In the latter case, one participant's eye-tracking data showed the highest number of blink events and made choices without fixating any of the items; the other two did not comply with the frame manipulation. To ensure familiarity with the snack items, all the participants in the study had lived in the UK for 1 year or more (average 17 years).

### Perceptual experiment

Forty volunteers were recruited for the second experiment. Thirty-two participants (22 females, 10 males, aged 19–50, mean age of 26.03) were included in the behavioural and regression analyses. Three participants were excluded for repetition of the confidence rating (criteria 4). Five participants were removed for criteria 5: four of them had performance close to chance level or did not followed the frame modification, and one participant presented difficulties for eye-tracking. Due to instability in parameter estimation (problem of MCMC convergence), four additional participants were removed from the GLAM modelling analysis.

All participants signed a consent form, and both studies were done following the approval given by the University College London, Division of Psychology and Language Sciences ethics committee.

## Eye-tracking

### Value and perceptual experiments

An Eyelink 1000 eye-tracker (SR Research) was used to collect the visual data. Left eye movements were sampled at 500 Hz. Participants rested their heads over a head support in front of the screen. Display resolution was of $1024 \times 768$ pixels. To standardise the environmental setting and the level of detectability, the lighting was monitored in the room using a dimmer lamp and light intensity was maintained at $4 \pm 0.5$ lx at the position of the head-mount when the screen was black.

Eye-tracking data were analysed initially using Data Viewer (SR Research), from which reports were extracted containing details of eye movements. We defined two interest-areas (IA) for left and right alternatives: two squares of $350 \times 350$ pixels in Value Experiment and two circles of 170 pixels of radius for Perceptual Experiment. The data extracted from the eye-tracker were taken between the appearance of the elements on the screen (snack items or circle with dots in experiments 1 and 2, respectively) and the choice button press (confidence report period was not considered for eye data analysis).

The time participants spent fixating on each IA was defined the dwelling time (DT). From it, we derived a difference in dwelling time ($\Delta$DT) for each trial by subtracting DT of the right IA minus the DT of the left IA. Starting and ending IA of each saccade were recorded. This information was used to determine the number of times participants alternated their gaze between IAs, that is, 'gaze shifts'. The total number of gaze shifts between IAs was extracted for each trial, producing the gaze shift frequency (GSF) variable.

## Data analysis: behavioural data

Behavioural measures during *like/dislike* and *most/fewest* frames were compared using statistical tests available in SciPy. Sklern toolbox in Python was used to perform logistic regressions on choice data. Fixation time series analysis was performed following *Kovach et al., 2014* methodology. We segmented the time series of all the trials in samples of 10 ms. We fixed all the trials time series to the beginning of the trial, when participant could start exploring the gaze-contingent alternatives. We considered an analysis window of 2000 ms after the presentation of stimuli for all the trials. Please notice that not all the trials have the same duration and no temporal normalisation was performed in this analysis. For each time sample, we obtained the gaze position and the difference in evidence (i.e. $\Delta$Value or $\Delta$Dots) for all trials across participants and then Pearson correlation was calculated. Permutations testing was used to assess the difference between the time series in *like/*

*dislike* and *most/fewest* frames. Instantaneous fixations (across trials and frames) were shuffled 200 times to create a null distribution of the difference of correlation coefficients between frames. False discovery rate (FDR) was used to correct for multiple tests the p-values obtained from the permutation test ($\alpha \leq 0.01$). All the hierarchical analyses were performed using lme4 package (*Bates et al., 2015*) for R integrated in a Jupyter notebook using the rpy2 package (https://rpy2.readthedocs.io/en/latest/). For choice models, we predicted the log odds ratio of selecting the item appearing at the right. Fixed-effects confidence interval were calculated by multiplying standard errors by 1.96. Additionally, we predicted confidence using a linear mixed-effects model. Predictors were all z-scored at participant level. Matplotlib/Seaborn packages were used for visualisation.

## Data analysis: attentional model - GLAM

To get further insight on potential variations in the evidence accumulation process due to the change in frames we used the Gaze-weighted Linear Accumulator Model (GLAM) developed by *Thomas et al., 2019*. GLAM is part of the family of linear stochastic race models in which different alternatives (i, i.e. left or right) accumulate evidence ($E_i$) until a decision threshold is reached by one of them, determining the chosen alternative. The accumulator for an individual option is described by the following expression:

$$E_i(t) = E_i(t-1) + \nu R_i + \epsilon_t$$
$$\text{with } \epsilon_t \sim N(0, \sigma) \text{ and } E_i(t=0) = 0$$

(5)

With a drift term ($\nu$) controlling the speed of relative evidence ($R_i$) integration and i.i.d. noise terms with normal distribution (zero-centered and standard deviation $\sigma$). $R_i$ is a term that expresses the amount of evidence that is accumulated for item i at each time point t. This is calculated as follows. We denote by $g_i$, the relative gaze term, calculated as the proportion of time that participants observed item i:

$$g_i = \frac{DT_i}{DT_1 + DT_2}$$

(6)

with DT as the dwelling time for item i during an individual trial. Let $r_i$ denote the value for item i reported during the initial stage of the experiment. We can then define the average absolute evidence for each item ($A_i$) during a trial:

$$A_i = g_i r_i + (1 - g_i)\gamma r_i$$

(7)

This formulation considers a multiplicative effect of the attentional component over the item value, capturing different rates of integration when the participant is observing item i or not (unbiased and biased states, respectively). The parameter $\gamma$ is the gaze bias parameter: it controls the weight that the attentional component has in determining absolute evidence. *Thomas et al., 2019* interpret $\gamma$ as follows: when $\gamma = 1$, bias and unbiased states have no difference (i.e. the same r is added to the average absolute evidence regardless the item is attended or not); when $\gamma < 1$, the absolute evidence is discounted for the biased condition; when $\gamma < 0$, there is a leak of evidence when the item is not fixated. Following *Thomas et al., 2019*, in our analysis, we allowed $\gamma$ to take negative values, but our results do not change if $\gamma$ is restricted to [0, 1] (*Appendix 6—figure 2*). Finally, the relative evidence of item i, $R_i^*$, is given by:

$$R_i^* = A_i - max_j(A_j) = A_i - A_j \; \rightarrow R_{right}^* = -R_{left}^*$$

(8)

Since our experiment considers a binary choice the original formulation of the model (*Thomas et al., 2019*), proposed for more than two alternatives, $R_i^*$ is reduced to subtract the average absolute evidence of the other item. Therefore, for the binary case, the $R_i^*$ for one item will be additive inverse of the other, for example if the left item has the lower value, we would have $R_{left}^* < 0$ and $R_{right}^* > 0$. Additionally, in their proposal for GLAM, *Thomas et al., 2019* noted that $R_i^*$ range will depend on the values that the participant reported, for example evidence accumulation may appear smaller if participant valued all the items similarly, since $R_i^*$ may be lower in magnitude. This may not represent the actual evidence accumulation process since participants may be sensitive to marginal

differences in relative evidence. To account for both of these issues, a logistic transformation is applied over $R_i^*$ using a scaling parameter $\tau$:

$$R_i = \frac{1}{1 + e^{-\tau R_i^*}} \tag{9}$$

In this case, $R_i$ will be always positive and the magnitude of the difference between $R_{left}$ and $R_{right}$ will be controlled by $\tau$, for example higher $\tau$ will imply a bigger difference in relative evidence (and hence accumulation rate) between left and right item. In the case that $\tau = 0$ the participant will not present any sensitivity to differences in relative evidence.

Given that $R_i$ represents an average of the relative evidence across the entire trial, the drift rate in $E_i$ can be assumed to be constant, which enables the use of an analytical solution for the passage of time density. Unlike aDDM (*Krajbich et al., 2010*), GLAM does not deal with the dynamics of attentional allocation process in choice. Details of these expressions are available at *Thomas et al., 2019*. In summary, we have four free parameters in the GLAM: $\nu$ (drift term), $\gamma$ (gaze bias), $\tau$ (evidence scaling), and $\sigma$ (normally distributed noise standard deviation).

The model fit with GLAM was implemented at a participant level in a Bayesian framework using PyMC3 (*Salvatier et al., 2016*). Uniform priors were used for all the parameters:

$$v \sim \text{Uniform}\left(1^{-10}, 0.01\right)$$

$$\gamma \sim \text{Uniform}(-1, 1)$$

$$\sigma \sim \text{Uniform}\left(1^{-10}, 5\right)$$

$$\tau \sim \text{Uniform}(0, 5)$$

## Value experiment

We fitted the model for each individual participant and for *like* and *dislike* frames, separately. To model participant's behaviour in the *like* frame, we used as input for GLAM the RTs and choices, plus BDM bid values and relative gaze for left and right alternatives for each trial. The original GLAM formulation (as presented above) assumes that evidence is accumulated in line with the preference value of a particular item (i.e. 'how much I like this item'). When information about visual attention is included in the model, the multiplicative model in GLAM assumes that attention will boost the evidence accumulation already defined by value. Our proposal is that evidence accumulation is a flexible process in which attention is attracted to items based on the match between their value and task-goal (accept or reject) and not based on value alone, as most of the previous studies have assumed. Since in the *dislike* frame the item with the lower value becomes relevant to fulfil the task, we considered the opposite value of the items ($r_{i,dislike} = 3 - r_{i,like}$, e.g. item with value 3, the maximum value, becomes value 0) as an input for GLAM fit. For both conditions, model fit was performed only on even-numbered trials using Markov-Chain-Monte-Carlo sampling, using implementation for No-U-Turn-Sampler (NUTS), four chains were sampled, 1000 tuning samples were used, 2000 posterior samples to estimate the model parameters. The convergence was diagnosed using the Gelman-Rubin statistic ($|\hat{R} - 1| < 0.05$) and also corroborating that the effective sample size (ESS) was high (ESS >100) for the four parameters ($\nu$, $\gamma$, $\sigma$, and $\tau$). Considering all the individual models, we found divergences in less than 3% of the estimated parameters. Model comparison was performed using Watanabe-Akaike Information Criterion (WAIC) scores available in PyMC3, calculated for each individual participant fit.

Pointing to check if the model replicates the behavioural effects observed in the data (*Palminteri et al., 2017*), simulations for choice and response time (RT) were performed using participant's odd trials, each one repeated 50 times. For each trial, value and relative gaze for left and right items were used together with the individual estimated parameters. Random choice and RT (within a range of the minimum and maximum RT observed for each particular participant) were set for 5% of the simulations, replicating the contaminating process included in the model as described by *Thomas et al., 2019*.

Additionally, we simulated the accumulation process in each trial to obtain a measure of balance of evidence (*Vickers, 1970*; *Vickers, 1979*) for each trial. The purpose of this analysis was to replicate the effect of ΣValue over confidence (check *Results* for details) and check if it arises from the accumulation process and its interaction with attention. Balance of evidence in accumulator models has been used previously as an approximation to the generation of confidence in perceptual and value-based decision experiments (*Vickers, 1979*; *Smith and Vickers, 1988*; *De Martino et al., 2013*). Consequently, using the value of the items and gaze ratio from odd-numbered trials, we simulated two accumulators (*Equation 5*), one for each alternative. Our simulations used the GLAM parameters obtained from participant's fit. Once the boundary was reached by one of the stochastic accumulators (fixed boundary = 1), we extracted the simulated RT and choice. The absolute difference between the accumulators when the boundary was reached ($\Delta e = |E_{right}(t_{final}) - E_{left}(t_{final})|$) delivered the balance of evidence for that trial. In total 37,200 trials were simulated (10 repetitions for each one of the trials done by the participants). A linear regression model to predict simulated $\Delta e$ using $|\Delta Value|$, simulated RT and ΣValue as predictors was calculated with the pooled participants' data. This model was chosen since it was the most parsimonious model obtained to predict participant's confidence in the Value Experiment (*Appendix 4—figure 1*). The best model includes GSF as predictor in the regression, but since GLAM does not consider the gaze dynamics we removed it from the model. $\Delta e$ simulations using a GLAM without gaze influence (i.e. equal gaze time for each alternative) were also generated, to check if gaze difference was required to reproduce ΣValue effect over confidence. The parameters fitted for individual participants were also used in the no-gaze difference simulation. The same linear regression model ($\Delta e \sim |\Delta Value| +$ simulated RT + ΣValue) was used with the data simulated with no-gaze difference.

## Perceptual experiment

In the Perceptual Experiment, we repeated the same GLAM analysis done in Value Experiment. Due to instabilities in the parameters' fit for some participants, we excluded four extra participants. Twenty-eight participants were included in this analysis. Additionally, the GLAM fit in this case was done removing outlier trials, that is, trials with RT higher than 3 standard deviations (within participant) or higher than 20 s. Overall less than 2% of the trials were removed. For *most* frame, relative gaze and perceptual evidence (number of dots) for each alternative were used to fit choice and RT. In a similar way to the consideration taken in the *dislike* case, we reassigned the perceptual evidence in the *fewest* frame ($r_{i,fewest} = 133 - r_{i,most} + 40$, considering that 133 is the higher number of dots presented and 40 dots the minimum) in a way that the options with higher perceptual evidence in the *most* frame have the lower evidence in the *fewest* frame. The same MCMC parameters used to fit the model for each participant in the Value Experiment were used in this case (again, only even-numbered trials were used to fit the model). As in the Value Experiment, model convergence was assessed using $\hat{R}$ and ESS. Overall, we observed divergences in less than 2% of parameter estimations across participants. Behavioural out-of-sample simulations (using the odd-numbered trials) and balance of evidence simulations (33,600 trials simulated in the Perceptual Experiment) were considered in this analysis. We tested the effect of ΣDots over confidence with a similar linear regression model than the one used in the Value Experiment. Pooled participants' data for $|\Delta Dots|$, simulated RT and ΣDots was used to predict $\Delta e$. $\Delta e$ simulations using a GLAM without gaze asymmetry were also calculated in this case. All the figures and analysis were done in Python using GLAM toolbox and custom scripts.

## Acknowledgements

This study was funded by a Sir Henry Dale Fellowship (102612/A/13/Z) awarded to Benedetto De Martino by the Wellcome Trust. Pradyumna Sepulveda was funded by the Chilean National Agency for Research and Development (ANID)/Scholarship Program/DOCTORADO BECAS CHILE/2017– 72180193. We thank Antonio Rangel for his valuable comments on an earlier version of the manuscript and Mariana Zurita for the help in the proofreading of the manuscript.

## Additional information

### Funding

| Funder | Grant reference number | Author |
|---|---|---|
| Chilean National Agency for Research and Development | Graduate student scholarship - DOCTORADO BECAS CHILE/2017 - 72180193 | Pradyumna Sepulveda |
| Wellcome Trust | Sir Henry Dale Fellowship (102612 /A/13/Z) | Benedetto De Martino |

The funders had no role in study design, data collection and interpretation, or the decision to submit the work for publication.

### Author contributions

Pradyumna Sepulveda, Conceptualization, Resources, Data curation, Software, Formal analysis, Funding acquisition, Validation, Investigation, Visualization, Methodology, Writing - original draft, Project administration, Writing - review and editing; Marius Usher, Supervision, Methodology, Writing - review and editing; Ned Davies, Amy A Benson, Data curation, Formal analysis, Investigation, Writing - review and editing; Pietro Ortoleva, Conceptualization, Supervision, Methodology, Writing - original draft, Writing - review and editing, Developed model for optimal information acquisition; Benedetto De Martino, Conceptualization, Supervision, Funding acquisition, Methodology, Writing - original draft, Project administration, Writing - review and editing

### Author ORCIDs

Pradyumna Sepulveda (iD) https://orcid.org/0000-0003-0159-6777
Marius Usher (iD) http://orcid.org/0000-0001-8041-9060
Ned Davies (iD) https://orcid.org/0000-0002-3357-8576
Amy A Benson (iD) http://orcid.org/0000-0002-8239-5266
Pietro Ortoleva (iD) https://orcid.org/0000-0002-5943-6621
Benedetto De Martino (iD) https://orcid.org/0000-0002-3555-2732

### Ethics

Human subjects: All participants signed a consent form and both studies were done following the approval given by the University College London, Division of Psychology and Language Sciences ethics committee (project ID number 1825/003).

### Decision letter and Author response

Decision letter https://doi.org/10.7554/eLife.60705.sa1
Author response https://doi.org/10.7554/eLife.60705.sa2

## Additional files

### Supplementary files

• Transparent reporting form

### Data availability

Data and codes used in this study are available at the Brain Decision Modelling Lab GitHub https://github.com/BDMLab/Sepulveda_et_al_2020 (copy archived at https://archive.softwareheritage.org/swh:1:rev:a04585ff20b1389c713709b543a1af420bd300c1/).

The following dataset was generated:

| Author(s) | Year | Dataset title | Dataset URL | Database and Identifier |
|---|---|---|---|---|
| Sepulveda P, Usher M, Davies N, Benson AA, Ortoleva P, De Martino B | 2020 | Sepulveda_et_al_2020 | https://github.com/BDMLab/Sepulveda_et_al_2020 | GitHub repository, Sepulveda et al 2020 |

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

# Appendix 1

## Task framing differences

### Value experiment

We examined how the frame manipulation impacted overall performance (*Appendix 1—figure 1A*). We defined 'accuracy' as the proportion of trials in which participant's reported values (BDM bid) correctly predicted their binary decision, that is, they select the item with highest value in the *like* frame and the one with lowest value in the *dislike* frame. Overall accuracy was not significantly different in both frames (Mean$_{Like}$ = 0.77; Mean$_{Dislike}$ = 0.75, t(30) = 1.71; p=0.1). We also found that participants had slightly slower reaction times (RTs) in the *dislike* frame (Mean$_{Like}$ = 2858.2 ms, Mean$_{Dislike}$ = 3152.7 ms; t(30) = −2.52; p<0.05). Participants reported lower confidence in the *dislike* frame (Mean$_{\Delta Confidence}$ = 0.19; t(30) = 4.49; p<0.001) and shifted their gaze (gaze shift frequency, GSF) between items more during *dislike* trials (Mean$_{\Delta|GSF|}$ = −0.110; t(30) = −2.99; p<0.01). These results overall suggest that the subjects may have found the *dislike* condition slightly less intuitive. Although this did not affect their performance, it slightly reduced their confidence and increased RT and GSF.

As observed in previous studies (*Folke et al., 2017*; *De Martino et al., 2013*), we found that choice accuracy was modulated by confidence: decisions in which participants reported high-confidence were more accurately predicted by the value estimate collected before the experiment – the slope of the logistic curve is steeper in the case of high confidence (*Figure 1B*, *Results* section). In this study, this effect is replicated in both *like* (low confidence: β = 0.769; high confidence: β = 1.633) and *dislike* (low confidence: β = −0.642; high confidence: β = −1.363) frames. Note that the inversion of the sign of the slopes in *like* vs *dislike* frames indicate that participants were performing the task correctly (Δβ$_{Like-Dislike}$: t(30) = 8.14, p<0.001), selecting the item with lower value during the *dislike* frame (*Figure 1C*, *Results* section). Choice accuracy (steepness of the slopes) was not significantly different between *like* and *dislike* frames (Δ|β$_{Like-Dislike}$|: t(30) = 1.58, p=0.124).

### Perceptual experiment

We repeated the same analysis for the behavioural performance in *most* and *fewest* frames (*Appendix 1—figure 1B*). In contrast to the Value Experiment, we observed a slight reduction in accuracy in participant responses for the *fewest* frame (Mean$_{Most}$ = 0.77, Mean$_{Few}$ = 0.74, t(31) = 2.46; p<0.05); unlike the Value Experiment, however, we did not find differences in RTs (Mean$_{Most}$ = 4029.57 ms, Mean$_{Few}$ = 3975.59 ms; t(31) = 0.32; p=0.75). During the *fewest* frame participants reported lower confidence (Mean$_{\Delta Confidence}$=0.24; t(31) = 5.62; p<0.001) and shifted their gaze more between alternatives (Mean$_{\Delta|GSF|}$ = -0.17; t(31) = -4.15; p<0.001), as observed in the Value Experiment.

Participants also reported higher confidence in trials that better discriminated the number of dots (*Figure 1E*, *Results* section). This effect was replicated in both *most* (low confidence: β = 1.142; high confidence: β = 2.164) and *fewest* (low confidence: β = −1.118; high confidence: β = −2.010) frames. The inversion of the sign of the slopes in *most* vs *fewest* frames also shows that participants were performing correctly (Δβ$_{Most-Few}$: t(31) = 22.22, p<0.001); the magnitude of the slopes was not significantly different between the two frames (Δ|β$_{Most-Few}$|: t(31) = 0.79, p=0.434; *Figure 1F*, *Results* section). This pattern of results mirrors the pattern seen in the Value Experiment.

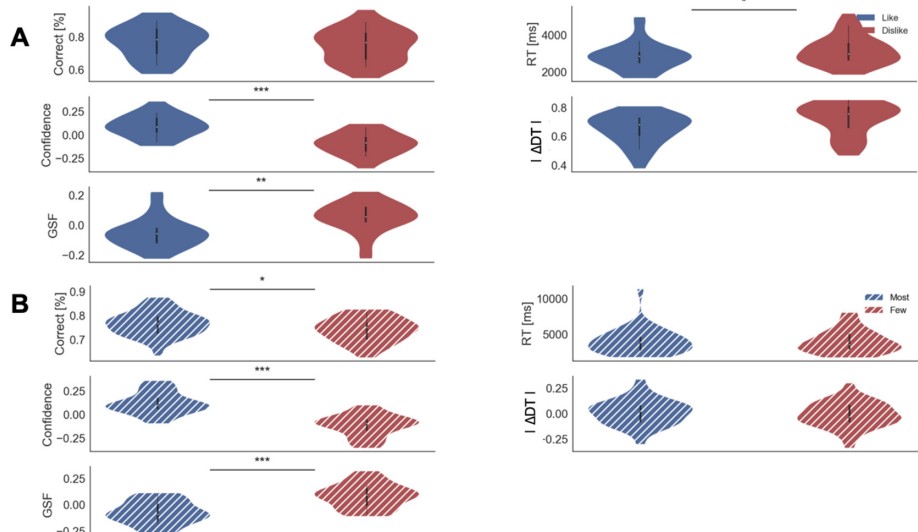

**Appendix 1—figure 1.** Behavioural results for Value (**A**) and Perceptual (**B**) Experiments. Confidence, DDT, and GSF values have been z-scored per participant. In the violin plot, red and blue areas indicate the distribution of the parameters across participants. Black bars present the 25, 50, and 75 percentiles of the data. Solid colour indicates the Value Experiment and striped colours indicate the Perceptual Experiment. RT: reaction time; ΔDT: Difference in Dwell Time; GSF: Gaze Shift Frequency.

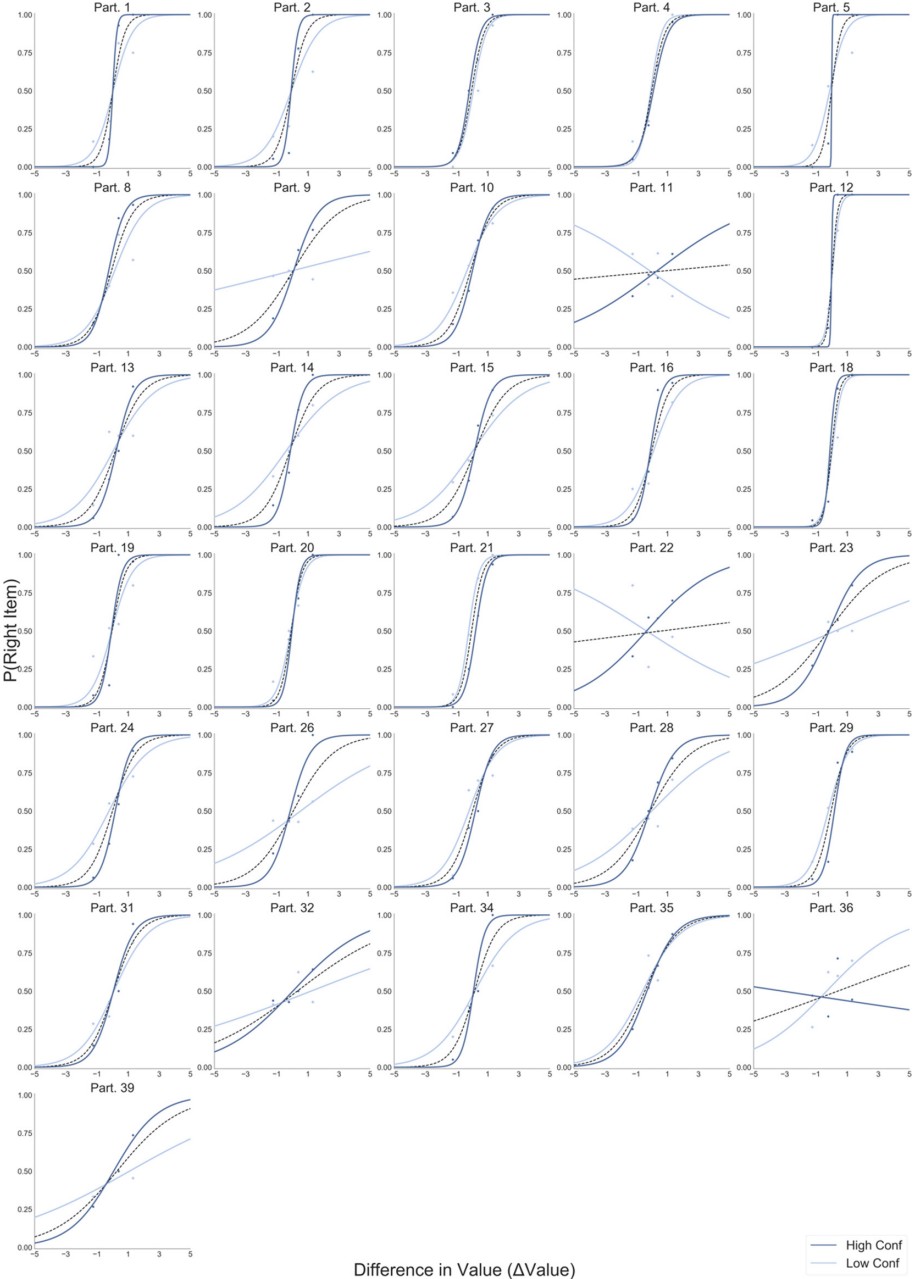

**Appendix 1—figure 2.** Logistic regression predicting choice from the difference in value between the two items (ΔValue). All participants in the Value Experiment, *like* frame, are presented. Light blue lines depict the logistic fit calculated using only low confidence trials. Dark blue lines show the logistic fit only for high confidence trials. Segmented black line considers the logistic regression calculated using all the trials.

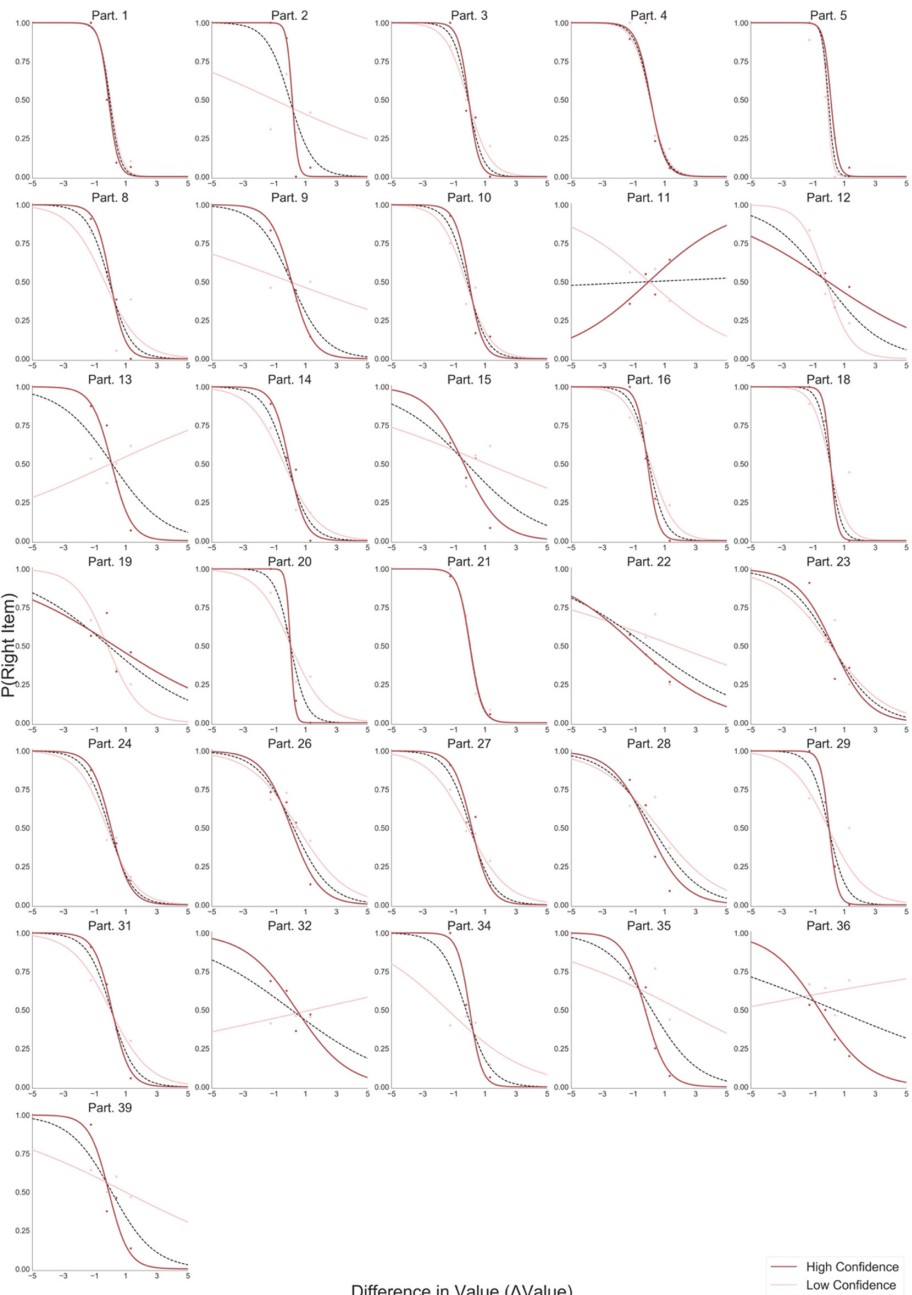

**Appendix 1—figure 3.** Logistic regression predicting choice from the difference in value between the two items (ΔValue). All participants in the Value Experiment, *dislike* frame, are presented. Light red lines depict the logistic fit calculated using low confidence trials. Dark red lines show the logistic fit using high confidence trials. Segmented black line considers the logistic regression calculated with all the trials.

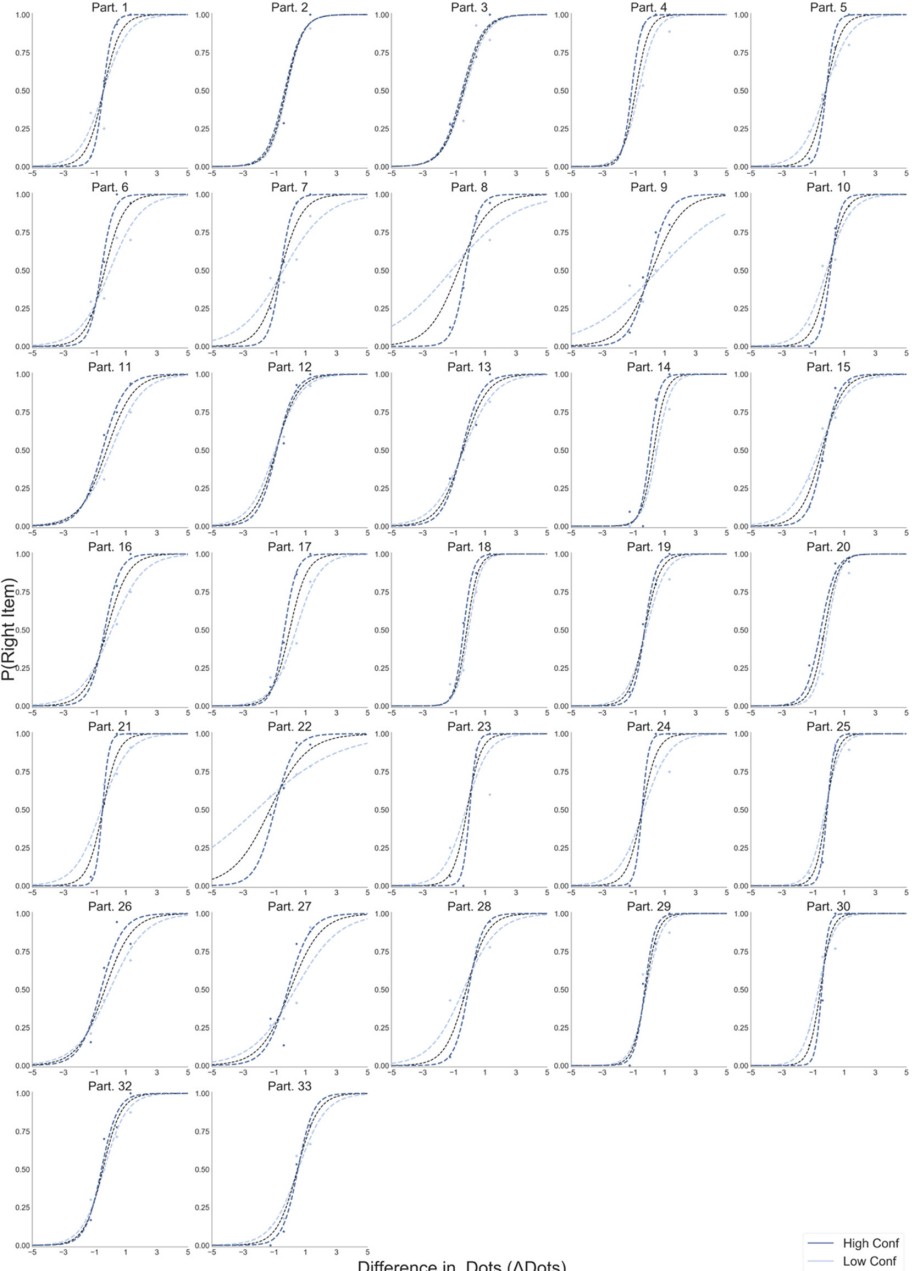

**Appendix 1—figure 4.** Logistic regression predicting choice from the difference in number of dots between the two circles (ΔDots). All participants in the Perceptual Experiment, *most* frame, are presented. Light blue lines depict the logistic fit calculated using only low confidence trials. Dark blue lines show the logistic fit only for high confidence trials. Segmented black line considers the logistic regression calculated with all the trials.

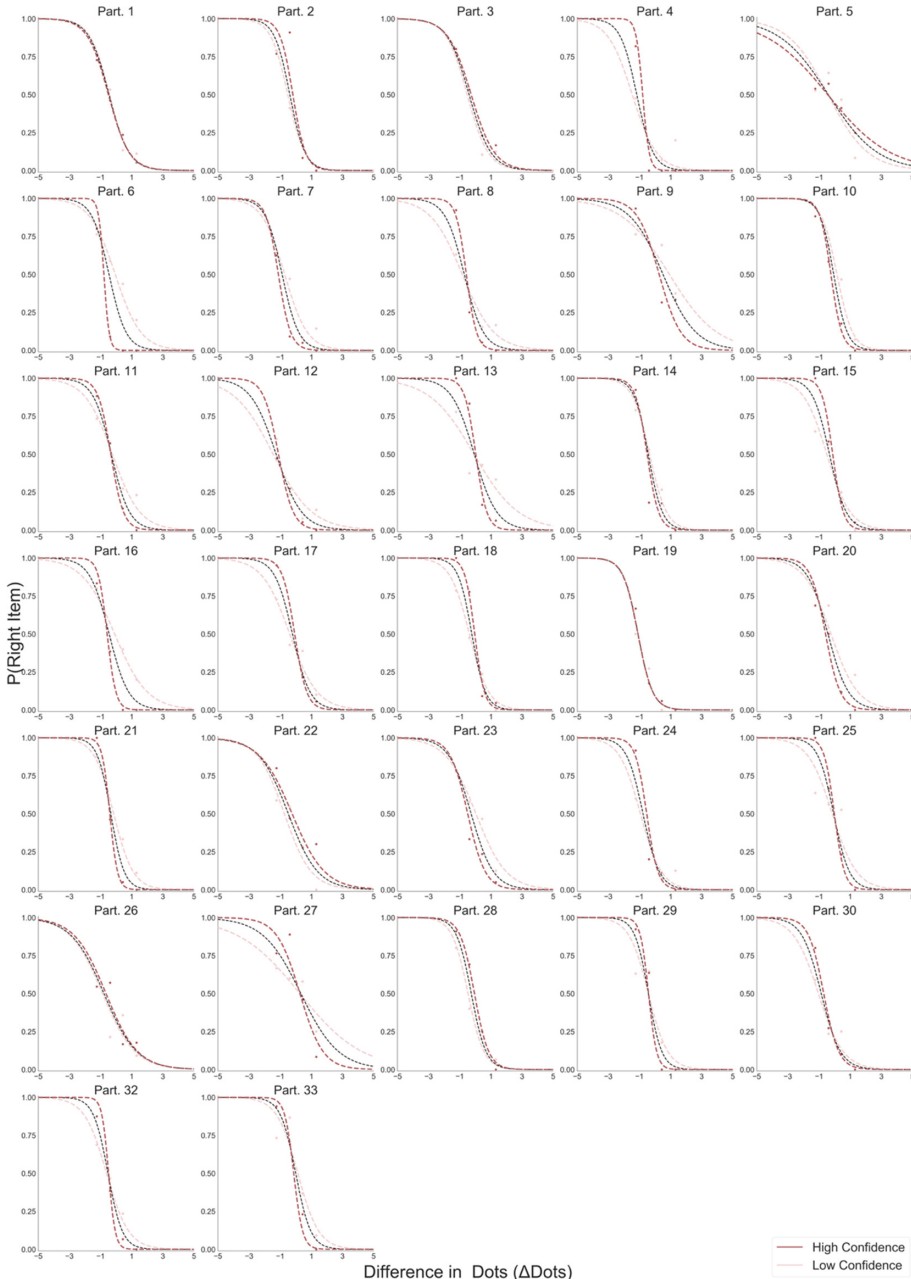

**Appendix 1—figure 5.** Logistic regression predicting choice from the difference in number of dots between the two circles (ΔDots). All participants in the Perceptual Experiment, *fewest* frame, are presented. Light red lines depict the logistic fit calculated using only low confidence trials. Dark red lines show the logistic fit only for high confidence trials. Segmented black line considers the logistic regression calculated with all the trials.

# Appendix 2

## Choice regression models

**Appendix 2—table 1.** Hierarchical logistic models for choice.

| Models | Formulas |
|---|---|
| Model 1 | Choice ~ ΔValue |
| Model 2 | Choice ~ ΔValue + Confidence |
| Model 3 | Choice ~ ΔValue + Confidence + ΣValue |
| Model 4 | Choice ~ ΔValue + Confidence + ΣValue + ΔDT |
| Model 5 | Choice ~ ΔValue + Confidence + ΣValue + ΔDT + ΔValue * Confidence |
| Model 6 | Choice ~ ΔValue + Confidence + ΣValue + ΔDT + ΔValue * Confidence + ΔValue * ΣValue |
| Model 7 | Choice ~ ΔValue + Confidence + ΣValue + ΔDT + ΔValue * Confidence + ΔValue * ΣValue + Confidence * ΔDT |
| Model 8 | Choice ~ ΔValue + Confidence + ΣValue + ΔDT + GSF + ΔValue * Confidence + ΔValue * ΣValue + Confidence * ΔDT + ΔValue * GSF |

In Value Experiment: ΔValue: difference in value; ΣValue: summed value; ΔDT: difference in dwell time; GSF: gaze shift frequency. In Perceptual Experiment similar models were compared but replacing ΔValue for ΔDots and ΣValue for ΣDots.

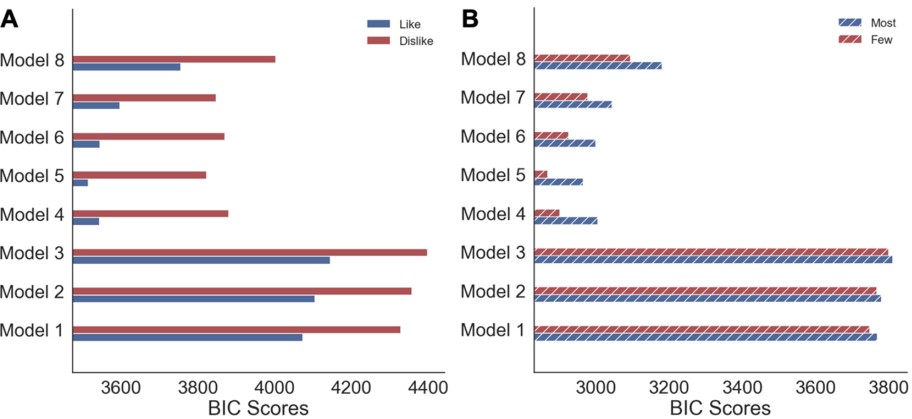

**Appendix 2—figure 1.** Model comparison of hierarchical logistic regressions for choice. (**A**) Value and (**B**) Perceptual Experiments. Solid colour indicates the Value Experiment and striped colours indicate the Perceptual experiment.

## Value experiment

Using a logistic hierarchical regression model, we investigated which factors modulated choice-proportion, defined here as the probability of choosing the item on the right side of the screen. We report here the results of the most parsimonious model (i.e. the model with a lowest BIC; *Appendix 2—figure 1*) fitted to the *like* and *dislike* frames independently (*Figure 2B*, *Results* section). In *Appendix 2—table 1*, we present the parameters for each factor included in the model. In the *like* frame, the difference in the value of the right item minus left item (ΔValue) had a positive influence on choice-proportion, that is, participants selected the items that had higher value. This is reversed in the *dislike* frame: ΔValue is now a *negative* predictor of choice, that is, participants selected the items that had lower value. In both conditions, confidence enhanced the effect of ΔValue, as shown

by the interaction between ΔValue and confidence in the *like* and *dislike* frame. These results confirm the findings presented in *Figure 1B* (Results section) while controlling for other relevant variables. Unsurprisingly, confidence and summed value (ΣValue, the added value of both alternatives) were found to show no main effect on the choice-proportion. As discussed in the Results section, gaze allocation (difference in dwell time, ΔDT) is directed to the chosen item in both frames, that is, the parameters are positive for ΔDT in *like* and *dislike* frame (*Appendix 2—table 2*).

**Appendix 2—table 2.** Statistical results for the hierarchical linear models for choice in Value Experiment.

Z-values for the regression coefficients and their statistical significance are presented for both frames. To check significant differences of the regression coefficients between like and dislike frames repeated samples t-tests between the participants' regression coefficients were calculated.

| | Choice value experiment (n = 31) | | | | | |
| | Like | | Dislike | | Like - Dislike | |
| | Z | P | Z | P | T | P |
|---|---|---|---|---|---|---|
| ΔValue | 7.917 | <0.001 | −8.652 | <0.001 | 10.74 | <0.001 |
| ΔDT | 6.448 | <0.001 | 6.75 | <0.001 | 2.31 | <0.05 |
| ΔValue x Conf | 5.446 | <0.001 | −4.681 | <0.001 | 9.55 | <0.001 |

*Confidence and ΣValue did not have a significant effect over choice in the regression.

## Perceptual experiment

As in the Value Experiment, we used a logistic hierarchical regression to determine the relevant factors modulating perceptual choice (choosing the circle with dots on the right side of the screen) (*Figure 2D*, Results section). We found that the most parsimonious model for choice was the same used in the Value Experiment, where *like* and *dislike* were replaced by *most* and *fewest* frames (*Appendix 2—figure 1B*). In the *most* frame, the difference in the number of dots of the right alternative minus the left one (ΔDots) had a positive influence over choice; that is, participants tended to select the circle with more dots. As expected, this pattern was reversed in the *fewest* frame: ΔDots was a negative predictor of choice. As in the Value Experiment, confidence modulated the effect of ΔDots in *most* and *fewest* frames. The sum of dots presented in both circles during a trial (ΣDots) was found not to have a significant effect on either frame, as expected. However, as discussed in the *Results* section, confidence was found to be a negative predictor of choice in *most* and *fewest* frames. This means participants had a bias to report higher confidence when they chose the left circle. In a similar way to the Value Experiment, participants spend more time fixating the chosen alternative in both frames, with ΔDT effect being positive in *most* and *fewest* frames (*Appendix 2—table 3*).

**Appendix 2—table 3.** Statistical results for the hierarchical logistic models for choice in Perceptual Experiment.

Z-values for the regression coefficients and their statistical significance are presented for both frames. Repeated samples t-tests between the participants' regression coefficients in most and fewest frames were calculated.

| | Choice perceptual experiment (n = 32) | | | | | |
| | Most | | Fewest | | Most - Fewest | |
| | Z | P | Z | P | T | P |
|---|---|---|---|---|---|---|
| ΔDots | 14.905 | <0.001 | −14.394 | <0.001 | 30.32 | <0.001 |
| Confidence | −2.823 | <0.01 | −6.705 | <0.001 | 6.67 | <0.001 |
| ΔDT | 10.249 | <0.001 | 10.449 | <0.001 | −2.17 | <0.05 |
| ΔDots x Conf | 8.677 | <0.001 | −6.23 | <0.001 | 23.69 | <0.001 |

*ΣDots did not have a significant effect over choice in the regression.

In a study by *Kovach et al., 2014* a design similar to our value-based experiment was implemented. Participants were required to indicate the item to keep and the one to discard. They found, similarly to our findings in the Value Experiment, that the overall pattern of attention was mostly allocated according the task goal. However, in the first few hundred milliseconds, these authors found that attention was directed more prominently to the most valuable item in both conditions. We did not replicate this last finding in our experiment, but one possible reason for this discrepancy is that the experiment by Kovach and colleagues presented both items on the screen at the beginning of the task – unlike in our task, in which the item was presented in a gaze-contingent way (to avoid processing in the visual periphery). This setting might have triggered an initial and transitory bottom-up attention grab from the most valuable (and often most salient) item before the accumulation process started.

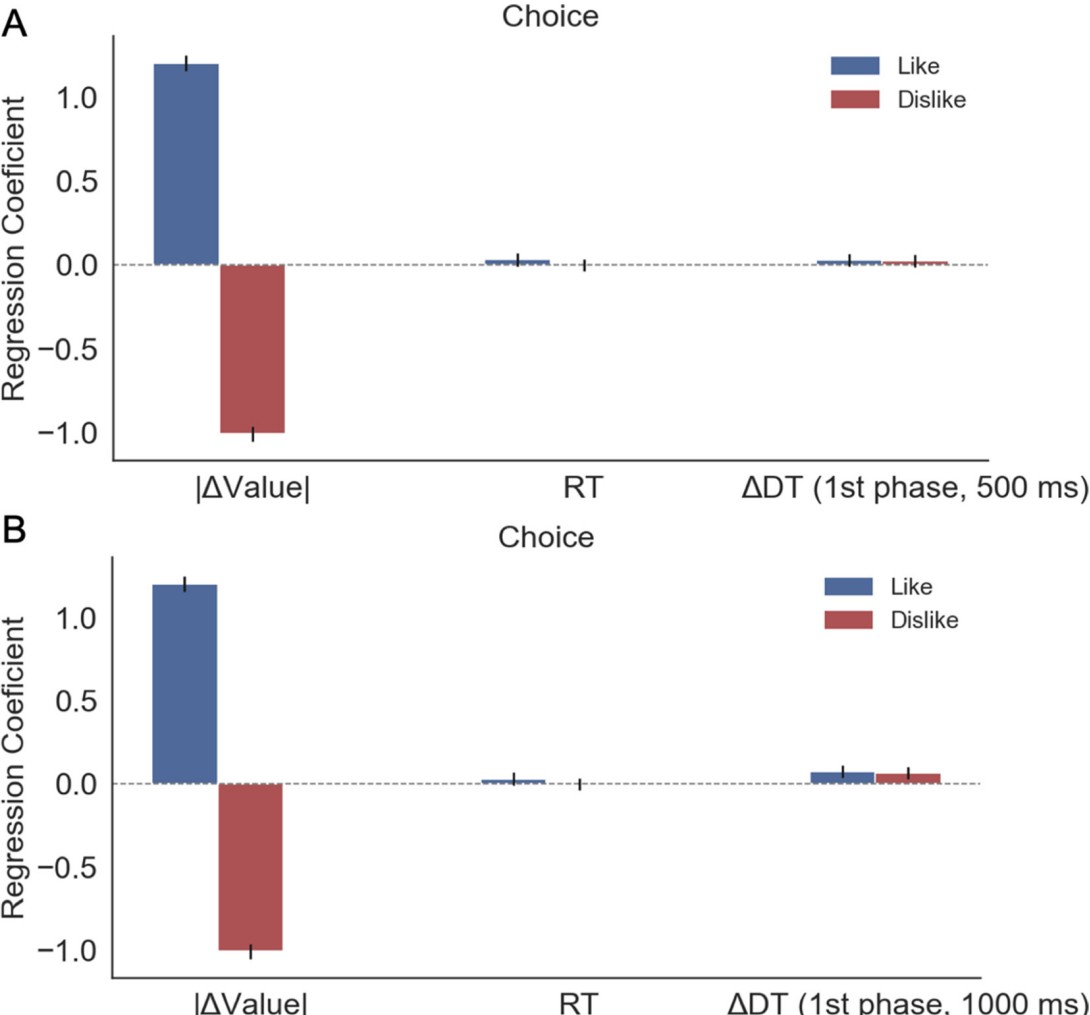

**Appendix 2—figure 2.** *Kovach et al., 2014* conducted a study in which participants have to choose food items in 'keep' and 'discard' frames, in a similar way to our Value Experiment. Gaze allocation was found to gravitate towards the chosen item overall, although during the initial moments of the trial (≈ 500 ms), they reported that gaze was directed towards the preferred item. To check if this effect appears in our Value Experiment we ran a regression model to predict choice (i.e. probability of choosing the item presented on the right side of the screen). We restricted the time to estimate ΔDT to the first 500 ms of the trial and used that variable as a predictor of choice in our model (**A**).
*Appendix 2—figure 2 continued on next page*

*Appendix 2—figure 2 continued*

We did not find a significant effect of gaze over choice in that period. This difference may be caused by the way the alternatives were presented during the decision time: while in *Kovach et al., 2014* both alternatives were always displayed on screen during deliberation time, in our experiment the presentation was gaze contingent (i.e. participants needed to explore both items at the beginning of the trial to identify the available items). (**B**) We recalculated the model considering the initial 1000 ms of the trial and we observe how ΔDT starts to increase its effect over choice. The positive effect of ΔDT over choice is only significant (z = 1.97, p<0.05) in the *like* frame; in *dislike* frame the small effect is only a trend (z = 1.081, p=0.07). However, at 1000 ms ΔDT is already starting to be allocated to the option coherent with the behavioural responses required by the frame, not to preference.

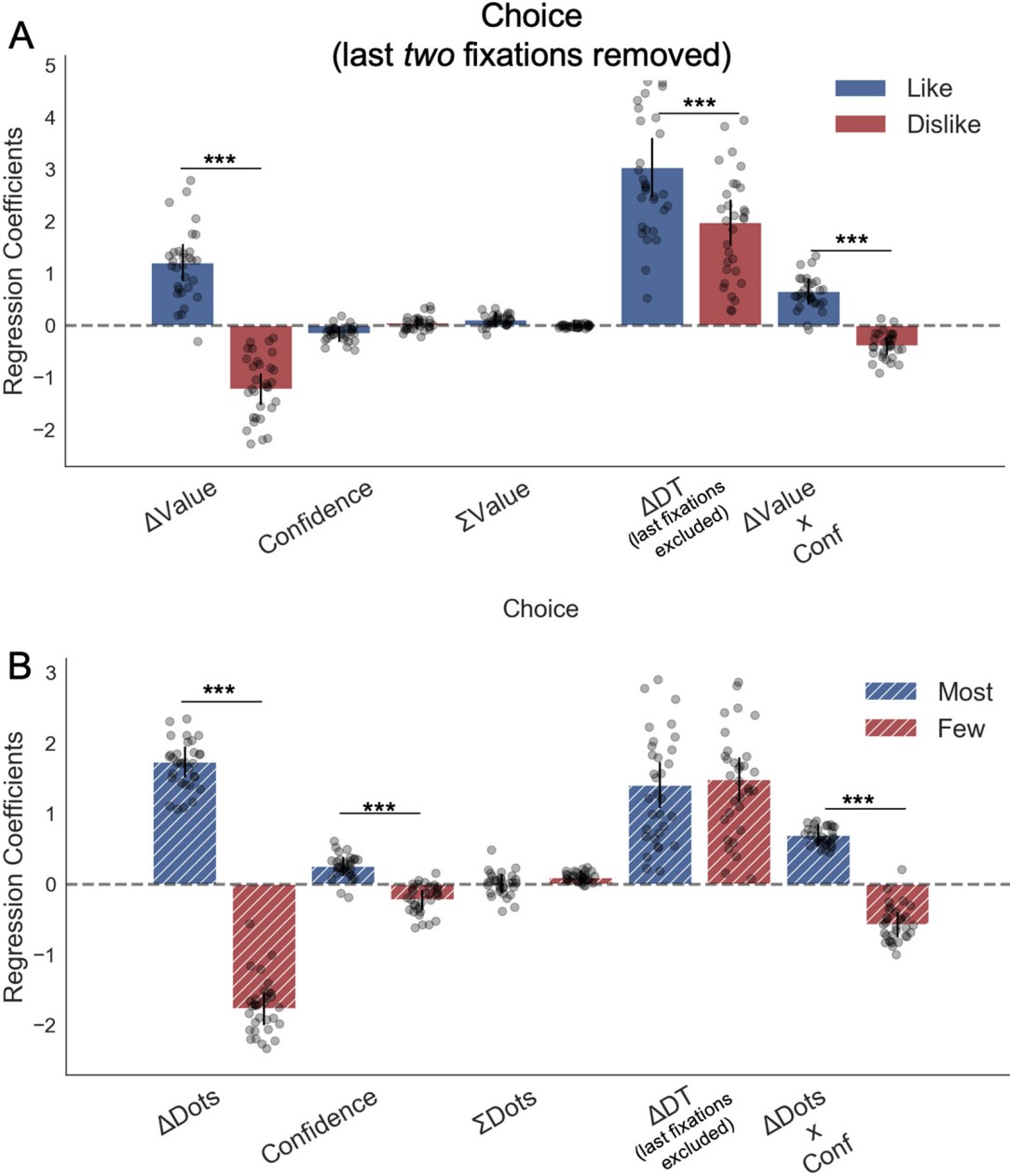

*Appendix 2—figure 3 continued on next page*

**Appendix 2—figure 3.** Choice behaviour excluding last fixations. To assess the influence that last fixations have on the goal-relevant gaze asymmetries we repeated the hierarchical logistic modelling of choice (probability of choosing right item) in Value (**B**) and Perceptual (**D**) Experiments, excluding the last two fixations from the analysis. Note the two last fixations rather than only the last fixation, because this avoids statistical artifacts. All the results from the main analysis were confirmed: participants preferentially gazed at the item they chose in both frames (positive $\Delta$DT effect in both experiments). All predictors were z-scored at the participant level. In both regression plots, bars depict the fixed-effects and dots the mixed-effects of the regression. Error bars show the 95% confidence interval for the fixed effect. In Value Experiment: $\Delta$Value: difference in value between the two items (Value$_{right}$– Value$_{left}$); RT: reaction time; $\Sigma$Value: summed value of both items; $\Delta$DT: difference in dwell time (DT$_{right}$– DT$_{left}$), excluding the last two fixations; Conf: confidence. In Perceptual Experiment: $\Delta$Dots: difference in dots between the two circles (Dots$_{right}$– Dots$_{left}$); $\Sigma$Dots: summed number of dots between both circles. ***p<0.001, **p<0.01, *p<0.05.

## Appendix 3

### Fixation analysis

In the main text, we reported the analysis of last fixation and how its allocation to the (chosen) goal-relevant alternative is modulated by value/number of dots. This result confirmed the findings in *Krajbich et al., 2010* and expanded them to *dislike* frame and the perceptual realm. To give a more complete view of the fixations properties, we additionally performed a similar analysis to *Krajbich et al., 2010* for first and middle fixations.

It is important to notice that in our Value and Perceptual Experiments, at the beginning of each trial participants do not visualise the options since the presentation is gaze contingent. Therefore, an initial exploration is required to identify the alternatives involved in the decision. In *Krajbich et al., 2010* both options are visible from the beginning of the trial, however, participants' initial fixation is still randomly allocated.

For the analysis of middle fixations, if blank fixations were recorded between fixations to the same item, then those fixations were assigned to that item (e.g. 'Right', 'Blank', 'Right' was considered as 'Right', 'Right', 'Right'). Trials without middle fixations (i.e. only a first and a last fixation) were removed from the analysis. Trials with no item fixations for more than 40 ms at the beginning of the trial were also removed. In the following figures, results from *Krajbich et al., 2010* are presented together with our findings, as a reference.

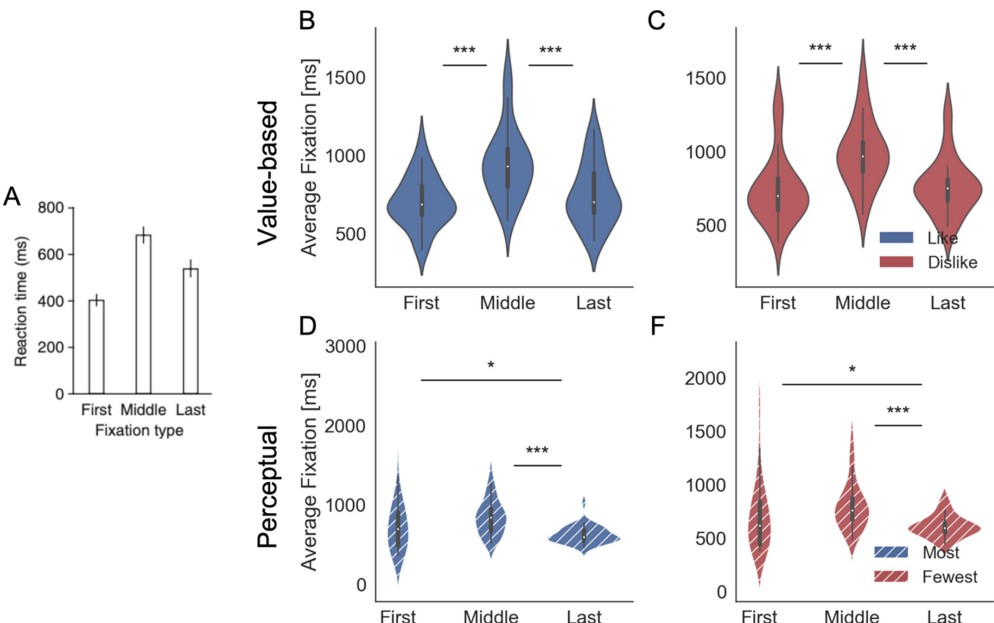

**Appendix 3—figure 1.** Fixation duration by type. Middle fixations indicate any fixations that were not the first or last fixations of the trial. (**A**) In *Krajbich et al., 2010* middle fixations were found to be longer than first and last fixations on average. For our Value Experiment, in *like* (**B**) and *dislike* (**C**) frames, and Perceptual Experiment, in *most* (**D**) and *fewest* (**F**) frames, the same pattern emerges with middle usually longer that first and last fixations. Violin plots depict the distribution of participant's average fixation time. Panel A reproduced from *Krajbich et al., 2010*. ***p<0.001, **p<0.01, *p<0.05.

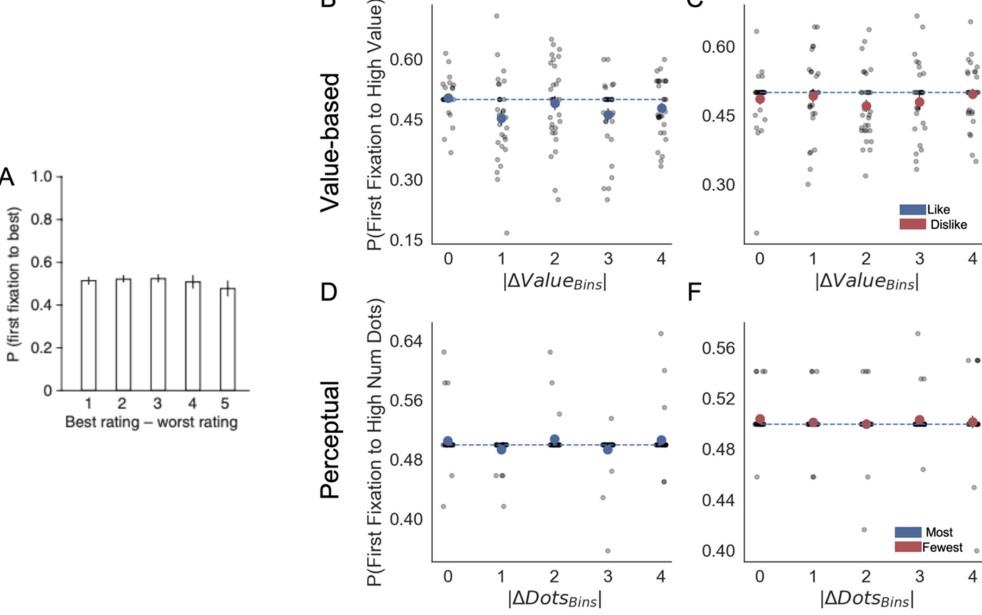

**Appendix 3—figure 2.** Fixation properties: probability that the first fixation is to the best item. (**A**) *Krajbich et al., 2010* reported that the probability is not significantly different from 50%, unaffected by the difference in ratings or difficulty (in our experiments difficulty is equivalent to the absolute difference item value, |ΔValue|, and absolute difference in number of dots, |ΔDots|). A similar pattern emerges in our Value Experiment, for *like* (**B**) and *dislike* (**C**) frames, and Perceptual Experiment, for *most* (**D**) and *fewest* (**F**) frames. Participant responses did not diverge from chance. Importantly, while in *Krajbich et al., 2010* participants can see both alternatives from the beginning of the trial, our presentation was gaze contingent. Segmented blue line indicates chance level. Light grey dots correspond to individual participants' probability of first fixation to high value/number of dots alternatives for each bin. Red or blue circles indicate the average for that bin considering all the participants. Panel A reproduced from *Krajbich et al., 2010*.

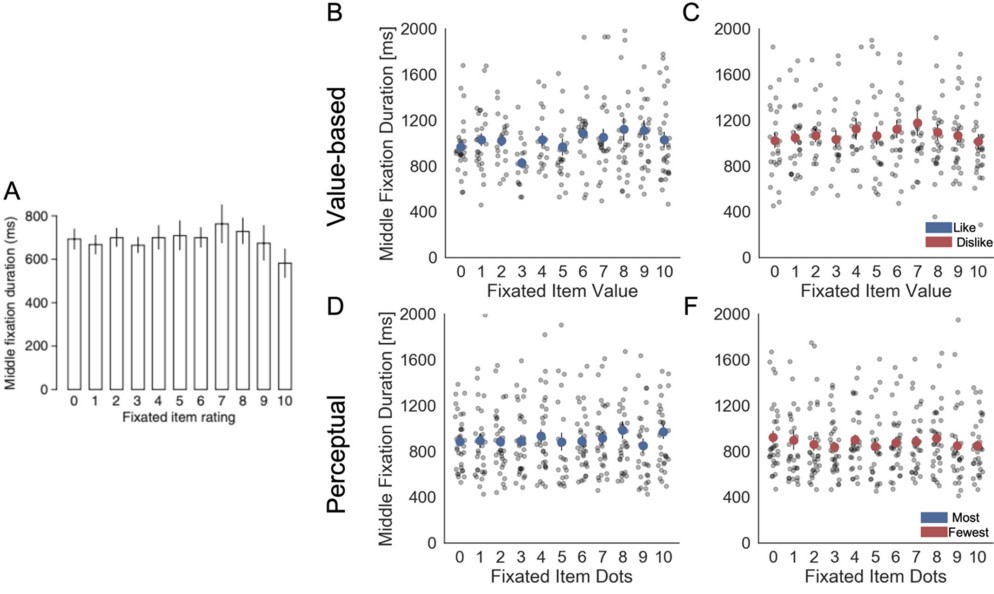

*Appendix 3—figure 3 continued on next page*

Appendix 3—figure 3 continued

**Appendix 3—figure 3.** Fixation properties: middle fixation duration as a function of the rating (value or number of dots) of the fixated item. (**A**) *Krajbich et al., 2010* reported that middle fixations durations were independent of the value of the fixated items. In Value Experiment, we found that middle fixation duration was independent of the value of the fixated item in *like* frame (**B**); however, a slight yet significant effect in *dislike* (**C**) frame was found (hierarchical linear regression estimate: $\beta_{Dislike} = 0.025$, $t(27.35) = 3.441$, $p<0.001$). In the Perceptual Experiment, for the *most* (**D**) frame we found a significant effect of fixated value ($\beta_{Most} = 0.017$, $t(29.51) = 3.013$, $p<0.01$), but not for *fewest* (**F**) frame. Light grey dots correspond to individual participants' middle fixation durations for each bin. Red or blue circles indicate the average for that bin considering all the participants. For the hierarchical linear regression, z-scored data at participant levels was used. Panel A reproduced from *Krajbich et al., 2010*.

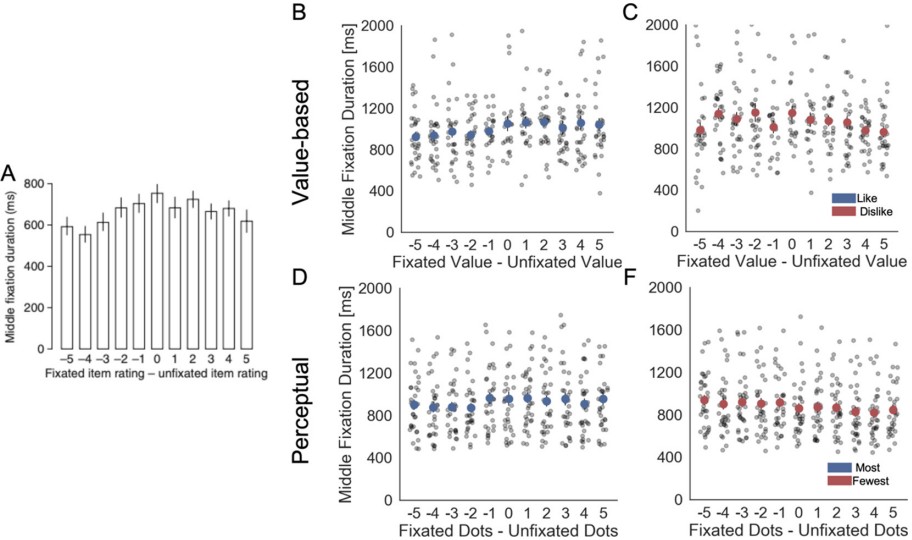

**Appendix 3—figure 4.** Fixation properties: middle fixation duration as a function of the difference in ratings (value or number of dots) between the fixated and unfixated items. (**A**) *Krajbich et al., 2010* reported a slight but significant dependency of middle fixations durations on the difference in value between items. In our Value Experiment, we found that in *like* (**B**) and *dislike* (**C**) this relationship was significant (hierarchical linear regression estimate: $\beta_{Like} = 0.015$, $t(28.22) = 2.192$, $p<0.05$; $\beta_{Dislike} = -0.027$, $t(28.22) = -4.415$, $p<0.001$). Similarly, in the Perceptual Experiment, *most* (**D**) and *fewest* (**F**) frames, the dependence was found also significant ($\beta_{Most} = 0.01$, $t(29.51) = 2.663$, $p<0.01$; $\beta_{Few} = -0.027$, $t(29.51) = -6.330$, $p<0.001$). Interestingly, a positive sign of the effect in *like* and *most* frames indicates that middle fixations tend to be longer for the option with the higher value or number of dots. On the other hand, the negative sign of the effect indicates that middle fixations would be longer for the option with lower value or number of dots in *dislike* and *fewest* frames. Light grey dots correspond to individual participants' middle fixation durations for each bin. Full red or blue circles indicate participant's average. Data are binned across participants for visualisation. All the factors and the predicted variable in the hierchical regression were z-scored at participant level. Panel A reproduced from *Krajbich et al., 2010*.

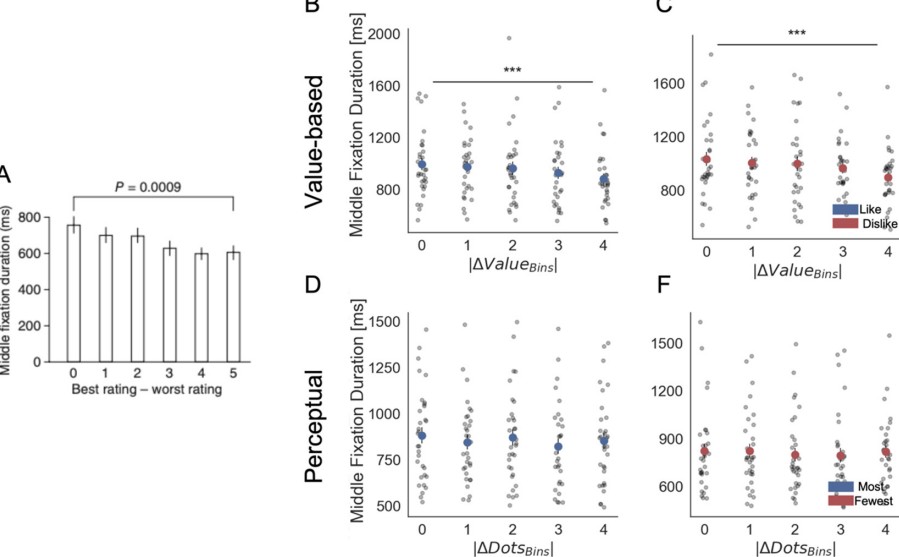

**Appendix 3—figure 5.** Fixation properties: middle fixation duration as a function of the difference in ratings between the best- and worst-rated items (difficulty of the trial). In our experiments, |Δ Value| and |ΔDots| represent the difficulty of the trials. (**A**) *Krajbich et al., 2010* reported a dependency of middle fixations durations on difficulty, with longer fixations in more difficult decisions. In our Value Experiment, in *like* (**B**) and *dislike* (**C**) frames a similar pattern was found: longer middle fixations for more difficult (lower |ΔValue|) trials (hierarchical linear regression estimate: $\beta_{Like}$ = -0.029, t(28.22) = -2.262, p<0.05; $\beta_{Dislike}$ = -0.047, t(28.22) = -4.415, p<0.001). The same relationship was found only in the *most* frame (**D**) but no in the *fewest* frame (**F**) in the Perceptual Experiment ($\beta_{Most}$ = -0.037, t(29.51) = -3.985, p<0.001; $\beta_{Few}$ = -0.024, t(29.51) = -1.623, p=0.10). Light grey dots correspond to individual participants' middle fixation durations for each bin. Full red or blue circles indicate participant's average. Data are binned across participants for visualisation. All the factors and the predicted variable in the hierchical regression were z-scored at participant level. Panel A reproduced from *Krajbich et al., 2010*. Tests presented here are based on a paired two-sided t-test between the first and last bin.***p<0.001, **p<0.01, *p<0.05.

## Appendix 4

## Confidence regression models

**Appendix 4—table 1.** Hierarchical linear models for confidence.

| Models | Formulas |
| --- | --- |
| Model 1 | Confidence ~ \|ΔValue\| |
| Model 2 | Confidence ~ \|ΔValue\| + RT |
| Model 3 | Confidence ~ \|ΔValue\| + RT + GSF |
| Model 4 | Confidence ~ \|ΔValue\| + RT + GSF + ΣValue |
| Model 5 | Confidence ~ \|ΔValue\| + RT + GSF + ΣValue + ΔDT |

In Value Experiment: |ΔValue|: absolute difference in value; RT: reaction time; ΣValue: summed value; ΔDT: difference in dwell time; GSF: gaze shift frequency. In Perceptual Experiment similar models were compared, but replacing ΔValue for ΔDots and ΣValue for ΣDots.

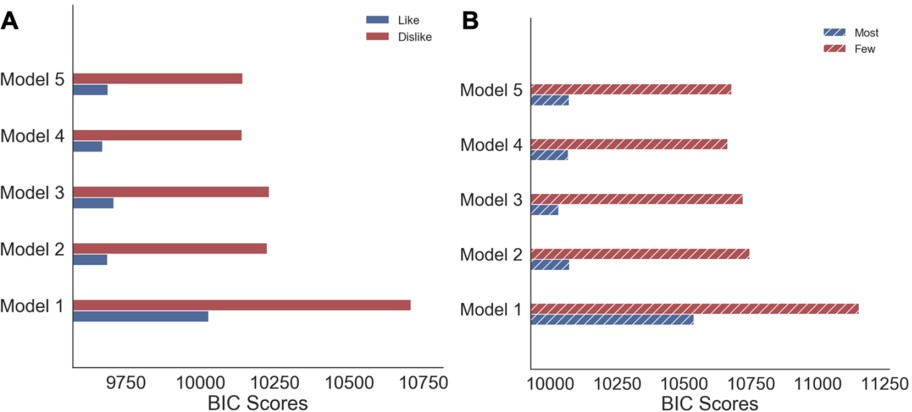

**Appendix 4—figure 1.** Model comparison of hierarchical linear regressions for confidence. (**A**) Value and (**B**) Perceptual Experiments. Solid colour indicates the value-based experiment and striped colours indicate the perceptual experiment.

**Appendix 4—table 2.** Statistical results for the hierarchical linear models for confidence in Value Experiment.

Z-values for the regression coefficients and their statistical significance are presented for the two frames. Repeated samples t-tests between the participants' regression coefficients in like and dislike frames were calculated.

| | Confidence value experiment | | | | | |
| --- | --- | --- | --- | --- | --- | --- |
| | Like | | Dislike | | Like - Dislike | |
| | Z | P | Z | P | T | P |
| \|ΔValue\| | 5.465 | <0.001 | 6.3 | <0.001 | −4.72 | <0.01 |
| RT | −6.373 | <0.001 | −7.739 | <0.001 | ns | |
| GSF | −2.365 | <0.05 | −2.589 | <0.05 | ns | |
| ΣValue | 3.206 | <0.001 | −4.492 | <0.001 | 9.91 | <0.001 |

**Appendix 4—table 3.** Statistical results for the hierarchical linear models for confidence in Perceptual Experiment.

Z-values for the regression coefficients and their statistical significance are presented for the two frames. Repeated samples t-tests between the participants' regression coefficients in most and fewest frames were calculated.

| | Confidence perceptual experiment | | | | | |
| | Most | | Fewest | | Most - Fewest | |
| | Z | P | Z | P | T | P |
|---|---|---|---|---|---|---|
| \|ΔValue\| | 3.546 | <0.001 | 7.571 | <0.001 | −4.554 | <0.001 |
| RT | −7.599 | <0.001 | −5.51 | <0.001 | ns | |
| GSF | −4.354 | <0.001 | −5.204 | <0.001 | ns | |
| ΣDots | 2.061 | <0.05 | −7.135 | <0.001 | 14.621 | <0.001 |

## Appendix 5

## GLAM – model comparison and out-of-sample simulations

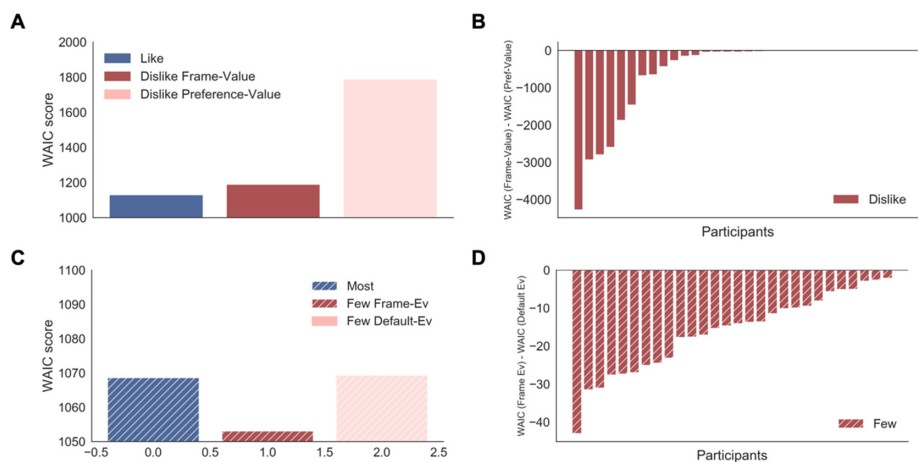

**Appendix 5—figure 1.** GLAM model comparison. (**A**) Average WAIC scores for *like* and *dislike* GLAM models fitted at individual level. In the *dislike* frame, two possible models are compared: preference-value, value reported in the BDM bid was used directly to fit the data; and frame-value, value was adjusted to comply with the frame modification (see Methods for more details). The model accounting for goal-relevant evidence in the *dislike* frame had a better fit. (**B**) Individual WAIC differences between *dislike* models fitted with frame-value and preference-value. Negative differences indicate best fits for the frame-value in all the participants. (**C**) Average WAIC scores for *most* and *fewest* GLAM models fitted at individual level. In the *fewest* frame, two possible models are compared: default-evidence, the number of dots was used directly to fit the data, and frame-evidence, evidence was adjusted to comply with the frame modification (i.e. the opposite of the number of dots was used as evidence). (**D**) Individual WAIC differences between *fewest* models fitted with frame-evidence and default-evidence. Negative differences indicate best fits for the frame-evidence in all the participants. Solid colour indicates the value-based experiment and striped colours indicate the perceptual experiment.

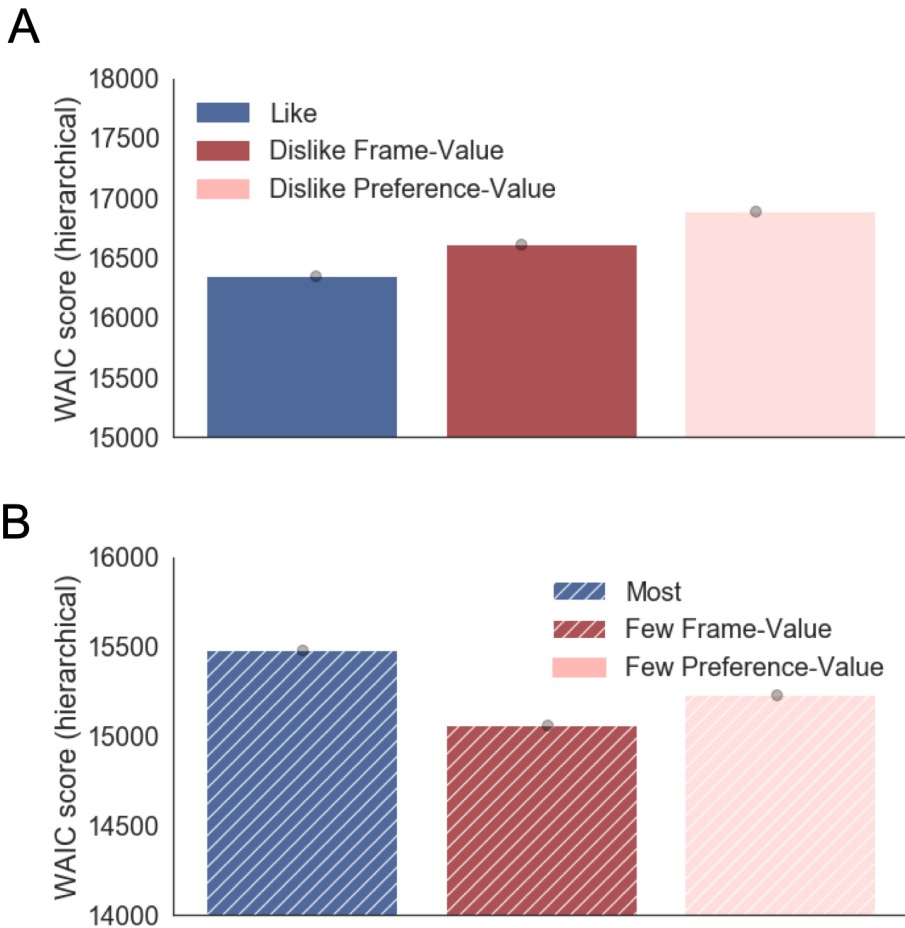

**Appendix 5—figure 2.** Hierarchical GLAM model comparison. (**A**) Value Experiment. WAIC scores for *like* and *dislike* GLAM models fitted hierarchically. In the *dislike* frame, two possible models are compared: preference-value, input values corresponding to the preferences reported at the beginning of the experiment (BDM bid); and frame-value, in which value was adjusted to comply with the frame modification (see Methods for more details). In *dislike* frame, the model accounting for goal-relevant resulted the most parsimonious of the two. (**B**) Perceptual Experiment. WAIC scores for *most* and *fewest* GLAM models fitted hierarchically. In the *fewest* frame, two possible models are compared: default-evidence, the number of dots was used directly to fit the data, and frame-evidence, evidence was adjusted to comply with the frame modification (i.e. the opposite of the number of dots was used as evidence).

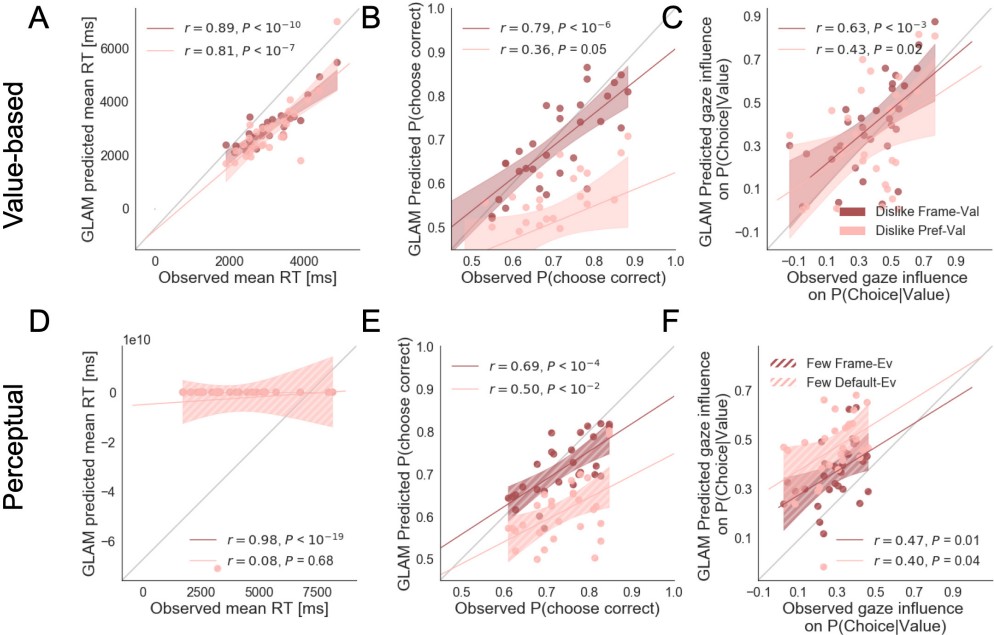

**Appendix 5—figure 3.** Individual out-of-sample prediction from the GLAM model for behavioural measures in Value (*dislike*) (**A–C**) and Perceptual (*fewest*) Experiments (**D–F**). In the *dislike* frame, two models are used to generate simulations: preference-value, value reported in the BDM bid was used directly to fit GLAM model; and frame-value, the values were adjusted to comply with the frame modification. The model predicts participants mean reaction time (RT) (**A**), probability of choosing the best item (i.e. item with lower value) (**B**) and the influence of gaze in choice probability (C, check Results section for more details on gaze influence measure). The frame-value model correlates better with the observed data. In the Perceptual Experiment, *fewest* frame, also two possible models are used to generate simulations: default-evidence, the number of dots was used directly to fit the data, and frame-evidence, the evidence was adjusted to comply with the context modification (i.e. opposite of the number of dots). We show the correlation between the data and simulations for RT (**D**), the probability of choosing the best alternative (i.e. alternative with fewer dots) (**E**) and gaze influence (**F**). In this case, frame-evidence model also predicts the behaviour in the *fewest* frame better. The results corresponding to the models using frame-evidence are presented in red and the models using default-evidence in pink. Dots depict the average of predicted and observed measures for each participant. Lines depict the slope of the correlation between observations and the predictions. The shadowed region presents the 95% confidence intervals, with full colour representing Value Experiment and striped colour the Perceptual Experiment. Model predictions are simulated using parameters estimated from individual fits for even-numbered trials.

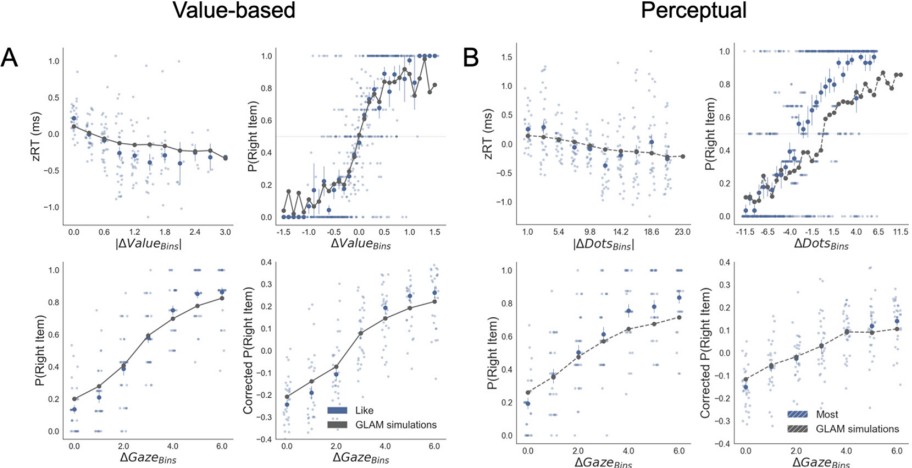

**Appendix 5—figure 4.** Replication of behavioural effect of interest by simulations using the GLAM fitted for *like* (**A**) and *most* frames (**B**). The four panels present four relevant behavioural relationships found in the data: (top left) faster responses (shorter RT) when the choice is easier (i.e. easier choices are found with higher |ΔValue| in value-based and higher |ΔDots| in perceptual); (top right) probability of choosing the right alternative increases when the difference in evidence (value or number of dots) is higher in the alternative at the right side of the screen (ΔValue and ΔDots are calculated considering right minus left options); (bottom left) the probability of choosing an alternative depends on the gaze difference; and (bottom right) the gaze influence on choice depending on the difference in gaze time between both alternatives. Solid blue dots depict the mean of the data across participants in *like* and *most* frames. Light blue dots present the mean value for each participant. In the Value Experiment, the solid grey lines show the average for model simulations. In the Perceptual Experiment, segmented grey lines show the model simulations. Data are binned for visualisation.

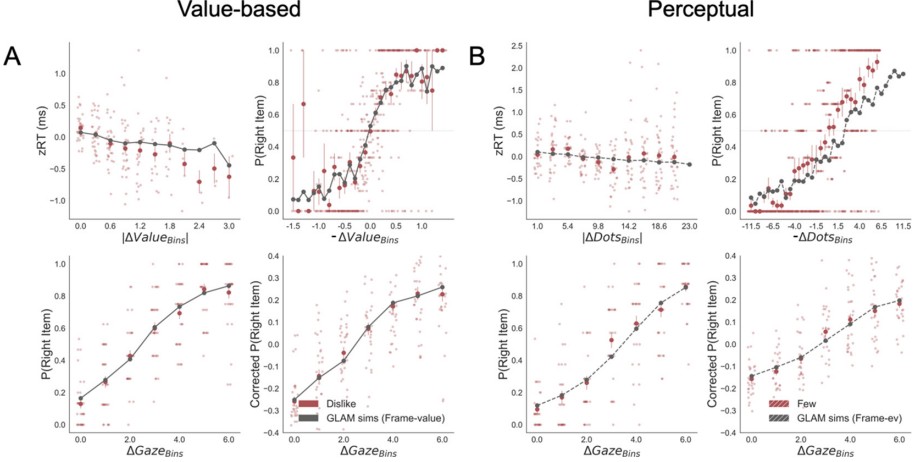

**Appendix 5—figure 5.** Replication of behavioural effect of interest by simulations using the GLAM fitted for *dislike* (**A**) and *fewest* frames (**B**). Frame-relevant evidence was used to fit the model. The four panels present four relevant behavioural relationships found in the data. Top left: faster responses (shorter RT) when the choice is easier (i.e. easier choices are found with higher |ΔValue| in value-based and higher |ΔDots| in perceptual). Top right: probability of choosing the right alternative increases when the difference in evidence (value or number of dots) is lower in the alternative at the right side of the screen (notice that -ΔValue and -ΔDots are calculated considering

*Appendix 5—figure 5 continued on next page*

left minus right options). Bottom left: the probability of choosing the right alternative depends on the gaze difference favouring the right option. Bottom right: the gaze influence on choice depending on the difference in gaze time between both alternatives. Solid red dots depict the mean of the data across participants in *dislike* and *fewest* frames. Light red dots present the mean value for each participant. In the Value Experiment, the solid grey lines show the average for model simulations. In the Perceptual Experiment, segmented grey lines show the model simulations. Data are binned for visualisation.

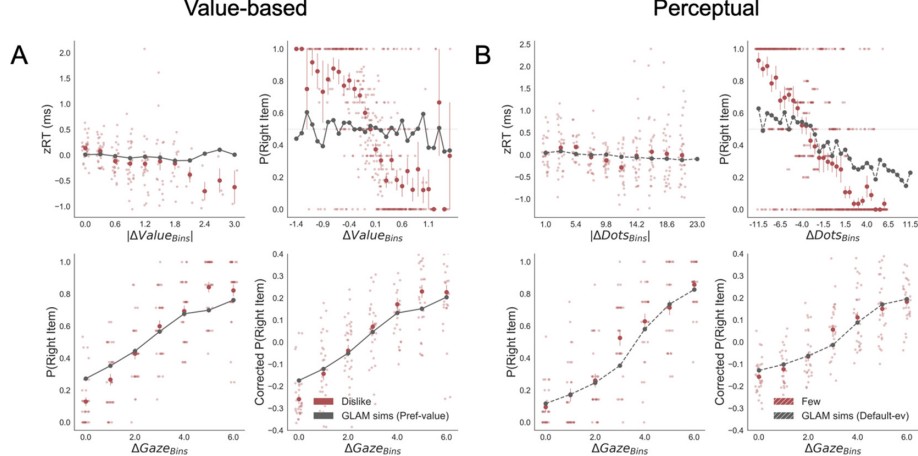

**Appendix 5—figure 6.** Replication of behavioural effect of interest by simulations using the GLAM fitted for *dislike* (**A**) and *fewest* frames (**B**). In this case, the models were fitted without adapting the values and dot numbers to the evidence that was relevant for the particular frame, that is the preference value and the default number of dots were used to fit the model in the *dislike* and *fewest* frame, respectively. The four panels present four relevant behavioural relationships found in the data. Top left: faster responses (shorter RT) when the choice is easier (i.e. easier choices are found with higher |ΔValue| in value-based and higher |ΔDots| in perceptual). Top right: probability of choosing the right alternative increases when the difference in evidence (value or number of dots) is lower in the alternative at the right side of the screen (ΔValue and ΔDots are calculated consider right minus left options). Bottom left: the probability of choosing the right alternative depends on the gaze difference favouring the right option. Bottom right: the gaze influence on choice depending on the difference in gaze time between both alternatives. No replication of the behavioural effect was found in this case for the relationship between RT -|ΔValue| and RT -|ΔDots| in *dislike* and *fewest* frames, respectively. Also P(right item)-ΔValue and P(right item)-ΔDots relationship was not replicated in *dislike* and *fewest* frames, respectively. Gaze effect seem to still keep its relationship, since gaze allocation time was not modified to account for the frame shift. Solid red dots depict the mean of the data across participants in *dislike* and *fewest* frames. Light red dots present the mean value for each participant. In the Value Experiment, the solid grey lines show the average for model simulations. In the Perceptual Experiment, segmented grey lines show the model simulations. Data are binned for visualisation.

# Appendix 6

## GLAM – parameter comparison

The results from the regression models presented in the *Results* section show that the nature of evidence integrated during the accumulation process depends on the frame in which participants make their choices. The Gaze-weighted Linear Accumulator Model (GLAM) predicts well participants' behaviour once frame-relevant evidence is employed to fit the model. Here we show the parameters obtained from this process. Four free parameters are fitted in GLAM: $\nu$ (drift term), $\gamma$ (gaze bias), $\tau$ (evidence scaling), and $\sigma$ (normally distributed noise standard deviation) (*Thomas et al., 2019*). For Value and Perceptual Experiments, we fitted the model in both frames and in each participant separately. The parameters were fitted using the even-numbered trials and in both studies the model fit was estimated using the WAIC score (used with Bayesian Models) (*Appendix 5—figure 1*).

### Value experiment

To explore variations in the process of accumulation of evidence characterised by GLAM, we compared the parameters obtained from the individual fit in *like* and *dislike* frames (*Appendix 6—figure 1A*). No significant variation between frames was found for the gaze bias (Mean $\gamma_{Like}$ = -0.14, Mean $\gamma_{Dislike}$ = 0.03, $\Delta\gamma_{Like-Dislike}$ = -0.17, t(30) = -1.66; p=0.11, ns), the scaling parameter ($\tau_{Like}$ = 2.81, $\tau_{Dislike}$ = 2.69, $\Delta\tau_{Like-Dislike}$ = 0.115, t(30) = 0.313; p=0.75, ns), and the noise term (Mean $\sigma_{Like}$ = 0.0075, Mean $\sigma_{Dislike}$ = 0.0074, $\Delta\sigma_{Like-Dislike}$ = 0.00012, t(30) = 0.342; p=0.734, ns). We observed a significantly higher value of the drift term, $\nu$, during the *like* frame ($\nu_{Like}$ = 5.60x10$^{-5}$, $\nu_{Dislike}$ = 4.53x10$^{-5}$, $\Delta\nu_{Like-Dislike}$ = 1.06 x 10$^{-5}$, t(30) = 3.44; p<0.01). This means that evidence is accumulated faster during the *like* frame, which gives us an insight into the differences in the evidence accumulation product of the change frame modification.

### Perceptual experiment

We also compared the parameters obtained from GLAM individual fit in the perceptual experiment (*Appendix 6—figure 1B*). No significant variation between frames was found for the scaling parameter ($\tau_{Most}$ = 0.34, $\tau_{Few}$ = 0.13, $\Delta\tau_{Most-Few}$ = 0.212, t(27) = 1.43; p=0.16, ns) or the drift term (Mean $\nu_{Most}$ = 3.8x10$^{-5}$, Mean $\nu_{Few}$ = 3.99x10$^{-5}$, $\Delta\nu_{Most-Few}$ = -1.92x10$^{-6}$, t(27) = -0.465; p=0.645, ns). The gaze bias is larger during the *fewest* frame ($\gamma_{Most}$ = 0.48, $\gamma_{Few}$ = 0.26, $\Delta\gamma_{Most-Few}$ = 0.22, t(27) = 2.61; p<0.05). The $\sigma$ parameter is also significantly different depending on the frame, with higher noise in the *most* frame ($\sigma_{Most}$ = 0.0073, $\sigma_{Few}$ = 0.0066, $\Delta\sigma_{Most-Few}$ = 0.0007, t(27) = 2.26; p<0.05). In summary, the accumulation process seems to be noisier and less affected by visual attention in the *most* frame. In both frames, the finding that $\gamma$ < 1 indicates that gaze modulates the accumulation of evidence.

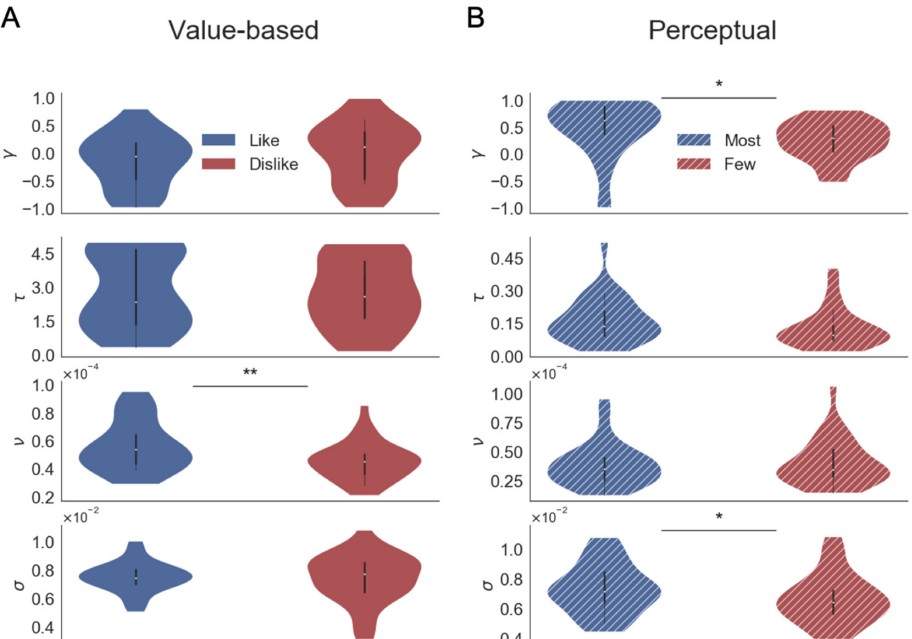

**Appendix 6—figure 1.** Parameters fitted at subject level using GLAM in Value (**A**) and Perceptual (**B**) Experiments. The free parameters are γ (gaze bias), τ (evidence scaling), ν (drift term), and σ (standard deviation of the normally distributed noise). In the Value Experiment, we found a significant decrease in the drift term during the *dislike* frame, maybe indicating a more uncertain decision process. The parameters in Perceptual Experiment were significantly different for gaze bias and noise term, with higher γ and σ values in the *most* frame. This may indicate a reduced effect of gaze on choice during the *most* frame and slightly less noisier accumulation process in the *fewest* frame. In each experiment, the GLAM parameters were fitted independently for each frame. In the violin plot, red and blue areas indicate the distribution of the parameters across participants. Black bars present the 25, 50, and 75 percentiles of the data. Solid colour indicates the Value Experiment and striped colours indicate the Perceptual Experiment.

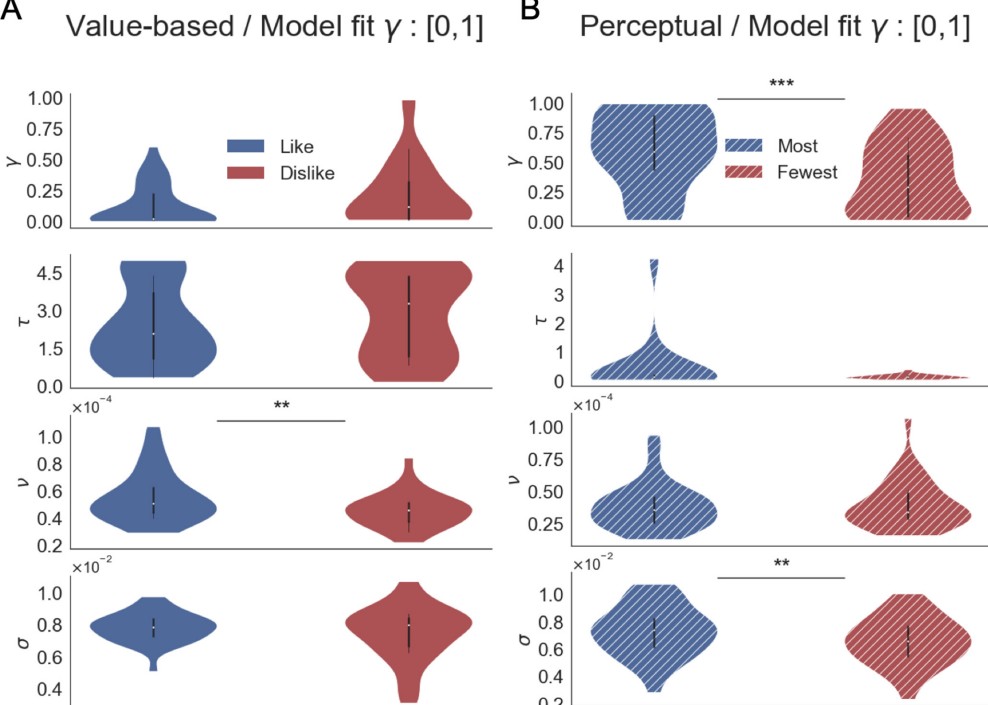

**Appendix 6—figure 2.** GLAM model parameters when the model fit is performed constraining γ to [0,1] range. *Thomas et al., 2019* describes a 'leakage' of evidence when γ < 0, which can be a conflicting assumption in this type of models. We corroborated that the differences between the parameters in *like/dislike* and *most/fewest* remain the same in comparison to the fit reported constraining γ to [−1,1].

## Appendix 7

### Attentional drift diffusion model

The attentional Drift Diffusion Model (aDDM) has been extensively used in literature to characterise the effect of attention over choice (*Krajbich et al., 2010*). Unlike GLAM, aDDM considers the dynamics of fixations during trials to fit the model. To further support our idea that goal-relevant evidence is accumulated, we fitted both Value and Perceptual datasets with the aDDM model, as implemented by *Tavares et al., 2017* (aDDM toolbox, https://github.com/goptavares/aDDM-Toolbox).

The aDDM model assumes that evidence is accumulated dynamically in a variable called the relative decision value (RDV) signal. RDV starts at 0 and it evolves over time, accumulating evidence until a barrier is reached (+1 or −1) which will define the alternative to be selected (right or left). Every time step, RDV changes according to $\mu\Delta t + \varepsilon t$, with $\mu$ the deterministic change (slope term) and $\varepsilon$ the Gaussian noise term. The fixation to the two alternatives will define the value of $\mu$: when the left option is fixated $\mu = d(r_{left} - \theta r_{right})$ and $\mu = d(r_{right} - \theta r_{left})$ for the right option. Therefore, the aDDM model considers three free parameters: d, $\sigma$, and $\theta$. The parameter d is a positive constant characterising the speed of integration; $\sigma$ is the standard deviation for a zero-mean Gaussian distribution for noise, and $\theta$ is the attentional parameter that controls the size of the attentional bias (range between 0 and 1). If $\theta = 1$, the model is reduced to a standard drift-diffusion model (DDM) without attentional bias.

### Group model fitting

The models were fitted to choice and RT data independently for *like* and *dislike* frames in our Value Experiment and for *most* and *fewest* frames in the Perceptual Experiment. The odd trials of the pooled data from 31 participants in value-based data and 32 participants for perceptual case was used to fit the models. The model considers the available evidence (item value and number of dots) and the sequence of fixations for each trial. As in GLAM, we fitted the parameters in *dislike* and *fewest* frames considering a version of the input values/evidence that accounted for the change in the objective of the task (i.e. reporting item not preferred or the alternatives with fewer dots, respectively). To compare, we also fitted another model using the evidence 'by default' (i.e. BDM bid values or number of dots in the circles). To account for the different ranges of item valuation used by the participants we normalised the value reports by binning at a participant level. In the Value Experiment, the data were separated in six bins using quantiles-based discretisation. In the Perceptual Experiment, given the distribution of the evidence (i.e three numerosity levels and smaller dots differences between two alternatives), we separated the dots data in eight bins. The maximum likelihood estimation (MLE) procedure was carried in iterative steps searching over a grid with the three model parameters. Initial grid was set to [0.001, 0.005, 0.01] for d, [0.01, 0.05, 0.1] for $\sigma$ and [0.01, 0.5, 1] for $\theta$. The likelihood for choice and RT in odd-trials, conditional to the pattern of fixations, was calculated for each combination of parameters in the grid (check *Tavares et al., 2017* for the details of the algorithm to simulate aDDM trials). The time step used for the estimation of aDDM was 10 ms. The set of parameters with lower negative log-likelihood (NLL) was used as centre of the grid for the next iteration. Therefore, the grid to search in the next iteration (t+1) was defined as $[d_t - \Delta d_t/2, d_t, d_t + \Delta d_t/2]$, $[\theta_t - \Delta\theta_t/2, \theta_t, \theta_t + \Delta\theta_t/2]$, and $[\sigma_t - \Delta\sigma_t/2, \sigma_t, \sigma_t + \Delta\sigma_t/2]$, considering the respective constrains of each parameter value. The iterative process finished once the improvement in the MLE of the proposed parameter solution was smaller than 0.05% ($|minNNL_{t+1} - minNNL_t| < 0.0005*minNNL_t$). The convergence was reached after two iterations in our models. In our results *Appendix 7—table 1*), we found that for both, *dislike* and *fewest* conditions, the model fitted using goal-relevant evidence had better performance than the model using default value or number of dots, as indicated by a lower NLL value.

**Appendix 7—table 1.** aDDM model parameters.
Estimated parameters for Value and Perceptual Experiments. Parameter description - d: speed of integration; $\sigma$: standard deviation for the noise distribution, $\theta$: attentional bias. NNL: negative log-likelihood of the models indicating goodness-of-fit.

| | Value-based | | | Perceptual | | |
| --- | --- | --- | --- | --- | --- | --- |
| | Like | Dislike Preference-values | Dislike Frame-values | Most | Fewest Default-evidence | Fewest Frame-evidence |
| d | 0.001 | 0 | 0.001 | 0.001 | 0.001 | 0.001 |
| σ | 0.05 | 0.05 | 0.05 | 0.05 | 0.05 | 0.05 |
| θ | 0 | 0 | 0 | 0.255 | 0 | 0.01 |
| NLL | 12441.012* | 13342.297 | 12640.837* | 13948.411* | 14169.154 | 13826.983* |

*Indicates the model with lower NLL for that frame

## Out-of-sample group simulations

To test the capacity of the model to predict out-of- sample, the aDDM with the best fitted parameters using odd-numbered trials was used to predict the behaviour observed on the even-numbered trials. We generated 40,000 simulations for the Value Experiment and 48,000 trials for the Perceptual Experiment. Fixations, latencies and inter-fixations transitions were sampled from empirical distributions, obtained from the pooled even-numbered trials across participants following the procedure used by *Tavares et al., 2017*.

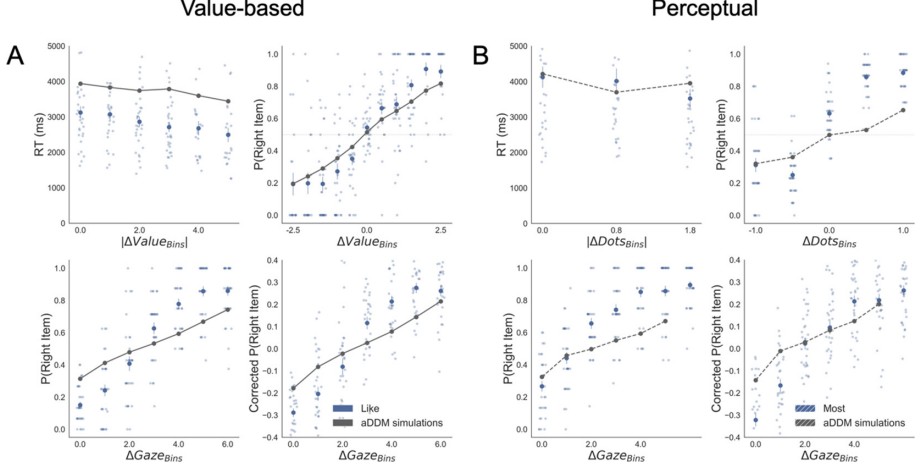

**Appendix 7—figure 1.** Replication of behavioural effects by aDDM simulations for *like* (**A**) and *most* frames (**B**). The four panels present four relevant behavioural relationships found in the data. Top left: faster responses (shorter reaction time, RT) when the choice is easier (i.e. easier choices are found with higher |ΔValue| and |ΔDots| in Value an Perceptual Experiments, respectively). Top right: probability of choosing the right alternative increases when the evidence towards the right item is higher (ΔValue and ΔDots are calculated considering right minus left options). Bottom left: the probability of choosing the item on the right side of the screen depends on the gaze time difference (ΔGaze, calculated as the time observing the right minus the left item). Bottom right: gaze influence on choice depending on the difference in ΔGaze (check Results section for more details on gaze influence). Solid blue dots depict the mean of the data across participants in *like* and *most* frames. Light blue dots show the mean value for each participant. In Value Experiment, the solid grey lines show the average for model simulations. In the Perceptual Experiment, segmented grey lines show the average for model simulations. Data and simulations were binned for visualisation.

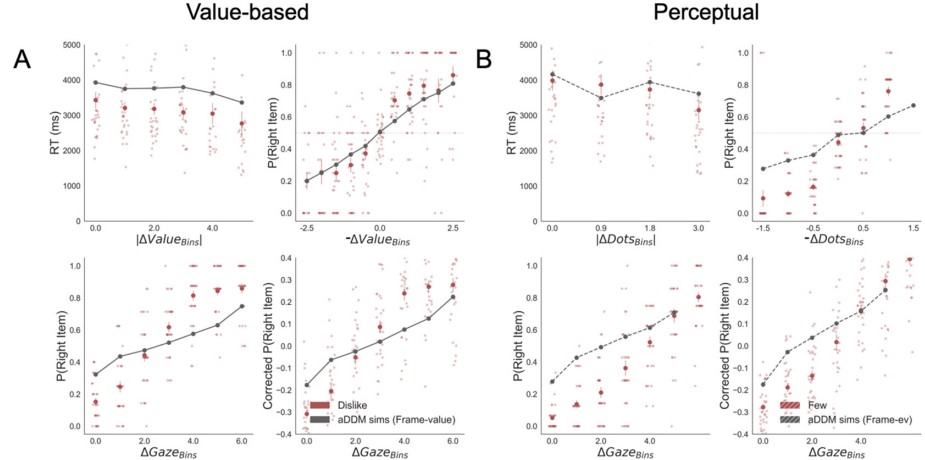

**Appendix 7—figure 2.** Replication of behavioural effects by aDDM simulations for *dislike* (**A**) and *fewest* (**B**) frames. Importantly, these models were fitted using goal-relevant evidence. The four panels present four relevant behavioural relationships found in the data. Top left: faster responses (shorter reaction time, RT) when the choice is easier (i.e. easier choices are found with higher |Δ Value| and |ΔDots| in Value and Perceptual Experiments, respectively). Top right: probability of choosing the right alternative increases when the evidence towards the left item is higher (-ΔValue and –ΔDots, that is, increment when left item is more valuable or has more dots than the right item). Bottom left: the probability of choosing the item on the right side of the screen depends on the gaze time difference (ΔGaze, calculated as the time observing the right minus the left item). Bottom right: gaze influence on choice depending on the difference in ΔGaze (check Results section for more details on gaze influence). Solid red dots depict the mean of the data across participants in *dislike* and *fewest* frames. Light red dots show the mean value for each participant. In Value Experiment, the solid grey lines show the average for model simulations. In the Perceptual Experiment, segmented grey lines show the average for model simulations. Data and simulations were binned for visualisation.

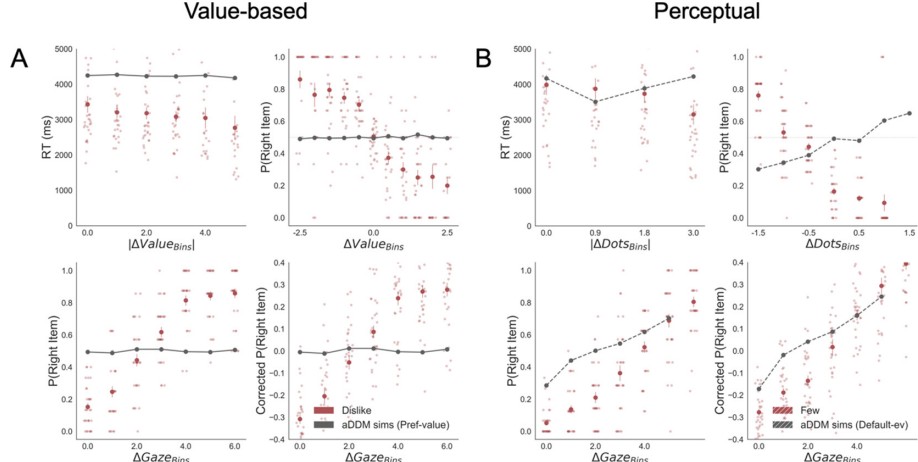

**Appendix 7—figure 3.** Replication of behavioural effects by aDDM simulations for *dislike* (**A**) and *fewest* frames (**B**). Importantly, these models were fitted using the default evidence in Value and Perceptual Experiments, that is, preference value and number of dots, respectively. Unlike the models fitted with goal-relevant evidence, these models do not capture reaction time (RT) and choice behaviour in dislike and fewest frames. The four panels present four relevant behavioural

*Appendix 7—figure 3 continued on next page*

*Appendix 7—figure 3 continued*

relationships found in the data. Top left: faster responses (shorter RT) when the choice is easier (i.e. easier choices are found with higher |ΔValue| and |ΔDots| in Value an Perceptual Experiments, respectively). Top right: probability of choosing the right alternative increases when the evidence towards the left item is higher (ΔValue and ΔDots are calculated considering right minus left options). Bottom left: the probability of choosing the item on the right side of the screen depends on the gaze time difference (ΔGaze, calculated as the time observing the right minus the left item). Bottom right: gaze influence on choice depending on the difference in ΔGaze (check Results section for more details on gaze influence). Solid red dots depict the mean of the data across participants in *dislike* and *fewest* frames. Light blue dots show the mean value for each participant. In Value Experiment, the solid grey lines show the average for model simulations. In the Perceptual Experiment, segmented grey lines show the average for model simulations. Data and simulations were binned for visualisation.

## Appendix 8

## GLAM – balance of evidence simulations

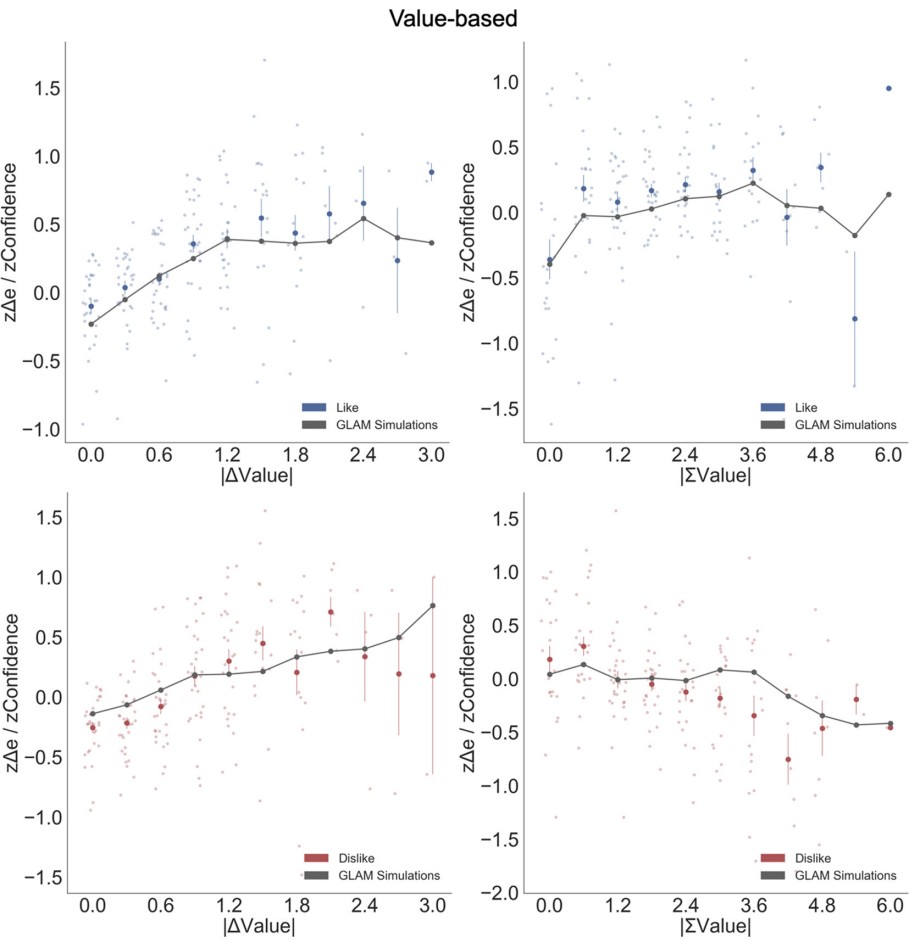

**Appendix 8—figure 1.** Balance of evidence simulations in the Value Experiment. The difference between accumulators (Δe) obtained from GLAM simulations matches participants' confidence. Top left: a higher value difference between the two items (|ΔValue|) increases confidence and simulated Δe. Top right: in the *like* frame, an increase in the summed value of the two alternatives (|ΣValue|) boosts confidence and simulated Δe. Bottom left: as in *like* frame, |ΔValue| boosted confidence and Δe in *dislike* frame. Bottom right: in the *dislike* frame, the effect of |ΣValue| over confidence flips: confidence and Δe decrease with higher values of the alternatives, accounting for the change in goal. Blue and red dots depict the (z-scored) confidence taken from participants in *like* and *dislike* frames (respectively). Grey line presents the model simulations for both separate frames. Data were segmented in 11 bins for ΔValue or ΣValue.

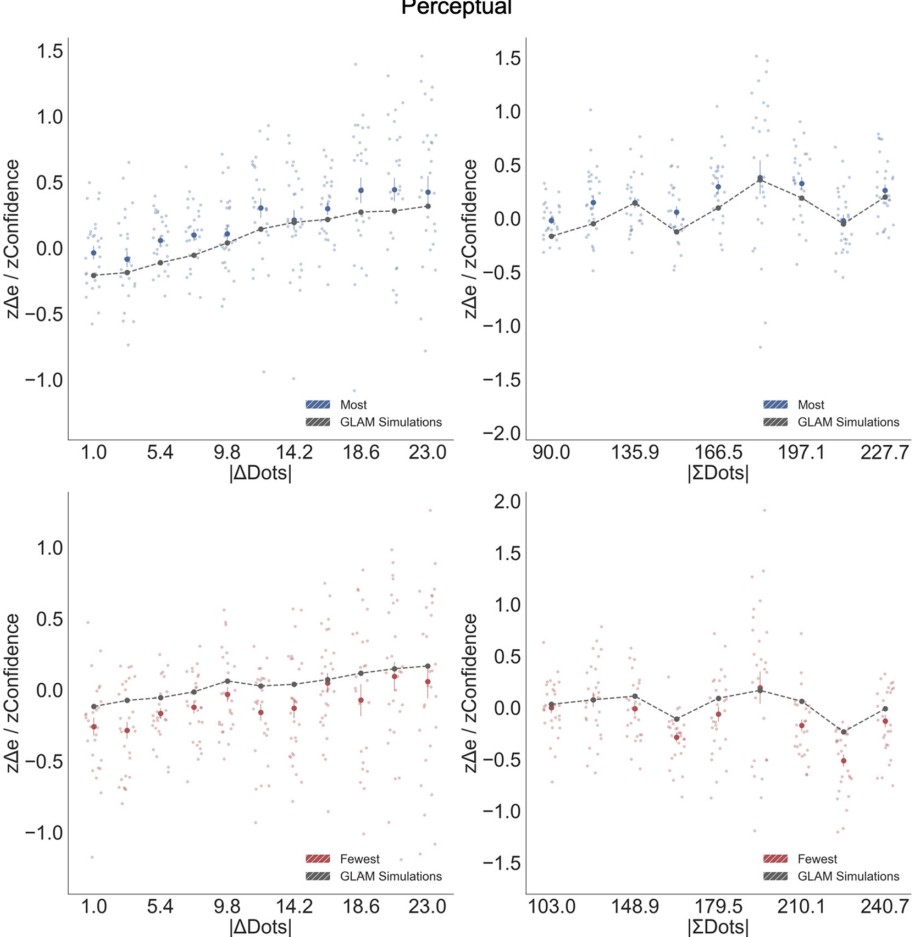

**Appendix 8—figure 2.** Balance of evidence simulations in the Perceptual Experiment. As in Value Experiment, the difference between accumulators (Δe) obtained from GLAM simulations matches participants' confidence. Top left: a higher difference in number of dots between the two circles (|ΔDots|) increases confidence and simulated Δe. Top right: in the *most* frame, an increase in the summed number of dots (|ΣDots|) boosts confidence and simulated Δe. Bottom left: as in *most* frame, |ΔDots| boosted confidence and Δe in *fewest* frame. Bottom right: in the *fewest* frame, the effect of |ΣDots| over confidence flips: confidence and Δe decrease with higher number of dots in both circles, accounting for the change in goal. Blue and red dots depict the (z-scored) confidence taken from participants in *most* and *fewest* frames (respectively). Grey line presents the model simulations for both separate frames. Data were segmented in 11 bins for ΔDots or ΣDots.

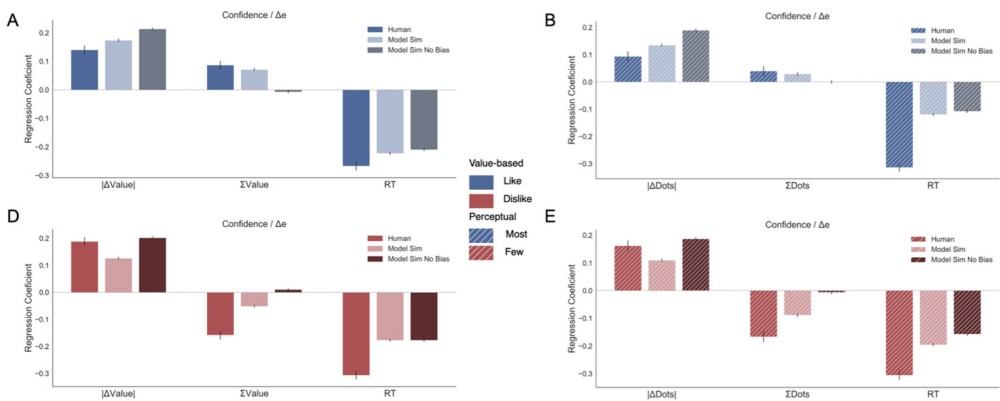

**Appendix 8—figure 3.** Pooled linear regressions to predict balance of evidence (Δe) simulations. Here the full model results for *Figure 6* (see Results section) are displayed. In Value Experiment, the full simulations of Δe replicated the pattern of results obtained in human data (confidence results), that is there is a flip in the sign of ΣValue effect over confidence between *like* (**A**) and *dislike* (**D**) frames. However, if the gaze asymmetry is removed, we found the effect of ΣValue over Δe disappears. The results in Perceptual Experiment, *most* (**B**) and *fewest* (**E**) frames, mirror the findings in the Value Experiment.

## Appendix 9

### Normative model – proof of propositions 1 and 2

All the uses of μ in this proof: $\mu_i$ is the mean of the belief on the value of item i after the agent has acquired one signal about item i; $\mu_i'$ is the mean of the belief on the value of item i after the agent has acquired two signals about item i.

We begin by proving Proposition 1. Recall that qualities $v_i$ are distributed independently according to a Normal distribution and that the agent knows it, thus holds a correct prior belief. Recall also that the agent has taken a sample, $x_i = v_i + \epsilon_i$, with $\epsilon_i$ independently and identically distributed with $\epsilon_i \sim N(0, \sigma_\epsilon^2)$. Because the prior belief is Normal, and because also the signal $x_i$ is Normally distributed around the true value, standard arguments give us that the posterior belief about $v_i$ is also Normal. Denote by $\mu_i$ and $\sigma_v^2$ the mean and the variance, respectively, of this posterior belief about $v_i$, for each i. Note that $\sigma_v^2$ is the same for all i (since, with Normal distributions, the variance of the posterior only depends on the variance of the prior and of the signal).

The agent can now acquire a second signal about only one of the items and needs to decide which item. Note that, after a second signal about item i is acquired, this will further change the belief about $v_i$. Denote by $\mu_i'$ the mean of this belief: that is, $\mu_i'$ is the mean of the belief about $v_i$ after the agent has acquired *two* signals about it.

Recall that $V(i)$ indicates the utility that the agent expects to have after acquiring the second signal about item i. Recall also that we denote by $i_1$ the item for which the agent has received the highest first signal, $i_2$ the second-highest, etc. Suppose first that the second signal acquired is not about the best item. Then, there are two possibilities. First, we may have that $\mu_{i_1} > \mu_i'$, that is, after the second signal, the posterior mean about the quality of i, $\mu_i'$, is below that of $i_1$, $\mu_{i_1}$. In that case the agent will choose $i_1$, and receive an expected quality of $\mu_{i_1}$. If instead $\mu_{i_1} < \mu_i'$, then the agent chooses i and has an expected quality $\mu_i'$. It follows that, for $i \neq i_1$, we have

$$V(i) = max\{\mu_{i_1}, \mu_i'\}.$$

For similar reasons, $V(i_1) = max\{\mu_{i_2}, \mu_{i_1}'\}$.

When the agent needs to decide which item to acquire a second signal about, however, the second signal has not been observed yet: we thus need to compute the expectation of $V(i)$. In order to compute this, the agent needs to form a belief about what will be the value of $\mu_i'$ *before* acquiring the second signal about $v_i$ (but after acquiring the first signal). Such belief must again be normally distributed, and have mean $\mu_i$. (This is because, of course, the expectation that the agent holds about the posterior mean before receiving the signal must be centered at the prior mean, which in this case is $\mu_i$.) Denote by $\theta$ the variance of this belief; again, this is the same for all *i*s. Thus, $\mu_i' \sim N(\mu_i, \theta)$.

We are now ready to prove the following claims.

## Claim 1. $E[V(i_1)] = E[V(i_2)]$.

**Proof.** Recall that, for $i \neq i_1$, we have $V(i) = max\{\mu_{i_1}, \mu_i'\}$ and $\mu_i' \sim N(\mu_i, \theta)$. This means that the belief about $V(i)$, for $i \neq i_1$, coincides with $N(\mu_i, \theta)$ for values above $\mu_{i_1}$, but has a mass point at $\mu_{i_1}$ equal to the probability that $N(\mu_i, \theta)$ is below $\mu_{i_1}$. If we denote by $f_\mu$ the Probability Density Function of $N(\mu, \theta)$, it follows that we have

$$E[V(i_2)] = \mu_{i_1} \int_{-\infty}^{\mu_{i_1}} f_{\mu_{i_2}}(x)dx + \int_{\mu_{i_1}}^{+\infty} x f_{\mu_{i_2}}(x)dx.$$

Recall also that $V(i_1) = max\{\mu_{i_2}, \mu_{i_1}'\}$. The belief about $V(i_1)$ coincides with $N(\mu_{i_1}, \theta)$ above $\mu_{i_2}$, but has a mass point at $\mu_{i_2}$ equal to the probability that $N(\mu_i, \theta)$ is below $\mu_{i_2}$. Then,

$$E[V(i_1)] = \int_{-\infty}^{\mu_{i_2}} \mu_{i_2} f_{\mu_{i_1}}(x)dx + \int_{\mu_{i_2}}^{+\infty} x f_{\mu_{i_1}}(x)dx.$$

Note that, by construction, we have

$$\mu_{i_1} = \int_{-\infty}^{\mu_{i_2}} x f_{\mu_{i_1}}(x)dx + \int_{\mu_{i_2}}^{+\infty} x f_{\mu_{i_1}}(x)dx.$$

It follows that

$$E[V(i_1)] - \mu_{i_1} = \int_{-\infty}^{\mu_{i_2}} (\mu_{i_2} - x) f_{\mu_{i_1}}(x)dx \qquad (A1)$$

and

$$E[V(i_2)] - \mu_{i_1} = \int_{\mu_{i_1}}^{+\infty} (x - \mu_{i_1}) f_{\mu_{i_2}}(x)dx. \qquad (A2)$$

But we also know that

$$\int_{-\infty}^{\mu_{i_2}} (\mu_{i_2} - x) f_{\mu_{i_1}}(x)dx = \int_{2\mu_{i_1} - \mu_{i_2}}^{+\infty} (x - 2\mu_{i_1} + \mu_{i_2}) f_{\mu_{i_1}}(x)dx = \int_{\mu_{i_1}}^{+\infty} (x - \mu_{i_1}) f_{\mu_{i_2}}(x)dx.$$

Together with *Equation A1 and A2*, this proves the claim.∎

## Claim 2. If N>2, $E[V(i_2)] > E[V(i_j)]$ for all $j>2$.

**Proof.** Recall that, for $i \neq i_1$, we have $V(i) = max\{\mu_{i_1}, \mu_i'\}$, where $\mu_i' \sim N(\mu_i, \theta)$. It follows that the beliefs about both $V(i_2)$ and $V(i_j)$ (held before the second signal is acquired) has support $[\mu_{i_1}, \infty)$. Denote by $F_i$ the Cumulative Density Function (CDF) of this belief. To prove the claim, we show that $F_{i_2}$ First Order Stochastically Dominates $F_{i_j}$ for all $j>2$, while the converse is not true: that is, we aim to show that for all $x$ in the support, $F_{i_2}(x) \leq F_{i_j}(x)$, strictly for some x. (Recall that a distribution F first order stochastically dominates another distribution G if for all x, the probability that F returns at least x is not below the probability that G returns x or more.) This implies $E[V(i_2)] > E[V(i_j)]$.

Let $\delta := \mu_{i_2} - \mu_{i_j}$ and note that we have $\delta > 0$ and that $N(\mu_{i_2}, \theta)(x + \delta) = N(\mu_{i_j}, \theta)(x)$ for all $x \in \mathbb{R}$. Since $V(i)$ coincides with $\mu_i'$ whenever that lies above $\mu_{i_1}$ and since $\mu_i' \sim N(\mu_i, \theta)$, it follows that, for all $x > \mu_{i_1}$, we have $(1 - F_{i_j}(x)) = (1 - F_{i_2}(x + \delta))$: the probability that $F_{i_j}$ assigns to $V(i_j)$ being x or higher is the same that $F_{i_2}$ assigns to $V(i_2)$ being $x + \delta$ or higher. Then, $F_{i_j}(x) = F_{i_2}(x + \delta)$. Because CDFs are increasing and $\delta > 0$, then $F_{i_2}(x) \leq F_{i_2}(x + \delta)$, thus $F_{i_2}(x) \leq F_{i_j}(x)$ for all $x > \mu_{i_1}$. Moreover, notice that we must have

$$F_{i_2}(\mu_{i_1}) = N(\mu_{i_2}, \theta)([-\infty, \mu_{i_1}]) < N(\mu_{i_j}, \theta)([-\infty, \mu_{i_1}]) = F_{i_j}(\mu_{i_1}).$$

That is, $F_{i_2}$ assigns to values below $\mu_{i_1}$ a lower probability than $F_{i_j}$ does. It follows that for all $x$ in the support $[\mu_{i_1}, \infty)$, we have $F_{i_2}(x) \leq F_{i_j}(x)$, strictly for some. Thus, $F_{i_2}$ First Order Stochastically Dominates $F_{i_j}$ for all $j>2$, while the converse is not true. The claim follows.

The two claims together prove Proposition 1.

## Proposition 2

The proof of Proposition 2 is identical once we replace $i_j$ by $i_{N+1-j}$ for $j = 1, \ldots, N$. Intuitively, the problem of maximising the expected utility of the remaining items is strategically equivalent to the problem of choosing the lowest item, which, in turn, is symmetric to the problem of choosing the best item. *QED*.

