## [Decision Letter]

**Acceptance summary:**

Your detailed investigation of the interplay between visual attention and value-based choice from the broader perspective of goal-directed behavior has provided clear findings that challenge popular decision-making models in neuroeconomics – according to which visual attention towards a choice option increases its estimated subjective value. Your finding that visual attention modulates the integration of goal-relevant evidence rather than value, obtained using state-of-the-art cognitive modeling, is relevant to a broad research community spanning across cognitive psychology and neuroeconomics. Congratulations for an insightful article.

**Decision letter after peer review:**

Thank you for submitting your article "Visual attention modulates the integration of goal-relevant evidence and not value" for consideration by *eLife*. Your article has been reviewed by two peer reviewers, and the evaluation has been overseen by a Reviewing Editor and Christian Büchel as the Senior Editor. The following individuals involved in review of your submission have agreed to reveal their identity: Alexandre L S Filipowicz (Reviewer #2); Sebastian Gluth (Reviewer #3).

The reviewers have discussed the reviews with one another and the Reviewing Editor has drafted this decision to help you prepare a revised submission.

This manuscript describes a behavioral modeling study which aims at challenging popular decision-making models in neuroeconomics – including the “attentional drift diffusion model (aDDM)” – according to which attention towards a choice option increases its estimated subjective value (and thus its probability of being chosen). By asking participants to either choose their most preferred option (“like” frame) but also their least preferred option (“dislike” frame), the authors report that human subjects are more likely to unselect the option that they fixate more in the “dislike” frame – a finding not predicted by the aDDM. The authors report a similar framing effect in a perceptual decision-making task, and provide a theoretical validation of this framing effect. The research question of the interplay between attention and value-based choice, and the obtained findings, are relevant to a broad research community spanning across cognitive psychology and neuroeconomics. The methodology applied by the authors is sound and the manuscript is well-written.

Despite these merits, one important concern raised by the reviews is that the current manuscript does not provide an accurate description of the literature on this issue. While neuroeconomics does focus on “value” as the quantity being accumulated, existing research does not predict nor claim that “value” is the only quantity to which gaze-contingent effects should apply. Indeed, as raised by reviewer #2, gaze-dependent attention has already been shown to influence evidence accumulation in non-value-based decisions (Tavares et al., 2017), suggesting that this phenomenon is not restricted to value. Furthermore, as raised by reviewer #3, the current study overlaps with (Kovach et al., 2014) which has shown that the influence of attention on choice reverses in a “dislike” frame – something that the authors currently acknowledge only in the Appendix.

Nevertheless, reviewers agree that the aDDM has become dominant in neuroeconomics when it comes to modeling the interplay of attention, valuation and decision making. And so the current study conveys the important and timely message to the neuroeconomics community that attention is not primarily driven by values but by goals. To convey this message accurately, we think that a more balanced description of the existing literature is warranted in the Introduction and Discussion sections (including the work that provides clues toward the findings obtained by the authors). This will not significantly reduce the novelty of the study (which is not the main factor driving our evaluation in any case), but will provide a more accurate picture of the field and thus the key contributions of the study. The main strengths of the current study are the experimental design and the robust cognitive modeling methods which together provide an unambiguous answer to the research question addressed by the authors.

Beyond this first concern regarding the literature/theoretical framing of the study, there are other concerns shared by the reviewers, listed below, that the authors should address in a detailed point-by-point response before considering the manuscript as suitable for publication at *eLife*.

Main concerns (beyond the first concern raised above):

1) Effect of last fixation on framing effects. After reading the manuscript, it appears possible that the framing effects are entirely driven by the last fixation. That is, the increase in choice probability of the more disliked item might be entirely driven by the fact that the last fixation is usually made on the chosen option. Therefore, it would be important to repeat the analysis and exclude the last fixation to see whether the effects still exist when considering only middle and first fixations. It would also be very interesting to see how the effects develop over time within a trial – i.e., at which fixation number (or time point) the influence of attention on choice is strongest – or at which fixation number (or time point) the probability to fixate on the goal-relevant option is highest.

2) Optimal information acquisition model. The reviewers have found the proof of the optimal information acquisition model in the Appendix extremely hard to follow. Reviewer #3 rightly states that all of this may be fairly standard in econometrics, but *eLife* is a journal read by many life scientists, including neuroscientists and psychologists. *eLife* readers will have a very hard time following the proof. It is nice that the main text provides a general intuition of the proof, but the proof in the Appendix should also be amended with some more explanations and intuitions. Notations like *μ_i1_*and *μ_i2_*and then *μ_i1, 2_*(i.e., three different things that have almost exactly the same notation) are not very helpful. Regarding the optimal information acquisition model, there are two recent preprints that make a very similar argument, which is that it is optimal to allocate attention on the most promising choice candidates in the case of multi-alternative decision making: Callaway et al., PsyArXiv; and Jang et al., bioRxiv. These preprints should be cited, since they do not reduce in any way the novelty and appeal of the current study.

3) Effect of accumulation asymmetries on confidence. The authors nicely show that GLAM is only capable of reproducing the influence of sum(Value) / sum(Dots) by assuming the presence of an attention bias. The authors conclude that this indicates that the confidence effect may be "caused by the asymmetries in the accumulation process generated by visual attention". The authors should be more explicit to unpack their reasoning. The reason GLAM (with bias) can account for the confidence effect is the multiplicative interaction of attention and valuation: If there are two high-value items (let's say +10 and +10), then an attention bias of 0.3 leads to accumulation rates of 10 vs. 10*0.3 = 3, so a difference of 7. But if there are two low-value items (e.g., +1 and +1), then the difference is only 0.7. Thus, as soon as there is an imbalance in allocated attention between two options, the difference in accumulation rates between those two options is higher if the options have higher values. This will eventually lead to a larger effect on δ(e).

4) Effect of hunger on value-based decisions. Hunger likely plays an important factor to turn this from a cognitive task to a value-based task. However, other than stating that participants were instructed to fast for four hours, I found no mention of measures used to validate hunger. Is a four hour fasting window enough to make people feel hungry? If so, how was participant hunger measured and/or controlled?

5) Participant drop-out rate. The participant drop-out rate seems extremely high (~20-25%), and seems mainly due to difficulties with how the BDM scale was interpreted. With such a high overall drop-out rate, how can the authors be sure that participant responses on this measure are reliable (even the participants who were included in the analyses)? The authors should discuss explicitly this high drop-out rate.

---

## [Author Response]

Despite these merits, one important concern raised by the reviews is that the current manuscript does not provide an accurate description of the literature on this issue. While neuroeconomics does focus on “value” as the quantity being accumulated, existing research does not predict nor claim that “value” is the only quantity to which gaze-contingent effects should apply. Indeed, as raised by reviewer #2, gaze-dependent attention has already been shown to influence evidence accumulation in non-value-based decisions (Tavares et al., 2017), suggesting that this phenomenon is not restricted to value. Furthermore, as raised by reviewer #3, the current study overlaps with (Kovach et al., 2014) which has shown that the influence of attention on choice reverses in a “dislike” frame – something that the authors currently acknowledge only in the Appendix.Nevertheless, reviewers agree that the aDDM has become dominant in neuroeconomics when it comes to modeling the interplay of attention, valuation and decision making. And so the current study conveys the important and timely message to the neuroeconomics community that attention is not primarily driven by values but by goals. To convey this message accurately, we think that a more balanced description of the existing literature is warranted in the Introduction and Discussion sections (including the work that provides clues toward the findings obtained by the authors). This will not significantly reduce the novelty of the study (which is not the main factor driving our evaluation in any case), but will provide a more accurate picture of the field and thus the key contributions of the study. The main strengths of the current study are the experimental design and the robust cognitive modeling methods which together provide an unambiguous answer to the research question addressed by the authors.

Following your advice we have amended the Introduction and the Discussion of our paper to give a more balanced representation of the background work

Introduction:

“The most common interpretation is that attention is allocated to items based on their value and that looking or attending to an option boosts its value, either by amplifying it or by shifting it upwards by a constant amount. […] In the *like* frame, they had to indicate which snack they would like to consume at the end of the experiment; this is consistent with the standard tasks used in value-based decision studies.”

We have also made changes in our Discussion to include a more detailed revision of the Kovach et al., 2014 and Frömer et al., 2019 studies. Please see below for further details on other changes:

Discussion:

“Our findings speak in favour of a more general role played by attention in prioritising the information needed to fulfil a behavioural goal in both value and perceptual choices (Gottlieb et al., 2012; Kovach et al., 2014; Glickman et al., 2018). Importantly, the seeking of goal-relevant information is observed along the trial, opposing the assumption that attentional sampling is random except for the last fixation (Krajbich et al., 2010, Krajbich and Rangel, 2011; see Gluth et al., 2018; 2020, for additional support for this idea). Pavlovian influences have been proposed to play a key role in the context of accept /reject framing manipulation.”

“Notable exceptions are two recent studies from Frömer and colleagues and Kovach and colleagues. […] However, both studies were developed considering only a standard appetitive *like* frame (Krajbich et al., 2010 study was used as benchmark in both cases).”

Main concerns:1) Effect of last fixation on framing effects. After reading the manuscript, it appears possible that the framing effects are entirely driven by the last fixation. That is, the increase in choice probability of the more disliked item might be entirely driven by the fact that the last fixation is usually made on the chosen option. Therefore, it would be important to repeat the analysis and exclude the last fixation to see whether the effects still exist when considering only middle and first fixations. It would also be very interesting to see how the effects develop over time within a trial – i.e., at which fixation number (or time point) the influence of attention on choice is strongest – or at which fixation number (or time point) the probability to fixate on the goal-relevant option is highest.

We thank the reviewers for their acute observation. We conducted further analyses that have shown that the effect we report is not only driven by the last fixation. We present below the results of the 3 new analyses we have conducted (and which are now reported in our manuscript and Appendix):

1) First of all, as per their suggestion we have excluded the final two fixations from each trial. Note that we excluded two last fixations rather than only the last fixation, because this avoids statistical artifacts. If the last fixation is mostly on the chosen item (as it is) than the fixation before last one is on the non-chosen one and eliminating both is more balanced with regards to the differential amount of time (∆DT) the subjects looked on the two items. We then repeated the hierarchical regression analysis for choice, removing the last *two* fixations (note that trials with 3 or less fixations were discarded from this analysis). This analysis mirrors our original findings showing that gaze is preferentially allocated towards the chosen item (∆DT effect in *like: z*=10.54, *p*<0.001; *dislike*: *z*=8.87, *p*<0.001; ∆DT effect in *most: z*=8.39 *p*<0.001; *fewest*: *z*=9.15, *p*<0.001). We have included it in the Appendix 2—figure 3 and we mentioned the analysis in the main text.

2) Following the thoughtful suggestion of the reviewer we have now conducted a new analysis that explore the temporal evolution of the effect we reported. Using an approach similar to Kovach et al., 2014, we segmented the time series of all the trials in samples of 10ms. We then computed the Pearson correlation between gaze (left = 0 and right = 1) and the difference in evidence (∆Value or ∆Dots) for each time sample. The time series were locked to the beginning of the trial. These data clearly show that, from early on in the trial (~ 1000 ms) gaze distribution was correlated with the goal-relevant item for that trial (e.g. most *liked* item in like frame and less liked item in dislike frame). An almost identical pattern emerged in the Perceptual Experiment. We modified figure 3 (new panels C and D) to include this new compelling result and amended the main section of the manuscript.

3) Finally, we have analysed in more detail the middle fixations. Previous studies (Krajbich et al., 2010; Krajbich and Rangel, 2011, Tavares et al., 2017) have reported that the duration of middle fixations increases when the fixated item has higher value that the unfixated options. We show that in our experiments, in *like* and *most* frames, we replicate this same pattern of results. Critically (and according to our hypothesis), in *dislike* and *fewest* frames, this effect is flipped: the duration of middle fixations decreased as the value (or number of dots) of the fixated item increases. We have now included these new results in the appendix (Appendix 3—figure 4) and we mention them in the main text. We are very grateful to the reviewers for encouraging us to perform these further checks that we believe have substantially improved our manuscript.

The name of section 2.2 has been changed from “Last fixation in choice” to “Fixation effects in choice” and the text has been amended as follow:

“2.2 Fixations effects in choice

An important prediction of attentional accumulation models is that the chosen item is generally fixated last (unless that item is much worse than the other alternative), with the magnitude of this effect related to the difference in value between the alternatives. […] Note that these results are in line with the ones reported by Kovach et al., 2014. We see a very similar pattern of results in the Perceptual Experiment too (Figure 3D).”

In Materials and methods section we have included the details for the time course analysis:

“4.5 Data Analysis: Behavioural Data

Behavioural measures during *like/dislike* and *most/fewest* frames were compared using statistical tests available in SciPy. […] False discovery rate (FDR) was used to correct for multiple tests the P-values obtained from the permutation test (α ≤0.01).”

2) Optimal information acquisition model. The reviewers have found the proof of the optimal information acquisition model in the Appendix extremely hard to follow. Reviewer #3 rightly states that all of this may be fairly standard in econometrics, but eLife is a journal read by many life scientists, including neuroscientists and psychologists. eLife readers will have a very hard time following the proof. It is nice that the main text provides a general intuition of the proof, but the proof in the Appendix should also be amended with some more explanations and intuitions. Notations like μi1 and μ_i2_ and then μ_i1_,_2_ (i.e., three different things that have almost exactly the same notation) are not very helpful. Regarding the optimal information acquisition model, there are two recent preprints that make a very similar argument, which is that it is optimal to allocate attention on the most promising choice candidates in the case of multi-alternative decision making: Callaway et al., PsyArXiv; and Jang et al., bioRxiv. These preprints should be cited, since they do not reduce in any way the novelty and appeal of the current study.

Thank you for highlighting this issue. We have substantially rewritten and expanded the proof of Propositions 1 and 2. We have strived to make it more accessible for a wider audience, while at the same time maintaining a sufficient rigour expected for a proof in the field of economic decision theory. Also, to reduce the confusion with notation, we have added a paragraph at the beginning introducing our notation.

For this same reason, we have not included the proof in the main text but reported the Appendix 9 for the interested reader.

“Appendix 9: Normative Model -Proof of Propositions 1 and 2

All the uses of µ in this proof: µ_i_ is the mean of the belief on the value of item i after the agent has acquired one signal about item i; µ'_i_ is the mean of the belief on the value of item i after the agent has acquired two signals about item i. […] The proof of Proposition 2 is identical once we replace ij by iN+1−j for j=1,…,N. Intuitively, the problem of maximizing the expected utility of the remaining items is strategically equivalent to the problem of choosing the lowest item, which, in turn, is symmetric to the problem of choosing the best item. QED.**”**

Additionally, we thank the reviewer for pointing out to these relevant preprints that had escaped our notice, we have included them in our Discussion:

“To gain a deeper insight into our findings we developed a normative model of optimal information acquisition rooted in economic decision theory. […] However, both studies were developed considering only a standard appetitive *like* frame (Krajbich et al., 2010 study was used as benchmark in both cases).”

3) Effect of accumulation asymmetries on confidence. The authors nicely show that GLAM is only capable of reproducing the influence of sum(Value) / sum(Dots) by assuming the presence of an attention bias. The authors conclude that this indicates that the confidence effect may be "caused by the asymmetries in the accumulation process generated by visual attention". The authors should be more explicit to unpack their reasoning. The reason GLAM (with bias) can account for the confidence effect is the multiplicative interaction of attention and valuation: If there are two high-value items (let's say +10 and +10), then an attention bias of 0.3 leads to accumulation rates of 10 vs. 10*0.3 = 3, so a difference of 7. But if there are two low-value items (e.g., +1 and +1), then the difference is only 0.7. Thus, as soon as there is an imbalance in allocated attention between two options, the difference in accumulation rates between those two options is higher if the options have higher values. This will eventually lead to a larger effect on δ(e).

We thank the reviewers for suggesting us to unpack the effect of the attention bias in generating the asymmetric effect in the GLAM model. We have modified the paragraph in the Results, section 2.4.2 (Balance of Evidence and Confidence) to be more explicit in the presentation of the role of the multiplicative effect on the results.

“Overall, these results show how the model is capable of capturing the novel empirical effect on confidence we identified experimentally, giving computational support to the hypothesis that goal-relevant evidence is fed to second order processes like confidence. It also hints at a potential origin to the effects of the sum of evidence (i.e.,ΣValue,ΣDots) on confidence: asymmetries in the accumulation process, in particular the multiplicative effect of attention over accumulation of evidence, may enhance the differences between items that are more relevant for the frame. This consequentially boosts the level of confidence that participants have in their decisions.”

And we also included the following in the Discussion:

“In both experiments, the incorporation of goal-relevant evidence to fit the GLAM resulted in a better model fit compared with the model in which the value or perceptual evidence was integrated independently of the frame. […] Further empirical data will be required to test this idea more stringently.”

4) Effect of hunger on value-based decisions. Hunger likely plays an important factor to turn this from a cognitive task to a value-based task. However, other than stating that participants were instructed to fast for four hours, I found no mention of measures used to validate hunger. Is a four hour fasting window enough to make people feel hungry? If so, how was participant hunger measured and/or controlled?

We thank the reviewer for highlighting this point. In our study we did not acquire extra measures to account for the hunger level participants experienced during the experiment. The methodology presented here (i.e. asking participants to fast for a couple of hours before the task) is currently the state of the art in the studies using food snack as value items (just to mention a few: Krajbich et al., 2010; Krajbich and Rangel., 2011; Lim et al., 2011; De Martino et al., 2013; Polania et al., 2014; Folke et al., 2016; Tarantola et al., 2017; Bakkour et al., 2019; Polania et al., 2019). Virtually all these studies do not report additional measures to control for participants hunger. One exception is one of the studies conducted in our lab by Folke et al., 2016, which also used a 4 hour fast, as in our experiment. In that case, we collected blood samples in all participants to measure glucose levels previous to the experiment. It was reported that fasting participants presented levels of glucose comparable to the ones expected for fasting adults. However, when we compared these results with other results from our lab (and other labs) in which the blood test was not performed we discovered that there was virtually no difference in the pattern of behaviour between the two situations since participants showed a good level of compliance even in absence of test. For this reason, we decided in further studies (including this one) to not measure blood glucose level. This is an invasive procedure that was not justified from an ethical point of view given the lack of difference we observed. In any case, it is important to highlight that the main purpose behind making participants hungry was to get them engaged in the task, and willing to report their genuine preferences. Note that most participants were happy to pay out of their reimbursement a price that was significantly higher than the price that they would have paid for same product in a retail shop. This in order to eat the item in the hour that they were forced to stay (after fasting a total 5.5 hours: 4 hours before + 1.5 hour of the experiment). This provide a further indirect suggestion that they were not engaged simply in a cognitive task of price estimation. Finally, the fact that our perceptual study shows virtually an identical pattern of result provide further evidence that the effect we observe is likely to affect in general how evidence is collected and not value specific effect. In a way this is the main take home message of our study.

5) Participant drop-out rate. The participant drop-out rate seems extremely high (~20-25%), and seems mainly due to difficulties with how the BDM scale was interpreted. With such a high overall drop-out rate, how can the authors be sure that participant responses on this measure are reliable (even the participants who were included in the analyses)? The authors should discuss explicitly this high drop-out rate.

We thank the reviewers for this raising this issue. Here we have followed the criteria employed in previous studies in the field with a comparable number of % of subject exclusions. We apologize if our description of exclusion criteria gave the wrong impression that most of the exclusions were due to difficulty in using the BDM scale. This is not the case since the number of participants excluded because of misuses of the BDM scale was actually only 2. In our experiments, one of the main reasons for exclusion was criteria 4, i.e. participants reported exactly the same confidence rating for a high proportion of their choices. Specifically, 4 participants in Value and 3 in Perceptual Experiment were excluded by this reason. This is not unusual for studies measuring confidence (e.g. Folke et al., 2016; Fleming et al., 2018; Rollwage et al., 2018; Mazor et al., 2020) since it has been often described that some participants tend to report extremely low variability in using confidence scale (e.g. report all choices as high confidence.) That makes impossible to use that confidence as a useful dependent variable.

Another main reason to remove participants was the difficulties in the eye-tracking procedure (e.g. the participant was restless and the tracking was lost in some trials or there were many blinking events). Overall, 3 participants were excluded by eye-tracking problems. We checked that participants were giving reliable responses assessing their performance (i.e. they were reporting the item rated with higher/lower BDM value or number of dots, depending on the frame). We excluded 2 participants with accuracy issues in the Value Experiment and 4 participants in the Perceptual Experiment. In both experiments the excluded participants had performance close to chance level (50%) or they did not follow correctly the instructions being unaware of the change in frame. Please notice that the participants included in the study had, overall, performance around 75%.

Following the reviewer suggestion, we have now been clearer on the reasons for exclusion and we have included a detailed account of the criteria that caused the exclusion of the participants in the Materials and methods subsection 4.2 Exclusion criteria.